# Unique roles of co-receptor-bound LCK in helper and cytotoxic T cells

Veronika Horkova ●[1], Ales Drobek ●[1], Darina Paprckova ●[1],
Veronika Niederlova ●[1], Avishek Prasai[1], Valeria Uleri ●[1], Daniela Glatzova[1],
Markus Kraller[2], Michaela Cesnekova[1], Sarka Janusova[1], Eva Salyova[1],
Oksana Tsyklauri ●[1], Theresa A. Kadlecek[3], Katerina Krizova[1], René Platzer[2],
Kilian Schober[4,5], Dirk H. Busch ●[4], Arthur Weiss[3], Johannes B. Huppa ●[2] &
Ondrej Stepanek ●[1]✉

The kinase LCK and CD4/CD8 co-receptors are crucial components of the
T cell antigen receptor (TCR) signaling machinery, leading to key T cell fate
decisions. Despite decades of research, the roles of CD4–LCK and CD8–
LCK interactions in TCR triggering in vivo remain unknown. In this study,
we created animal models expressing endogenous levels of modified LCK
to resolve whether and how co-receptor-bound LCK drives TCR signaling.
We demonstrated that the role of LCK depends on the co-receptor to which
it is bound. The CD8-bound LCK is largely dispensable for antiviral and
antitumor activity of cytotoxic T cells in mice; however, it facilitates CD8[+]
T cell responses to suboptimal antigens in a kinase-dependent manner.
By contrast, the CD4-bound LCK is required for efficient development
and function of helper T cells via a kinase-independent stabilization of
surface CD4. Overall, our findings reveal the role of co-receptor-bound
LCK in T cell biology, show that CD4- and CD8-bound LCK drive T cell
development and effector immune responses using qualitatively different
mechanisms and identify the co-receptor–LCK interactions as promising
targets for immunomodulation.

Activation of the T cell antigen receptor (TCR) with its cognate peptide–
major histocompatibility complex (pMHC) triggers adaptive immune
responses to infection and cancer but is also involved in autoimmunity.
Cytotoxic CD8[+] and helper CD4[+] T cells have different functions in the
immune system, but their TCR signaling pathways are very similar. One
key difference is the usage of CD8 or CD4 invariant co-receptors rec-
ognizing MHC class I and MHC class II, respectively. Both co-receptors
interact with a Src-family kinase, LCK, which initiates TCR signal trans-
duction inside the cells by the TCR–CD3 complex[1].

The importance of the interactions between CD4 and CD8
co-receptors with LCK for T cell biology has been studied for decades[2,3]
using indirect techniques, including mathematical modeling[4–6] and
descriptive microscopy, biophysical and biochemical approaches on
isolated T cells[7–10] and/or analysis of mice expressing a gain-of-function
chimeric CD8.4 co-receptor[5,6,11]. However, the most powerful reverse
genetics approach, that is, phenotyping of primary T cells with a genetic
disruption of the CD4–LCK and/or CD8–LCK interaction, has not been
used. A single study close to this approach proposed the importance

[1]Laboratory of Adaptive Immunity, Institute of Molecular Genetics of the Czech Academy of Sciences, Prague, Czech Republic. [2]Institute for Hygiene
and Applied Immunology, Center for Pathophysiology, Infectiology and Immunology, Medical University of Vienna, Vienna, Austria. [3]Division of
Rheumatology, Rosalind Russell and Ephraim P. Engleman Arthritis Research Center, Department of Medicine, University of California, San Francisco,
CA, USA. [4]Institute for Medical Microbiology, Immunology, and Hygiene, Technical University of Munich, Munich, Germany. [5]Mikrobiologisches
Institut—Klinische Mikrobiologie, Immunologie und Hygiene, Universitätsklinikum Erlangen, Friedrich-Alexander-Universität (FAU) Erlangen-Nürnberg,
Erlangen, Germany. ✉e-mail: ondrej.stepanek@img.cas.cz

of the co-receptor–LCK interactions in the positive selection of MHC class I/MHC class II-restricted T cells, but it did not address the role of this interaction in the immune response[12].

Overall, the contribution of co-receptor–LCK interactions to T cell signaling and eventual fate decisions is still unclear. The intuitive model is that a co-receptor-recruited LCK phosphorylates the TCR–CD3 complex[4,5,13]. An alternative model proposes that this key phosphorylation event is preferentially performed by 'free' LCK[7,9]. In the latter scenario, co-receptor-bound LCK physically stabilizes the TCR–antigen interaction from inside the cell[8]. The experimental in vivo evidence for either of these models is missing.

In this study, we characterized the role of co-receptor-bound LCK in vivo using genetically modified mouse models. The importance and mode of action of co-receptor-bound LCK differs in cytotoxic and helper T cell lineages.

## Results

### Mouse models for studying the co-receptor–LCK interaction

We addressed the physiological relevance of the interaction between LCK and CD4/CD8 co-receptors using reverse genetics in mice. We generated knock-in mouse strains expressing endogenous levels of LCK bearing C20A.C23A (CA) or K273R (KR) amino acid substitutions and LCK-deficient ($Lck$^KO/KO) mice (Extended Data Fig. 1a–c). LCK^CA does not interact with CD4 and CD8 (refs. [12,14]; Extended Data Fig. 1d,e), and, thus, T cells in $Lck$^CA/CA mice rely exclusively on the co-receptor-unbound 'free' LCK. LCK^KR has no enzymatic activity[15], but the putative adaptor function of LCK should be preserved. To uncouple the proposed catalytic and adaptor roles of co-receptor-bound LCK, we produced $Lck$^CA/KR compound heterozygotes expressing one pool of strictly cytoplasmic 'free' LCK^CA together with a pool of kinase-dead LCK^KR interacting with co-receptors (Fig. 1a). If the TCR–CD3 complex is preferentially phosphorylated by 'free' LCK, as proposed previously[7,9], and the co-receptor-bound LCK carries the adaptor function[8], the $Lck$^CA/KR mice should have normal T cell development and function.

We tested the enzymatic activity of the LCK variants in two cell lines. The cotransfection of the mouse $Lck$ variants and their substrates CD247 (TCRζ) or ZAP70 into HEK293 cells showed that LCK^CA and wild-type LCK (LCK^WT) have a comparable activity, whereas LCK^KR lacks the kinase activity, as expected (Extended Data Fig. 2a,b). Accordingly, LCK-deficient Jurkat cells[16] reconstituted with human LCK^WT and LCK^CA showed a comparable phosphorylation of TCRζ and ZAP70 and overall tyrosine phosphorylation after stimulation with anti-TCR, whereas LCK^KR was not able to restore signaling (Extended Data Fig. 2c). Moreover, the $LCK$^KO Jurkat cells expressing OT-I TCR specific to K^b-OVA antigen reconstituted with LCK^WT or LCK^CA showed a comparable response to the antigen (measured as CD69 upregulation), whereas LCK^KR-expressing cells were unresponsive (Extended Data Fig. 2d).

Overall, we generated and validated mouse models tailored to uncover the role of co-receptor-bound LCK in vivo.

### T cell maturation with uncoupled LCK and co-receptors

$Lck$^KO/KO mice exhibited partial blocks at two key stages of T cell development in the thymus (Fig. 1b–d, Extended Data Fig. 3a–c and Supplementary Fig. 1a), as shown previously[17]. First, high frequencies of double-negative (DN) thymocytes (Fig. 1b) and, specifically, CD25^+CD44^− DN3 cells (Extended Data Fig. 3a,b) indicate inefficient pre-TCR signaling during β-selection. Second, low numbers of CD4^+ or CD8^+CD24A^−TCRB^+ (TCRβ^+) mature single-positive (mSP) thymocytes indicate defective positive selection of self-pMHC-restricted T cells (Fig. 1b–d). $Lck$^KR/KR mice showed an even more severe phenotype than the $Lck$^KO/KO mice (Fig. 1b–d and Extended Data Fig. 3c), suggesting that LCK^KR is a dominant-negative variant, preventing the phosphorylation of the TCR–CD3 complex by other kinases, such as FYN[17] (Extended Data Fig. 3d–f and Supplementary Fig. 1b).

$Lck$^CA/CA mice did not show the block at the DN stage (Extended Data Fig. 3a,b). By contrast, $Lck$^CA/CA mice had a low count of mature thymocytes, which was more pronounced in CD4^+ than in CD8^+ mSP thymocytes (Fig. 1b–d). $Lck$^CA/KR mice showed higher numbers of CD4^+ mSP thymocytes than $Lck$^CA/CA mice, but the formation of CD8^+ mSP thymocytes was comparable in these two strains (Fig. 1b–d). These results suggested a kinase-independent role of CD4–LCK, but not CD8–LCK, in thymocyte maturation. Heterozygous $Lck$^WT/KO, $Lck$^WT/KR and $Lck$^WT/CA mice showed normal counts of mature thymocytes, suggesting that a single $Lck$^WT allele is sufficient for proper T cell development (Extended Data Fig. 3c).

The numbers of mature CD4^+ and CD8^+ T cells in the lymph nodes (LNs) reflected their maturation in the thymus (Fig. 1e,f and Extended Data Fig. 3g). The exception was normal numbers of mature CD8^+ T cells in the $Lck$^CA/CA and $Lck$^CA/KR mice, apparently due to lymphopenia-induced proliferation coupled with the generation of CD44^+ antigen-inexperienced memory-like CD8^+ T cells[18,19] in these mice (Extended Data Fig. 3h,i). $Lck$^CA/CA mice showed a slightly higher frequency of FOXP3^+ regulatory T cells among CD4^+ T cells than $Lck$^WT/WT mice, which was reverted in the $Lck$^CA/KR mice (Extended Data Fig. 3j), indicating that regulatory T cells are less dependent on CD4–LCK than conventional T cells.

To study the intrinsic role of LCK variants in the development of CD4^+ and CD8^+ T cells, we generated mixed bone marrow (BM) chimeras by transplanting a 1:1 mixture of BM cells from congenic Ly5.1 mice and $Lck$-variant strains (Ly5.2) into irradiated Ly5.1/Ly5.2 host mice. We observed reduced numbers of peripheral $Lck$^CA/CA CD4^+ and CD8^+ T cells in comparison to WT cells (Fig. 1g,h and Extended Data Fig. 3k). The co-receptor-bound kinase-dead LCK in the $Lck$^CA/KR background partially rescued the numbers of CD4^+ T cells but not CD8^+ T cells (Fig. 1g,h).

Overall, $Lck$^CA/CA mice showed an incomplete block in the maturation of CD4^+ and CD8^+ T cells, which was partially rescued in the $Lck$^CA/KR mice in the CD4^+, but not CD8^+, compartment.

### Role of co-receptor–LCK in double-positive (DP) thymocyte maturation

To elucidate the role of $Lck$ variants in DP thymocytes, we assessed the expression of maturation markers by flow cytometry (Extended Data Fig. 4a,b) followed by unsupervised clustering using self-organizing maps[20,21]. This revealed a cluster of mature CD5, CD69 and TCRβ triple-high DP thymocytes (Fig. 2a and Extended Data Fig. 4c,d). This cluster was the least abundant in $Lck$^KR/KR and $Lck$^KO/KO mice (Fig. 2b). The percentage of mature DP thymocytes was lower in $Lck$^CA/CA mice than in $Lck$^WT/WT mice, which was largely rescued in the $Lck$^CA/KR mice (Fig. 2b). Because the overall expression of the activation markers was relatively high in DP thymocytes of $Lck$^CA/CA and $Lck$^CA/KR mice (Extended Data Fig. 4a,b), the partial development block of the $Lck$^CA/CA mice probably occurs only at the final steps of the maturation of DP thymocytes. Indeed, the comparison of basal phosphorylation levels of TCRζ and ZAP70 at particular differentiation stages showed that TCRβ^low thymocytes experience stronger TCR signaling in the $Lck$^CA/CA and $Lck$^CA/KR mice than in the $Lck$^WT/WT mice, but this difference disappears or even reverses during their maturation into postselection TCRβ^high DP thymocytes and subsequently mature TCRβ^high SP4 and SP8 stages (Extended Data Fig. 4e,f and Supplementary Fig. 1c).

To assess antigenic signaling in thymocytes, we crossed our $Lck$-variant mice with monoclonal OT-I TCR $Rag2$^KO/KO (henceforth OT-I) transgenic mice specific to ovalbumin-derived H2-K^b-SIINFEKL antigen (OVA). We stimulated thymocytes with T2-Kb cells presenting titrated doses of OVA or its altered peptide ligands (APL) with lower affinity to OT-I (Supplementary Fig. 2a). Whereas $Lck$^CA/CA and especially $Lck$^CA/KR SP8 T cells showed weaker responses to low-affinity APLs (T4 and G4) than $Lck$^WT/WT, we did not observe any differences in DP thymocytes (Fig. 2c). Accordingly, $Lck$^CA/CA and $Lck$^CA/KR thymocytes isolated from OT-I $B2m$^KO/KO mice, which are arrested at the preselection DP stage, showed similar (if not slightly increased) response as their $Lck$^WT/WT counterparts (Extended Data Fig. 5a).

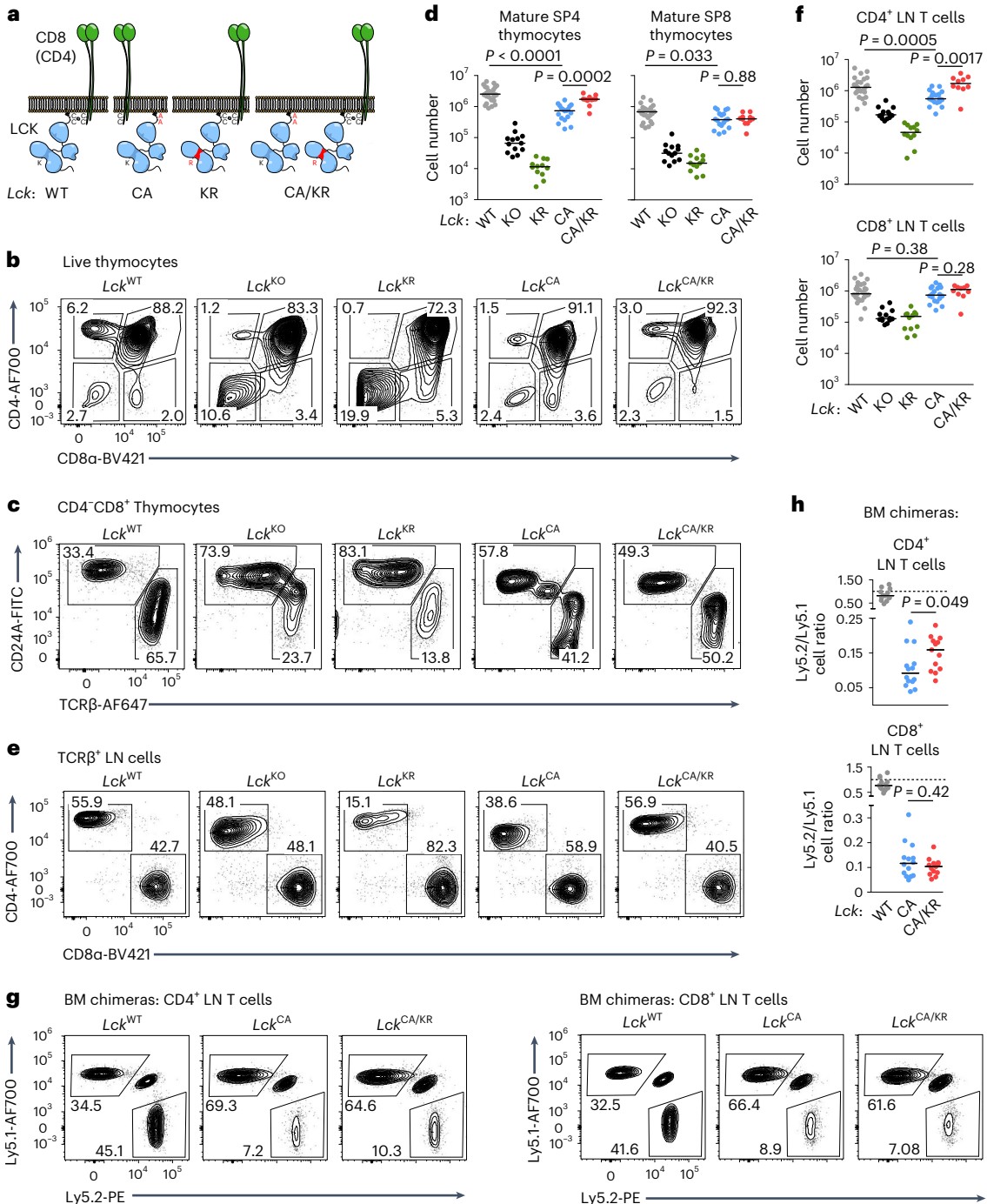

**Fig. 1 | Role of co-receptor–LCK interaction in T cell development. a**, Schematic representation of *Lck*-variant strains including the *Lck*^CA/^KR compound heterozygote. **b**–**d**, Thymocytes from indicated mice were analyzed by flow cytometry. A representative experiment is shown in **b** and **c**. Counts of mature SP4 (TCRβ^+CD24A^−CD4^+CD8α^−) and mature SP8 (TCRβ^+CD24A^−CD4^−CD8α^+) thymocytes in individual mice and medians are shown in **d**; *Lck*^WT/WT^: *n* = 25 mice and 11 independent experiments; *Lck*^KO/KO^: *n* = 13 mice and 7 independent experiments; *Lck*^KR/KR^: *n* = 12 mice and 6 independent experiments; *Lck*^CA/CA^: *n* = 18 mice and 8 independent experiments; *Lck*^CA/KR^: *n* = 11 mice and 5 independent experiments. **e**,**f**, CD4^+ (viable TCRβ^+CD4^+CD8α^−) and CD8^+ (viable TCRβ^+CD4^−CD8α^+) T cells in LNs were analyzed by flow cytometry. A representative experiment is shown (**e**). Cell counts for individual mice and

medians are shown in **f**; *Lck*^WT/WT^: *n* = 25 mice and 11 independent experiments; *Lck*^KO/KO^: *n* = 13 mice and 7 independent experiments; *Lck*^KR/KR^: *n* = 12 mice and 6 independent experiments; *Lck*^CA/CA^: *n* = 18 mice and 8 independent experiments; *Lck*^CA/KR^: *n* = 10 mice and 5 independent experiments. **g**,**h**, BM cells from indicated *Lck*-variant strains mixed with BM cells from congenic Ly5.1 WT mice were transplanted into Ly5.1/Ly5.2 heterozygous mice at a 1:1 ratio; *n* = 13 *Lck*^WT/WT^, *n* = 14 *Lck*^CA/CA^ and *n* = 13 *Lck*^CA/KR^ mice from three independent experiments. The ratio of LN CD4^+ or CD8^+ T cells derived from Ly5.1 and Ly5.2 BM at 8 weeks after transplantation was calculated, and a representative experiment (**g**) and results from individual mice and medians (**h**) are shown. A value of 1.0 is indicated by the dashed line. Statistical significance was calculated using a Mann–Whitney test.

Fetal thymic organ culture experiments with a negative selecting peptide (OVA), a partial negative selector (Q4R7) and a positive selecting peptide (Q4H7) revealed substantial developmental defects in *Lck*^KO/KO^

OT-I *B2m*^KO/KO^ thymocytes but not in *Lck*^CA/CA^ OT-I *B2m*^KO/KO^ thymocytes (Extended Data Fig. 5b and Supplementary Fig. 2b). The positive selector Q4H7 induced less CD8αβ SP cells in the *Lck*^CA/CA^ thymi than in the

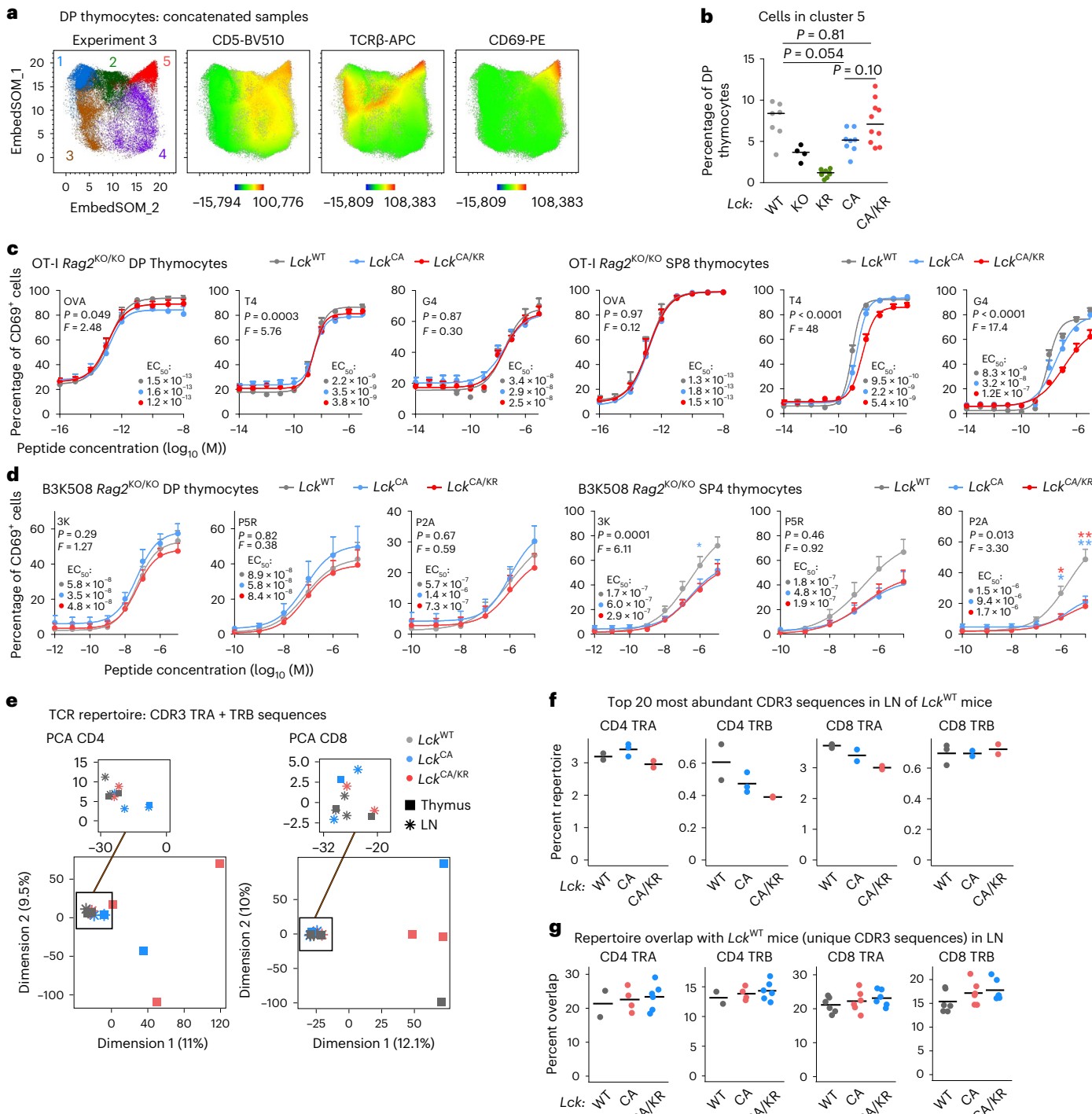

**Fig. 2 | A modest role of co-receptor–LCK interaction in DP thymocytes.**
**a**,**b**, EmbedSOM maps of concatenated DP thymocyte samples from *Lck*-variant mice. **a**, EmbedSOM maps show individual FlowSOM clusters and the relative expression of indicated markers. A representative experiment out of a total of five experiments is shown. **b**, Frequency of cells in cluster 5; $n = 7$ *Lck*[WT/WT], $n = 4$ *Lck*[KO/KO], $n = 9$ *Lck*[KR/KR], $n = 8$ *Lck*[CA/CA] and $n = 10$ *Lck*[CA/KR] in five independent experiments. Medians are shown. Statistical significance was calculated using a Mann–Whitney test. **c**, Thymocytes from indicated *Lck*-variant OT-I mice were activated with T2-Kb cells loaded with the indicated peptides (affinity: OVA > T4 > G4) and analyzed for CD69 expression by flow cytometry; $n = 3$ (OVA and T4) or 4 (G4) independent experiments/mice. **d**, Thymocytes of indicated *Lck*-variant B3K508 mice were activated with Ly5.1 splenocytes loaded with indicated peptides (affinity: 3K > P5R > P2A) and analyzed for CD69 expression by flow cytometry; $n = 8$ (3K and P2A in *Lck*[WT/WT] and *Lck*[CA/CA] mice), $n = 7$ (3K and P2A in *Lck*[CA/KR] mice) or $n = 5$ (P5R) independent experiments/mice. Data in **c** and **d** are shown as mean + s.e.m.

Differences in the EC50 and/or maximum of the fitted non-linear regression curves were tested using an extra sum of squares $F$-test. $F$, $P$ and EC50 values are shown. The significance of the differences between *Lck*[WT/WT] and *Lck*[CA/CA] mice (blue) and *Lck*[WT/WT] and *Lck*[CA/KR] mice (red) at individual concentrations in **d** was calculated using a Mann–Whitney test; *$P < 0.05$; **$P < 0.01$; no symbol, $P > 0.05$ (Supplementary Table 3). **e**–**g**, TCR repertoires of FACS-sorted CD4+ and CD8+ mSP cells in the LNs and thymi from indicated mice were profiled. UMI-corrected counts of TCRα (TRA) and TCRβ (TRB) CDR3 amino acid sequences were normalized after the removal of NKT TRAV11-TRAJ18 CDR3 sequences. Sample sizes are in Supplementary Table 4. **e**, Principal-component analysis of all samples. **f**, Percentage of the repertoire in the indicated samples constituted by the top 20 most frequent CDR3 amino acid sequences in the LNs of *Lck*[WT/WT] mice. **g**, The repertoire overlap was calculated as the percentage of unique CDR3 amino acid sequences in each sample present among the unique CDR3 sequences from each of the (non-identical) *Lck*[WT/WT] mice. Each dot represents a single comparison.

$Lck^{WT/WT}$ thymi (Extended Data Fig. 5b). However, this was compensated by higher numbers of CD8αα SP cells, which are induced by strong signals[22], suggesting that Q4H7 might act as a weak partial negative selector for the preselection DP thymocytes in the $Lck^{CA/CA}$ mice.

To study the role of the CD4–LCK interaction in the signaling of MHC class II-restricted thymocytes, we crossed our collection of the $Lck$-variant mice with TCR transgenic B3K508 $Rag2^{KO/KO}$ (henceforth B3K508) mice specific for H2-A$^b$-bound FEAQKAKANKAKAVD (3K) peptide[23]. The responses of DP thymocytes to Ly5.1 splenocytes presenting 3K or its APLs were comparable among the $Lck^{WT/WT}$, $Lck^{CA/CA}$ and $Lck^{CA/KR}$ strains (Fig. 2d). By contrast, the responses of $Lck^{CA/CA}$ and $Lck^{CA/KR}$ SP4 thymocytes were weaker than those of $Lck^{WT/WT}$ thymocytes (Fig. 2d).

Overall, these data indicate that the signaling in DP T cells is relatively normal in the $Lck^{CA/CA}$ and $Lck^{CA/KR}$ mice and that their T cell developmental defects occur only during the late stages of DP development and during the maturation of SP stages.

To assess how the absence of the co-receptor–LCK interactions shapes the T cell repertoire, we analyzed TCRα and TCRβ sequences in SP thymocytes and peripheral T cells from $Lck^{WT/WT}$, $Lck^{CA/CA}$ and $Lck^{CA/KR}$ mice (Supplementary Tables 1 and 2). We did not observe major differences in TRAV and TRBV usage, with the exception of the enrichment for natural killer T (NKT) cell typical segments TRAV11, TRBV1, TRBV13-2 and TRBV29 (ref. [24]) in the $Lck^{CA/CA}$ and $Lck^{CA/KR}$ SP4 thymocytes (Extended Data Fig. 6a,b). Accordingly, canonical NKT cell TCRα chains, TRAV11-TRAJ18 (Vα14–Jα18), were very abundant in SP4 thymocytes in $Lck^{CA/CA}$ and $Lck^{CA/KR}$ mice but not in peripheral CD4$^+$ T cells (Extended Data Fig. 6c). After removing the NKT sequences, the repertoires of $Lck^{CA/CA}$ and $Lck^{CA/KR}$ mice were slightly less diverse than the repertoires of $Lck^{WT/WT}$ mice (Extended Data Fig. 6d). Principle-component analysis revealed that the TCR repertoires of SP thymocytes in some mice differ from the other samples, but the repertoires of LN cells show only subtle differences and no clear separation of the strains (Fig. 2e and Supplementary Fig. 3). Accordingly, the most abundant peripheral TCR sequences in $Lck^{WT/WT}$ mice were frequent also in $Lck^{CA/CA}$ and $Lck^{CA/KR}$ mice (Fig. 2f and Supplementary Fig. 4). Finally, there was a substantial overlap of individual sequences in peripheral T cells among the three strains (Supplementary Fig. 5). The overlap of individual peripheral TCR sequences between the $Lck^{CA/CA}$ or $Lck^{CA/KR}$ mice and the $Lck^{WT/WT}$ mice was comparable to the overlap between individual $Lck^{WT/WT}$ mice (Fig. 2g).

Overall, the disruption of the co-receptor–LCK interaction does not reduce the development and signaling of DP thymocytes until they reach their final maturation stage. As a result, the peripheral repertoires of $Lck^{CA/CA}$ and $Lck^{CA/KR}$ mice are only minimally affected.

### 'Free' LCK is sufficient for largely adaptive immune responses

To study the effects of LCK variants on T cell function, we examined the $Lck$ strains for their antiviral and antitumor immunity, which is mediated mostly by CD8$^+$ T cells. The ability to clear lymphocytic choriomeningitis virus (LCMV) was comparable in the $Lck^{WT/WT}$ and $Lck^{CA/CA}$ mice, slightly impaired in the $Lck^{CA/KR}$ mice and substantially defective in the $Lck^{KR/KR}$ mice (Fig. 3a). The numbers of CD8$^+$ T cells in the spleen were lower in the infected $Lck^{CA/CA}$ and $Lck^{CA/KR}$ mice than in the $Lck^{WT/WT}$ mice and were further reduced in the $Lck^{KR/KR}$ mice (Extended Data Fig. 7a). A similar reduction was observed in CD4$^+$ T cells, with the notable difference that $Lck^{CA/KR}$ showed a partial rescue compared to $Lck^{CA/CA}$ (Extended Data Fig. 7b). We used D$^b$-GP33 and D$^b$-NP396 tetramers for the detection of CD8$^+$ T cells specific to the immunodominant LCMV epitopes. The frequency of LCMV-specific CD8$^+$ T cells was comparable in the $Lck^{WT/WT}$ and $Lck^{CA/CA}$ mice but was lower in the $Lck^{CA/KR}$ mice (Fig. 3b, Extended Data Fig. 7c and Supplementary Fig. 6a,b). These LCMV-specific T cells had an antigen-experienced phenotype (CD44$^+$CD49d$^+$) and formed a comparable fraction of KLRG1$^+$CD127$^-$ short-lived effectors in these strains (Extended Data Fig. 7d,e and Supplementary Fig. 6c,d). $Lck^{KR/KR}$ mice showed low numbers of LCMV-specific CD8$^+$ T cells, incomplete differentiation into

antigen-activated CD44$^+$CD49d$^+$ T cells and a bias toward the formation of short-lived effectors (Fig. 3b, Extended Data Fig. 7a–e and Supplementary Fig. 6b–d), which explained the defective viral clearance in this strain.

We observed impaired formation of CXCR5$^+$PD-1$^+$CD4$^+$ follicular helper T cells (T$_{FH}$) in the $Lck^{KR/KR}$ and $Lck^{CA/CA}$ mice during LCMV infection, which was partially rescued in the $Lck^{CA/KR}$ mice (Fig. 3c,d and Supplementary Fig. 6e). Only a small percentage of these T$_{FH}$ cells were FOXP3$^+$ follicular regulatory T cells (Extended Data Fig. 7f). The frequencies of CD4$^+$ T cells specific for an immunodominant GP66 LCMV epitope were comparable between the $Lck^{WT/WT}$ and $Lck^{CA/CA}$ mice and were slightly lower in the $Lck^{CA/KR}$ mice (Extended Data Fig. 7g and Supplementary Fig. 6e). The counts of GP66-specific CD4$^+$ T cells were comparable in the $Lck^{CA/CA}$ and $Lck^{CA/KR}$ mice and higher in the $Lck^{WT/WT}$ mice (Extended Data Fig. 7g). These GP66-specific T cells showed defective differentiation into FOXP3$^-$ T$_{FH}$ cells in the $Lck^{KR/KR}$ mice and to a lesser extent in $Lck^{CA/CA}$, but not in $Lck^{CA/KR}$, mice (Fig. 3e). Although the difference between $Lck^{CA/CA}$ and $Lck^{CA/KR}$ was not significant in this small cohort, it corresponded to the overall CD4$^+$ T cell population (Fig. 3d).

The $Lck^{KR/KR}$ and $Lck^{KO/KO}$ mice failed to hamper the growth of MC-38 carcinomas expressing OVA (Fig. 3f,g). The $Lck^{CA/CA}$ and $Lck^{CA/KR}$ mice showed slightly or substantially faster tumor growth than $Lck^{WT/WT}$ mice, respectively (Fig. 3f,g). We did not observe large differences in the number of total T cells in the tumor (Extended Data Fig. 7h and Supplementary Fig. 6f) among the strains. The numbers of antigen-specific K$^b$-OVA tetramer$^+$CD8$^+$ T cells infiltrating the tumor and tumor-draining LNs were comparable among the $Lck^{WT/WT}$, $Lck^{CA/CA}$ and $Lck^{CA/KR}$ mice but were lower in the $Lck^{KR/KR}$ strain (Extended Data Fig. 7i,j and Supplementary Fig. 6g–i). The suboptimal antitumor response in $Lck^{CA/CA}$ and $Lck^{CA/KR}$ mice is probably caused by impaired killing of tumor cells rather than by the absence of tumor-specific T cell clones.

Overall, the $Lck^{CA/CA}$ mice showed relatively normal antiviral and antitumor immune responses, suggesting that the interaction between CD8 and LCK is not essential for these types of immune protection. The $Lck^{CA/KR}$ mice showed defective tumor and viral clearance in these CD8$^+$ T cell-based models but partially rescued the $Lck^{CA/CA}$ phenotype in the CD4$^+$ T$_{FH}$ compartment. This indicated a differential kinase-independent function of CD8- and CD4-bound LCK.

### CD8–LCK promotes responses to suboptimal antigens

To study the roles of CD4- and CD8-bound LCK separately, we used MHC class I-restricted and MHC class II-restricted monoclonal mice. First, we investigated the $Lck$ variants in peripheral CD8$^+$ OT-I T cells. Whereas the $Lck^{KO/KO}$ and $Lck^{KR/KR}$ OT-I mice showed a severe developmental impairment, the $Lck^{CA/CA}$ mice had slightly more SP8 T cells than the $Lck^{WT/WT}$ mice (Fig. 4a–d), which was not observed in the polyclonal setting (Fig. 1d). This is probably connected with slightly stronger signaling of preselection DP thymocytes in the $Lck^{CA/CA}$ mice (Extended Data Fig. 5a,b), the absence of competing MHC class I-independent T cell clones and/or non-physiological regulation of the transgenic TCR expression. The number of SP8 T cells was reduced in the $Lck^{CA/KR}$ mice compared to in $Lck^{CA/CA}$ mice. Peripheral T cell counts were comparable in the $Lck^{WT/WT}$, $Lck^{CA/CA}$ and $Lck^{CA/KR}$ mice (Fig. 4d). We observed a similar phenotype using another MHC class I-restricted TCR transgenic mouse strain F5 $Rag1^{KO/KO}$ (Extended Data Fig. 8a–d).

To analyze the role of CD8-bound LCK in TCR signaling, we activated peripheral OT-I T cells with antigen-presenting cells loaded with OVA peptide or its lower-affinity APLs ex vivo using CD69 upregulation as a readout. The $Lck^{CA/CA}$ OT-I T cells showed a normal response to OVA but a reduced response to low-affinity OVA variants compared to the $Lck^{WT/WT}$ cells (Fig. 4e and Supplementary Fig. 7a). The $Lck^{CA/KR}$ OT-I T cells exhibited even weaker responses than the $Lck^{CA/CA}$ OT-I T cells (Fig. 4e), documenting the inhibitory role of CD8-bound kinase-dead LCK. The upregulation of CD69 and proliferation induced by the

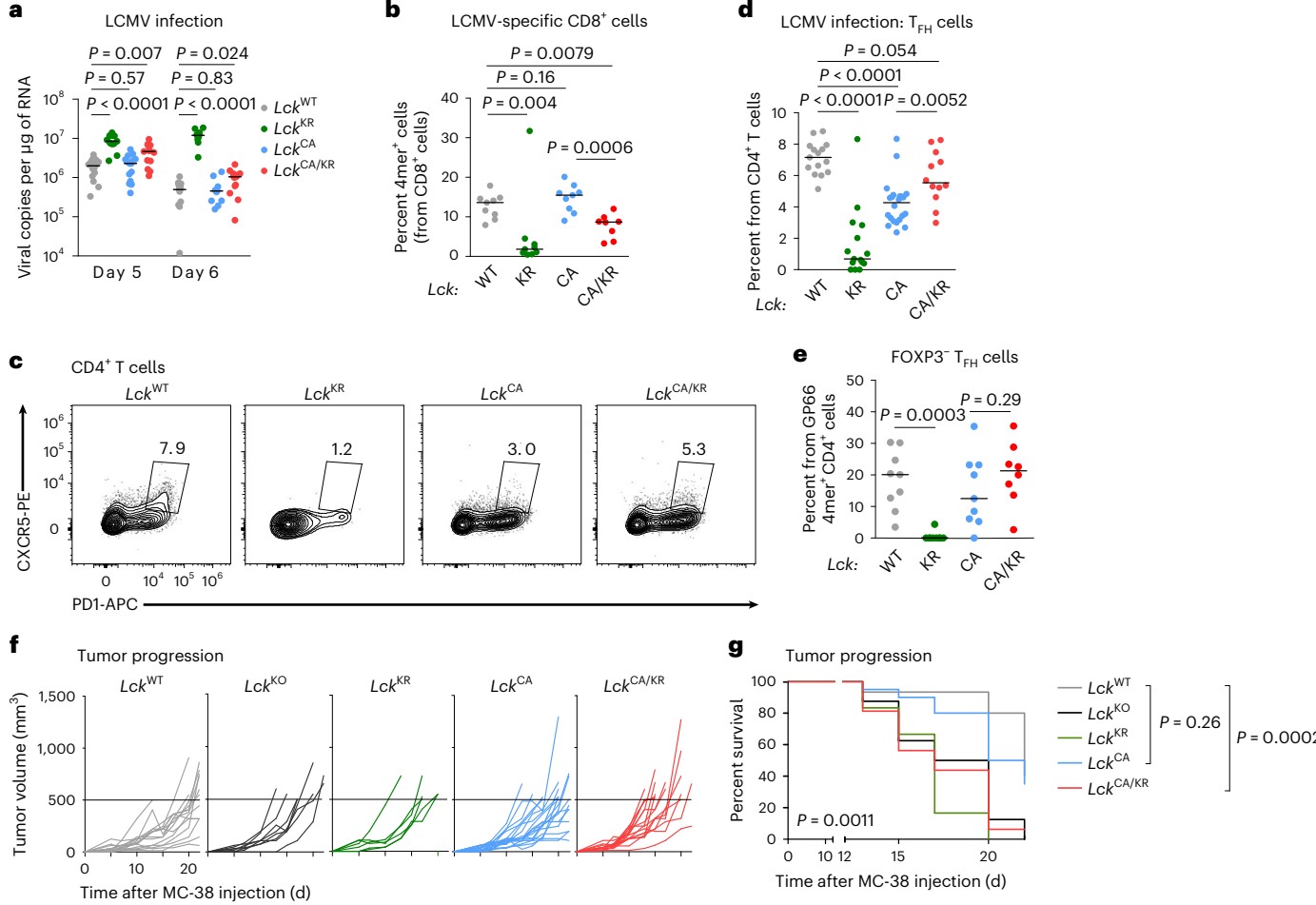

**Fig. 3 | Role of co-receptor–LCK interaction in T cell immunity. a,b,** Indicated *Lck*-variant mice were infected with LCMV. **a,** Viral titers in the spleens were determined by RT–qPCR on day 5 or 6 after infection; $n = 16$ *Lck*[WT/WT], $n = 15$ *Lck*[CA/CA], $n = 13$ *Lck*[CA/KR] and $n = 13$ *Lck*[KR/KR] mice in six independent experiments on day 5; $n = 12$ *Lck*[WT/WT], $n = 11$ *Lck*[CA/CA], $n = 13$ *Lck*[CA/KR] and $n = 10$ *Lck*[KR/KR] mice in five independent experiments on day 6 after infection. Median values are shown. Statistical significance was calculated using a Mann–Whitney test. **b,** The frequency of GP33 4mer+ and NP396 4mer+ cells from CD8+ cells on day 8 after infection is displayed; $n = 8$ (*Lck*[CA/KR]) or 9 (other strains) mice in two (*Lck*[CA/CA]) or three (other strains) independent experiments. Means are shown. Statistical significance was calculated using a Mann–Whitney test. **c,d,** Splenic CD4+ $T_{FH}$ cells were identified on day 8 after infection by flow cytometry. **c,** Representative mice are shown. **d,** Percentages of $T_{FH}$ cells among all CD4+ T cells in the indicated mice are shown. Data show the median values; $n = 15$ *Lck*[WT/WT], $n = 22$ *Lck*[CA/CA],

$n = 12$ *Lck*[CA/KR], and 15 *Lck*[KR/KR] mice in three independent experiments. Statistical significance was calculated using a Mann–Whitney test. **e,** Frequency of FOXP3− $T_{FH}$ cells from GP66 4mer+CD4+ cells on day 8 after infection; $n = 8$ (*Lck*[CA/KR]) or 9 (other strains) mice in two (*Lck*[CA/CA]) or three (other strains) independent experiments. Statistical significance was calculated using a Mann–Whitney test. **f,g,** MC-38 carcinoma cells ($0.5 \times 10^6$) were injected into indicated mice subcutaneously, and tumor growth was monitored; $n = 15$ *Lck*[WT/WT], $n = 20$ *Lck*[CA/CA], $n = 16$ *Lck*[CA/KR], $n = 8$ *Lck*[KO/KO] and $n = 6$ *Lck*[KR/KR] mice in two (*Lck*[KR/KR] mice) or four (other strains) independent experiments. **f,** Tumor growth in individual *Lck*-variant mice is shown. Dashed lines show the endpoint of the experiment (tumor volume of 500 mm³). **g,** Percentage of mice with a tumor smaller than 500 mm³ in time is shown. The statistical significance was tested using a log-rank (Mantel–Cox) test (all groups) and a Gehan–Breslow–Wilcoxon test (individual groups).

co-receptor-independent activation with anti-CD3/CD28 beads were comparable among these three strains (Extended Data Fig. 8e,f).

To separate LCK-dependent and LCK-independent roles of CD8, we analyzed the antigenic response of human OT-I Jurkat cells[25] devoid of CD8 or expressing WT CD8αβ (CD8[WT]) or LCK-binding mutant CD8α[C215.217A]β (CD8[CA])[14]. Jurkat cells expressing CD8[WT] and CD8[CA] showed ~330-fold and ~35-fold lower responses to OVA-pulsed antigen-presenting cells than CD8− cells, respectively (Fig. 4f and Supplementary Fig. 7b). These results indicated that CD8 contributes to T cell activation in LCK-dependent and LCK-independent manners.

To elucidate the role of CD8-bound LCK in the antigenic response in vivo, we adoptively transferred the *Lck*-variant OT-I T cells into congenic Ly5.1 mice followed by infection with transgenic *Listeria monocytogenes* (*Lm*) expressing OVA or its lower-affinity APLs. Whereas there were no large differences in the responses to OVA, the

expansion induced by low-affinity APLs followed the hierarchy *Lck*[WT/WT] > *Lck*[CA/CA] > *Lck*[CA/KR] (Fig. 5a).

We examined the ability of the *Lck*-variant OT-I T cells to hamper tumor progression following their adoptive transfer into T cell-deficient *Cd3e*[KO/KO] mice bearing small MC-38 OVA tumors. The antitumor activity of OT-I cells followed the hierarchy *Lck*[WT/WT] > *Lck*[CA/CA] > *Lck*[CA/KR] (Fig. 5b,c).

It has been proposed that the CD8–LCK interaction might stabilize antigen binding[8]. We assessed the role of CD8–LCK in antigen avidity using three different assays. Whereas the K[b]-OVA and K[b]-T4 tetramer staining (Fig. 5d) and the on-cell $k_{off}$ measurements[26] (Fig. 5e) indicated that CD8–LCK indeed stabilizes the TCR–antigen interaction, two-dimensional (2D) affinity measurements using the antigen nested in a lipid bilayer did not reveal substantial differences (Fig. 5f). Regardless of the slight discrepancies between these methods,

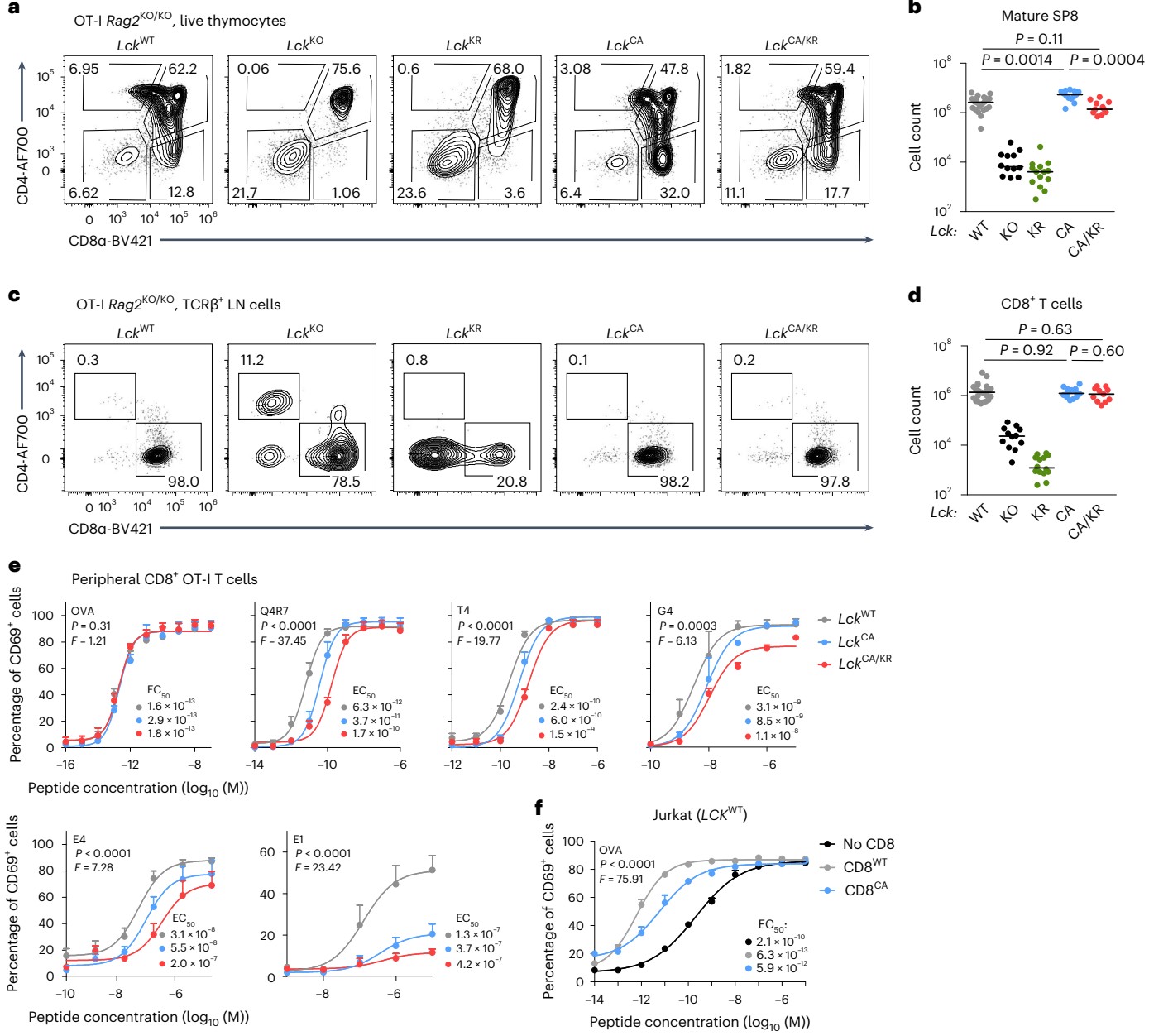

**Fig. 4 | CD8-bound LCK is largely dispensable for positive selection and T cell activation. a–d**, Thymi (**a** and **b**) and LNs (**c** and **d**) of indicated *Lck*-variant OT-I mice were analyzed by flow cytometry. **a**, Expression of CD4 and CD8α in representative mice (gated on viable cells). **b**, Numbers of mSP8 (viable CD4⁻CD8α⁺CD24⁻TCRβ⁺) T cells. Individual mice and medians are shown; $n = 22$ $Lck^{WT/WT}$ mice in 12 independent experiments, $n = 12$ $Lck^{KO/KO}$ mice in 8 independent experiments, $n = 14$ $Lck^{KR/KR}$ mice in 6 independent experiments, $n = 13$ $Lck^{CA/CA}$ mice in 7 independent experiments and $n = 11$ $Lck^{CA/KR}$ mice in 3 independent experiments. The statistical significance was tested using a Mann–Whitney test. **c**, CD4⁺ and CD8⁺ T cells in representative mice (gated on viable TCRβ⁺ cells). **d**, Counts of CD8⁺ T cells. Individual mice and medians are shown; $n = 23$ $Lck^{WT/WT}$ mice in 12 independent experiments, $n = 12$ $Lck^{KO/KO}$ mice in 8 independent experiments, $n = 14$ $Lck^{KR/KR}$ mice in 6 independent experiments, $n = 13$ $Lck^{CA/CA}$ mice in 7 independent experiments and $n = 11$ $Lck^{CA/KR}$ mice in 3 independent experiments. Statistical significance was tested using

a Mann–Whitney test. **e**, T cells isolated from LNs of indicated *Lck*-variant OT-I $Rag2^{-/-}$ mice were activated ex vivo with T2-Kb cells loaded with the indicated peptides (affinity: OVA > Q4R7 > T4 > G4 > E4 > E1) overnight and analyzed for expression of CD69 by flow cytometry. Data are shown as the mean + s.e.m.; $n = 3$ independent experiments/mice for OVA and Q4R7 and $n = 4$ independent experiments/mice for T4, G4, E4 and E1. Differences in the EC₅₀ and/or maximum of the fitted non-linear regression curves were tested using an extra sum of squares $F$-test. $F$, $P$ and EC₅₀ values are shown. **f**, Jurkat cells expressing OT-I TCR and LCK^WT were transduced with CD8^WT or with CD8^CA. These Jurkat cells were activated with T2-Kb cells loaded with OVA peptide overnight and analyzed for CD69 expression by flow cytometry. Data are shown as mean + s.e.m.; $n = 3$ independent experiments/mice. Differences in the EC₅₀ and/or maximum of the fitted non-linear regression curves were tested using an extra sum of squares $F$-test, and $P$, $F$ and EC₅₀ values are shown.

CD8–LCK-mediated stabilization of the TCR–antigen binding does not explain the differences in our functional assays because $Lck^{CA/KR}$ OT-I T cells showed intermediate antigen binding but the weakest antigenic response.

Transgenic mice expressing a chimeric CD8.4 co-receptor (extracellular part from CD8α, intracellular part from CD4) were previously reported to have stronger TCR signaling, leading to altered development and cell fate[5,11,27,28]. These phenotypes were attributed to the

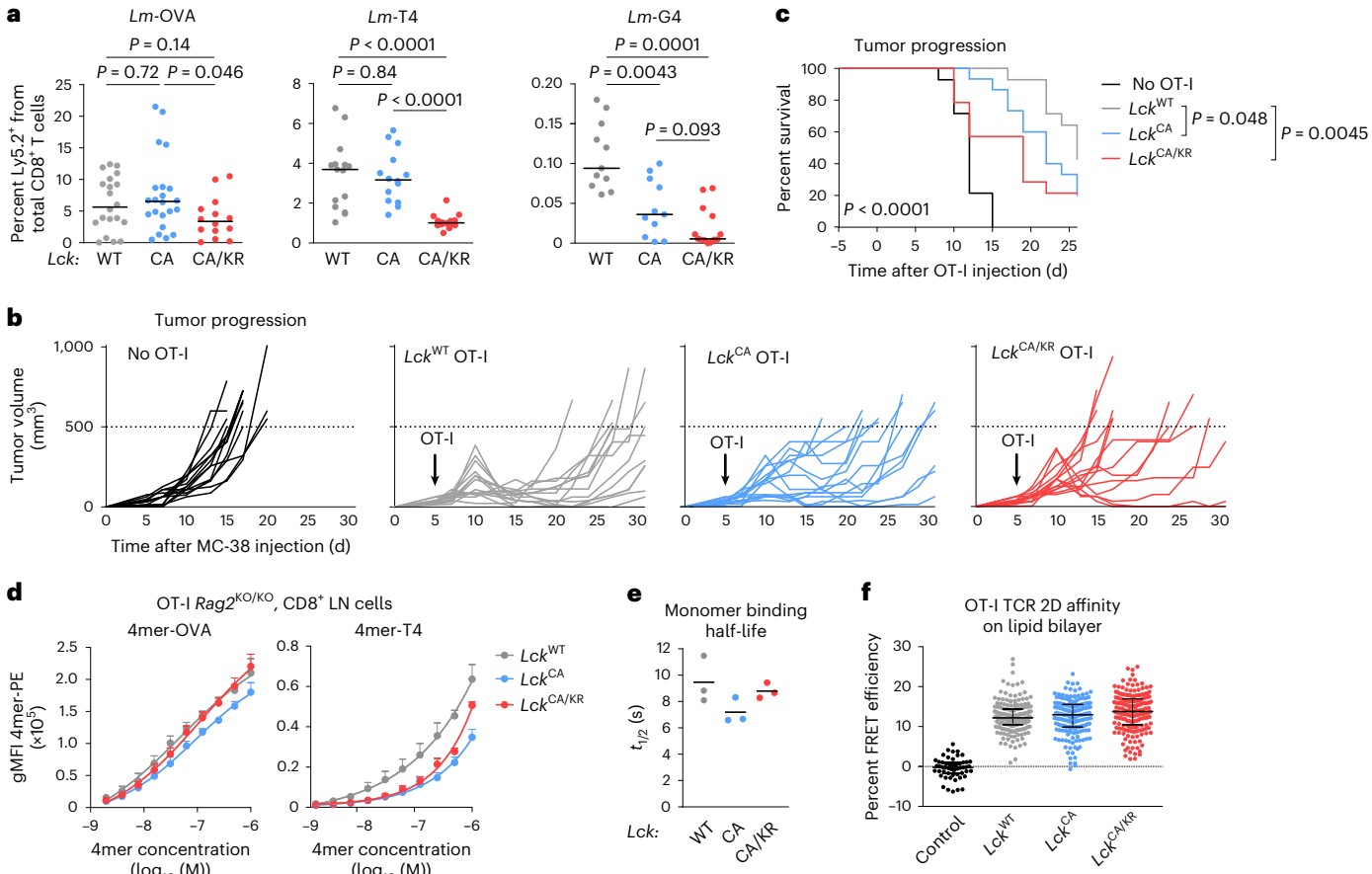

**Fig. 5 | CD8-bound LCK enhances responses to suboptimal antigens in vivo.**
**a**, T cells from indicated *Lck*-variant OT-I mice were transferred into Ly5.1 mice followed by infection with indicated *Lm* strains. The percentage of donor cells among all CD8⁺ T cells on day 5 after infection was quantified by flow cytometry. Medians are shown. Statistical significance was calculated using a Mann–Whitney test; $n = 20$ *Lck*[WT/WT], $n = 22$ *Lck*[CA/CA] and $n = 14$ *Lck*[CA/KR] mice in five (*Lck*[CA/KR]) or seven (other strains) independent experiments for *Lm*-OVA; $n = 15$ *Lck*[WT/WT], $n = 14$ *Lck*[CA/CA] and $n = 14$ *Lck*[CA/KR] mice in four independent experiments for *Lm*-T4; $n = 11$ *Lck*[WT/WT], $n = 11$ *Lck*[CA/CA] and $n = 13$ *Lck*[CA/KR] mice in three independent experiments for *Lm*-G4. **b,c**, MC-38 carcinoma cells ($0.5 \times 10^6$) were injected into *Cd3e*[KO/KO] mice subcutaneously. OT-I T cells ($0.2 \times 10^6$) were adoptively transferred into these mice 5 d later, and the size of the tumor was monitored; $n = 14$ mice for no OT-I, *Lck*[WT/WT] OT-I and *Lck*[CA/KR] OT-I or 15 mice for *Lck*[CA/CA] OT-I in four independent experiments. **b**, Tumor growth in individual *Lck*-variant mice is shown. Dashed lines show the endpoint (tumor volume of 500 mm³). **c**, The percentage of mice with a tumor smaller than 500 mm³ in time is shown. The statistical significance was tested using a log-rank (Mantel–Cox) test (all groups) and a Gehan–Breslow–

Wilcoxon test (individual groups). **d**, CD8⁺ T cells from indicated *Lck*-variant OT-I mice were stained with a dilution series of fluorescently labeled K[b]-OVA and K[b]-T4 tetramer and analyzed by flow cytometry. Mean values + s.e.m. are shown; $n = 4$ (*Lck*[WT/WT], *Lck*[CA/CA] for K[b]-OVA) or $n = 3$ (other samples) independent experiments/mice; gMFI, geometric mean fluorescent intensity. **e**, CD8⁺ T cells from indicated *Lck*-variant OT-I mice were stained with K[b]-OVA-streptactin multimers. Dissociation of K[b]-OVA monomer after the addition of free biotin was measured by flow cytometry. Individual experiments and means are shown; $n = 3$ independent experiments/mice. **f**, CD8⁺ T cells from indicated *Lck*-variant OT-I mice were stained with AF555-labeled anti-TCRβ scF$_v$ and added to planar supported lipid bilayers with ICAM-1 and AF647-labeled K[b]-OVA monomers. Relative TCR:K[b]-OVA occupancy was measured as a fluorescence resonance energy transfer between the donor and acceptor fluorophores. *Lck*[WT/WT] OT-I T cells adhered to the lipid bilayer without K[b]-OVA monomers were used as a negative control. Individual cells, medians and interquartile ranges are shown; $n = 54$ control, $n = 189$ *Lck*[WT/WT], $n = 179$ *Lck*[CA/CA] and $n = 184$ *Lck*[CA/KR] cells in two independent experiments.

supraphysiological coupling of CD8.4 with LCK, but a formal proof was missing. We compared CD8[WT] and CD8.4 OT-I mice on the *Lck*[WT/WT] and *Lck*[CA/CA] background. We observed two previously published phenotypes of CD8.4: a bias toward SP4 differentiation in the thymus[28] and massive formation of antigen-inexperienced memory T cells[27] in the *Lck*[WT/WT], but not *Lck*[CA/CA], mice (Extended Data Fig. 8g,h). This implied that the gain of function of the CD8.4 allele is mediated via LCK binding.

Overall, these data suggested that the interaction of CD8 with LCK is dispensable for the responses to high-affinity antigens but enhances signaling to suboptimal antigens. The kinase-dead LCK coupled to CD8 downmodulates the T cell response to suboptimal antigens.

## LCK supports surface CD4⁺ and helper T cell responses
We studied the role of the CD4–LCK interactions using B3K508 mice. *Lck*[KO/KO] and *Lck*[KR/KR] B3K508 thymocytes showed a developmental

block (Fig. 6a–d). In contrast to the polyclonal mice (Fig. 1b,d), the *Lck*[CA/CA] and *Lck*[CA/KR] mice had comparable (or even slightly higher) counts of SP4 and peripheral CD4⁺ T cells as the *Lck*[WT/WT] mice (Fig. 6a–d). These results indicated that the CD4–LCK interaction is not required for the commitment of MHC class II-restricted T cells to the CD4⁺ T cell lineage.

The *Lck*[CA/CA] B3K508 T cells exhibited weaker ex vivo antigenic responses to the cognate 3K peptide and its intermediate- and low-affinity APLs (P5R and P2A, respectively) than the *Lck*[WT/WT] B3K508 T cells (Fig. 6e). The *Lck*[CA/KR] B3K508 T cells partially rescued defective responses to high-affinity 3K and intermediate-affinity P5R antigens but not to a low-affinity antigen P2A (Fig. 6e). The upregulation of CD69 and proliferation induced by the co-receptor-independent activation with anti-CD3/CD28 beads were comparable among these three strains (Extended Data Fig. 9a,b).

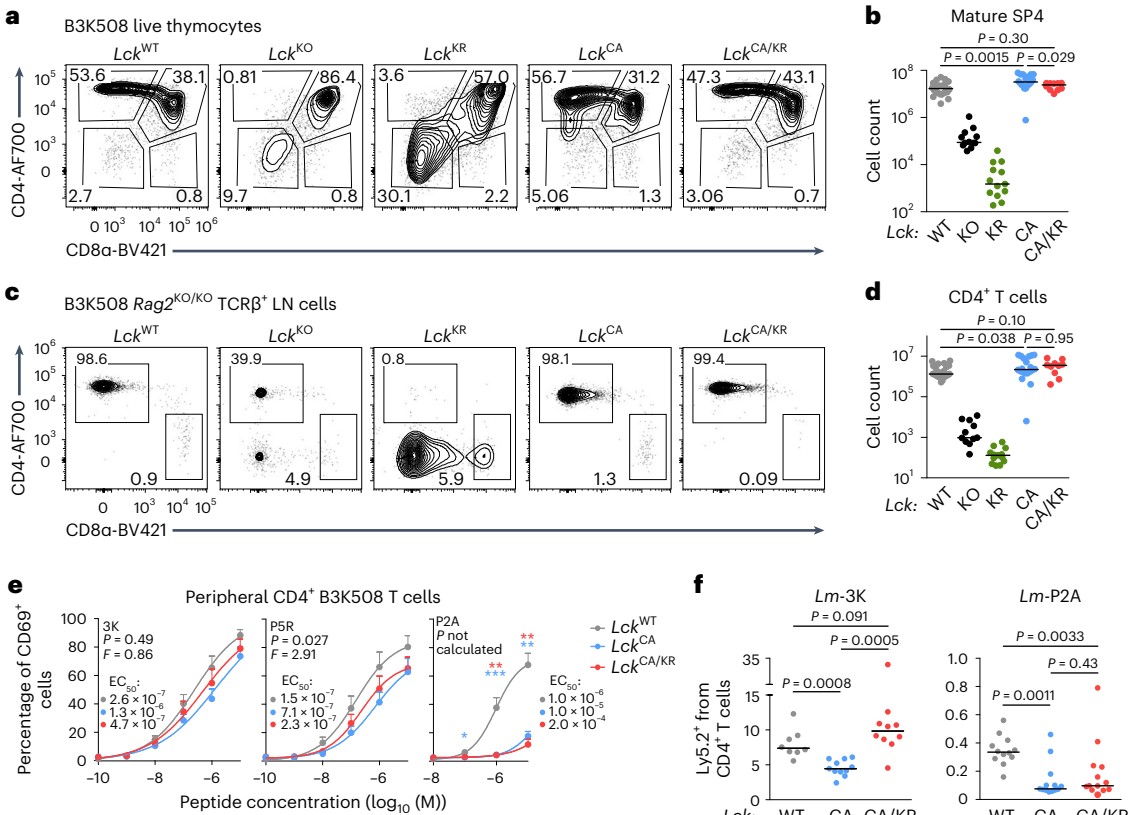

**Fig. 6 | Kinase-dependent and kinase-independent roles of CD4-bound LCK in T cell responses. a–d**, Thymi (**a** and **b**) and LNs (**c** and **d**) of indicated B3K508 mice were analyzed by flow cytometry. Statistical significance was calculated using a Mann–Whitney test. **a**, Expression of CD4 and CD8 in representative mice (gated on viable cells). **b**, Numbers of mSP4 (viable CD4⁻CD8α⁺CD24⁻TCRβ⁺) T cells; $Lck^{WT/WT}$: $n = 24$ mice in 10 independent experiments; $Lck^{KO/KO}$: $n = 12$ mice in 7 independent experiments; $Lck^{KR/KR}$: $n = 13$ mice in 6 independent experiments; $Lck^{CA/CA}$: $n = 19$ mice in 8 independent experiments; $Lck^{CA/KR}$: $n = 10$ mice in 3 independent experiments. Individual mice and medians are shown. **c**, CD4⁺ and CD8⁺ T cells in representative mice (gated on viable TCRβ⁺ cells). **d**, Numbers of CD4⁺ T cells in individual mice and medians are shown; $Lck^{WT/WT}$: $n = 26$ mice in 11 independent experiments; $Lck^{KO/KO}$: $n = 11$ mice in 7 independent experiments; $Lck^{KR/KR}$: $n = 13$ mice in 6 independent experiments; $Lck^{CA/CA}$: $n = 20$ mice in 9 independent experiments; $Lck^{CA/KR}$: $n = 10$ mice in 3 independent experiments. **e**, T cells isolated from LNs of indicated $Lck$-variant B3K508 mice were activated ex vivo with splenocytes from Ly5.1 mice loaded with 3K peptide or APLs with decreasing affinity (3K > P5R > P2A) overnight and analyzed for the

expression of CD69 by flow cytometry. Mean values + s.e.m. are shown; number of independent experiments/mice: $n = 7$ ($Lck^{CA/KR}$) or 8 (other strains) for 3K, 5 for P5R and 6 ($Lck^{CA/KR}$) or 8 (other strains) for P2A. Differences in the $EC_{50}$ and/or maximum of the fitted non-linear regression curves were tested using an extra sum of squares $F$-test. $F$, $P$ and $EC_{50}$ values are shown. The significance of the differences at individual concentrations was calculated using a Mann–Whitney test and is displayed between $Lck^{WT/WT}$ and $Lck^{CA/CA}$ mice (blue stars) and $Lck^{WT/WT}$ and $Lck^{CA/KR}$ mice (red stars); *$P < 0.05$; **$P < 0.01$; no symbol, $P > 0.05$ (Supplementary Table 5). **f**, Indicated $Lck$-variant B3K508 $Rag2^{KO/KO}$ T cells were transferred into Ly5.1 mice followed by infection with $Lm$ expressing 3K or P2A. Expansion of B3K508 T cells was measured as percentage among total CD4⁺ T cells on day 5 after infection by flow cytometry. Results for individual mice and medians are shown; $n = 8$ $Lck^{WT/WT}$, $n = 11$ $Lck^{CA/CA}$ and $n = 10$ $Lck^{CA/KR}$ mice from three independent experiments for $Lm$-3K; $n = 12$ $Lck^{WT/WT}$, $n = 13$ $Lck^{CA/CA}$ and $n = 13$ $Lck^{CA/KR}$ mice from three independent experiments for $Lm$-P2A. Statistical significance was calculated using a Mann–Whitney test.

In line with the results from the ex vivo activation, we observed weaker responses of the $Lck^{CA/CA}$ B3K508 T cells to $Lm$ expressing 3K or low-affinity P2A in vivo (Fig. 6f). The $Lck^{CA/KR}$ T cells rescued responses to $Lm$-3K but not to $Lm$-P2A (Fig. 6f). These data indicated that CD4-coupled LCK has a kinase-independent role in T cell activation, but the response to low-affinity antigens requires the catalytic activity of CD4-bound LCK.

We observed that LCK stabilizes surface CD4 levels in a kinase-independent manner in peripheral CD4⁺ T cells (Fig. 7a,b). By contrast, surface CD8 levels were largely LCK independent (Extended Data Fig. 9c). The interaction with LCK stabilized surface CD4 also in thymocytes, but this effect was much weaker in DP thymocytes (Extended Data Fig. 9d) than in SP4 thymocytes (Extended Data Fig. 9e) or LN CD4⁺ T cells (Fig. 7a,b).

To address the role of LCK in stabilizing CD4 in human T cells, we measured surface CD4 levels in WT and LCK-deficient ($LCK^{KO}$) Jurkat cell lines[16]. $LCK^{KO}$ Jurkat cells expressed very low levels of surface CD4

(Extended Data Fig. 9f), which could be reverted by transducing these cells with $LCK^{WT}$ or $LCK^{KR}$ but not $LCK^{CA}$ (Fig. 7c). The downregulation of surface CD4 in the absence of its interaction with LCK was mediated by protein kinase C (PKC), as the PKC inhibitor elevated surface CD4 levels specifically in T cells from the $Lck^{CA/CA}$, but not $Lck^{WT/WT}$ or $Lck^{CA/KR}$, mice (Fig. 7d and Extended Data Fig. 9g). Using electron microscopy, we observed that the absence of the CD4–LCK interaction modulates CD4 distribution in the plasma membrane. LCK is present in clusters in $Lck^{CA/CA}$ CD4⁺ T cells but is relatively uniform in $Lck^{WT/WT}$ CD4⁺ T cells (Fig. 7e,f). Overall, the interaction of CD4 with LCK is required for its proper surface localization in CD4⁺ T cells and SP4 thymocytes and to a lesser extent in DP thymocytes.

## Discussion

We generated $Lck^{CA/CA}$, $Lck^{KR/KR}$ and $Lck^{CA/KR}$ knock-in mouse strains to resolve decades-long disputes about the role of CD8- and CD4-bound LCK in T cell biology. Our data indicate that the co-receptor–LCK

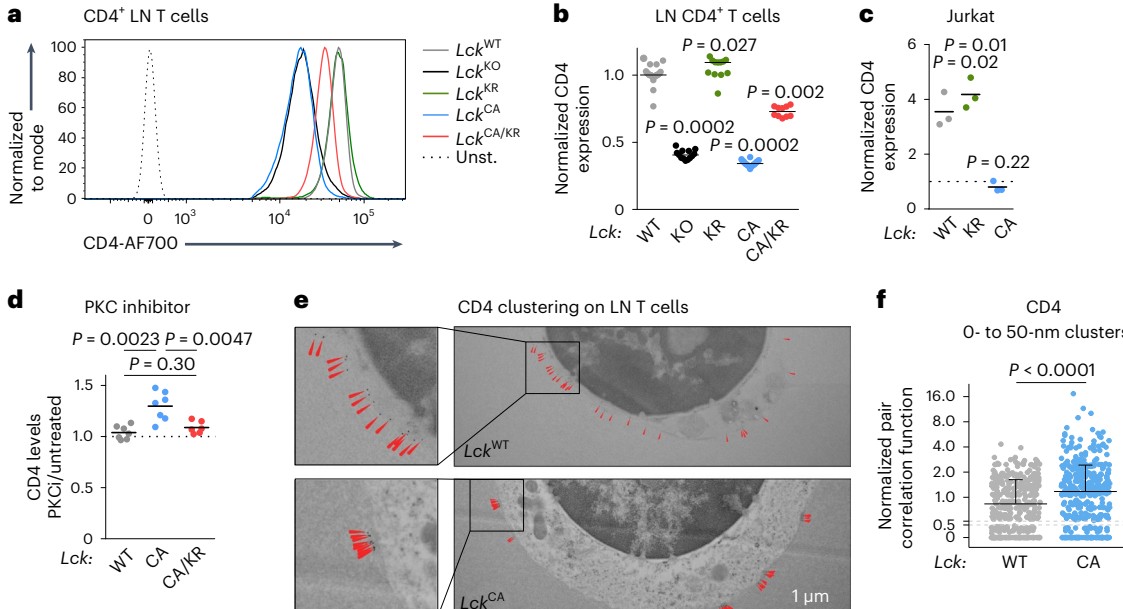

**Fig. 7 | LCK retains CD4 on the surface of T cells in a kinase-independent manner. a,b**, Relative CD4 surface levels on CD4+ T cells isolated from indicated *Lck* variants were determined by flow cytometry. **a**, Representative mice; Unst., unstained. **b**, Normalized CD4 expression (average gMFI of *Lck*WT/WT was set as 1 in each experiment). Individual mice and medians are shown; *n* = 25 *Lck*WT/WT, *n* = 13 *Lck*KO/KO, *n* = 12 *Lck*KR/KR, *n* = 18 *Lck*CA/CA and *n* = 10 *Lck*CA/KR mice in 11 independent experiments. The statistical significance was calculated using a Wilcoxon signed-rank test (versus the value of 1). **c**, Surface CD4 levels on *LCK*KO Jurkat cells transduced with indicated *LCK* variants were determined by flow cytometry (gMFI) and normalized to CD4 levels on untransduced *LCK*KO Jurkat cells (=1, indicated by a dashed line). Independent experiments and means are shown; *n* = 3 independent experiments. The statistical significance was calculated using a one-value *t*-test. **d**, LN T cells of indicated *Lck*-variant strains were incubated with a PKC inhibitor Ro-32-0432 overnight. CD4 expression was analyzed by flow cytometry (gMFI) and normalized to untreated cells from the same mouse (=1, indicated by a dashed line); *n* = 7 independent experiments/ mice for *Lck*WT/WT and *Lck*CA/CA and *n* = 6 for *Lck*KR/KR. Independent experiments and means are shown. The statistical significance was tested using a Mann–Whitney test; PKCi, PKC inhibitor. **e,f**, CD4 on LN CD4+ T cells from *Lck*WT/WT or *Lck*CA/CA B3K508 *Rag2*KO/KO mice was visualized with gold-labeled antibodies using transmission electron microscopy. **e**, Representative cells are shown. The arrows indicate CD4 molecules. **f**, Normalized pair correlation function values for CD4 clustering (clusters of 0–50 nM) were calculated for each cell. Individual values and medians + s.d. are shown; *n* = 358 *Lck*WT/WT and *n* = 423 *Lck*CA/CA cells from three independent experiments. The statistical significance was tested using a Mann–Whitney test.

---

interactions contribute to optimal T cell development and immune responses. However, the defects observed in mice with disrupted CD8–LCK and CD4–LCK interactions were much more subtle than in the LCK-deficient mice or mice expressing kinase-dead LCK, indicating that 'free' LCK can at least partially promote TCR signaling in vivo.

The genetic disruption of the co-receptor–LCK interaction did not impair the development of three tested monoclonal MHC class I- or MHC class II-restricted thymocytes. It is unclear why a previous study reported a severe block in the development of thymocytes expressing MHC class II-restricted AND TCR[12]. It is possible that it was caused by their experimental system (transgenic expression of LCK) or by a unique feature of the AND TCR. However, such a feature is not a low level of self-reactivity, as the normally developing F5 T cells are weakly self-reactive[27,29,30], whereas AND T cells are among the most self-reactive transgenic TCR clones[31].

Polyclonal *Lck*CA/CA mice developed relatively normal peripheral TCR repertoires and formed virus-specific and tumor-specific MHC class I- and MHC class II-restricted T cells. However, unlike the monoclonal *Lck*CA/CA mice, the polyclonal *Lck*CA/CA mice showed a block in the development of mature SP thymocytes. This difference can be caused by the non-physiological timing of TCR expression in the TCR transgenic mice, which alters the outcome of thymic checkpoints[32,33]. Another major factor is the presence of unconventional T cells in polyclonal mice and their comparative advantage over MHC class I/MHC class II-restricted T cells in the absence of the co-receptor–LCK interaction[12]. Indeed, we observed largely increased frequencies of NKT cell clones among the *Lck*CA/CA thymocytes. Although it is tempting to speculate that the unconventional T cells suppress the formation MHC class

I/MHC class II-restricted ones in a direct competition in the *Lck*CA/CA mice, our experiments with the BM chimeras show that the suboptimal formation of mature CD4+ and CD8+ T cells in the *Lck*CA/CA and *Lck*CA/KR mice is at least partially intrinsic. Overall, these results indicate that the co-receptor–LCK interaction is not essential for the formation of MHC class I/MHC class II-restricted T cells and their proper CD4/CD8 lineage commitment. However, in the absence of the co-receptor–LCK interaction, the maturation of MHC class I/MHC class II-restricted T cell clones is affected, whereas the formation of MHC class I/MHC class II-independent T cells is augmented.

We observed that the maturation and TCR signaling of early DP thymocytes is not blocked but is even slightly enhanced in the *Lck*CA/CA mice. The plausible explanation is that the *Lck*CA/CA mutation strips LCK from both CD4 and CD8 co-receptors, which leads to a large pool of 'free' LCK. For MHC class I-restricted TCRs, the loss of CD8-bound LCK probably decreases the responsiveness to self-antigens, but this is (over)compensated by the release of CD4-bound LCK into the 'free' LCK pool and vice versa for MHC class II-restricted thymocytes. It has been proposed that the co-receptor-bound LCK has a lower kinase activity than 'free' LCK[9], which would enhance the signaling in *Lck*CA/CA thymocytes. However a contradictory study proposed that the co-receptor binding enhances LCK activity[34], and we did not observe any differences between the response of *Lck*WT/WT and *Lck*CA/CA peripheral T cells to co-receptor-independent antibody-mediated TCR signaling. *Lck*CA/CA thymocytes have only a partial block in the formation of the most mature DP stage, which does not fully explain the relatively strong loss of SP4 and SP8 thymocytes. This suggests that a previously unappreciated signaling checkpoint might occur at the very transition

of postselection DP thymocytes into the SP stage and/or during the maturation of SP thymocytes. Our *Lck*-variant mice seem to be promising tools for further elucidating the role of LCK in fate decisions of conventional and unconventional T cells at different stages of maturation.

Our data show that the importance of CD8–LCK and CD4–LCK for T cell development and immune responses differs. CD8-bound LCK plays only a relatively minor role in the development of cytotoxic T cells. Although *Lck*[CA/CA] CD8[+] T cells are outcompeted by *Lck*[WT/WT] cells in the mixed BM chimeras, peripheral homeostatic proliferation[35] apparently compensates for the inefficient formation of CD8[+] T cells. *Lck*[CA/CA] mice also exhibit normal or near-normal antiviral and antitumor immunity and CD8[+] T cell responses to high-affinity antigens. The role of CD8–LCK in peripheral T cells seems to be largely limited to enhancing the signaling induced by low-affinity antigens. By contrast, CD8[+] T cells in *Lck*[CA/KR] mice show defective antigenic responses, suggesting that CD8-bound kinase-dead LCK[KR] inhibits activity of the 'free' LCK[CA]. Plausibly, LCK phosphorylates the TCR complex only when localized in a unique site. Co-receptor-bound LCK[KR] might preferentially occupy this position, preventing the 'free' LCK[CA] from initiating TCR signaling in the compound heterozygotes. This scenario is in line with the recent observation that co-receptor-bound LCK disables TCR triggering by pMHCs with reversed docking orientations[10,36]. However, following disruption of the CD8–LCK interaction, these reversely binding antigens induce TCR signals with comparable strength to canonical ligands[10]. Further investigation is required for understanding of the relationship between the localization of LCK and its ability to phosphorylate the TCR–CD3 complex. Overall, although we observed that the CD8-bound LCK enhances pMHCI tetramer binding, the functional assays did not reveal any biological importance of a previously proposed kinase-independent adaptor role of CD8-bound LCK[8].

CD4[+] T cells are more dependent on the co-receptor–LCK interaction than CD8[+] T cells. Defective maturation and activation of *Lck*[CA/CA] CD4[+] T cells is partially rescued with a pool of CD4-interacting kinase-dead LCK present in the *Lck*[CA/KR] mice, implying a kinase-independent role of CD4-bound LCK. Indeed, LCK promotes surface CD4 localization and its homogenous distribution in the plasma membrane, especially in mature T cells. The regulation of CD4 stability and trafficking by LCK was observed previously in transgenic non-lymphoid cell lines[37,38], but its relevance for T cell biology was not investigated before.

The responses to low-affinity antigens require the kinase activity of CD4-bound LCK, which is analogous to CD8[+] T cells. It is plausible that the CD4-bound kinase-dead LCK might have a dominant-negative role in TCR triggering under certain conditions. The ambiguous roles of the kinase-dead LCK might explain the rescue phenotype in the *Lck*[CA/KR] mice in some aspects of T cell biology (for example, formation of mature CD4[+] T cells and signaling of B3K508 CD4[+] T cells) but not in some other assays (signaling of B3K508 SP4 thymocytes and formation of NKT cells).

As we did not observe impaired development of monoclonal T cells in the *Lck*[CA/CA] and *Lck*[CA/KR] mice, we concluded that the defective responses of mature CD4[+] T cells and CD8[+] T cells observed in monoclonal and polyclonal *Lck*[CA/CA] and *Lck*[CA/KR] mice are probably largely intrinsic. However, we cannot formally exclude that some relevant differences are imprinted already during thymic development in the knock-in strains.

The differential role of co-receptor-bound LCK in the response to high- and low-affinity antigens could be potentially used for the development of novel strategies for treating autoimmune diseases. The specific inhibition of co-receptor-bound LCK should impair autoimmune T cell clones with relatively low antigen affinity[39,40] without inhibiting protective high-affinity T cell responses to infections. Moreover, disruption of the interaction between co-receptors and LCK might modulate the balance between cytotoxic and helper T cell responses, which could be beneficial in the tumor treatment.

## Online content

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

## Methods

### Mice

Mice had a C57BL/6J background (Charles River). For isolation of thymi for immunoblotting, 4- to 8-week-old mice were used. For fetal thymic organ cultures, embryos of embryonic age 15.5 were used. In other experiments, 6- to 12-week-old mice were used. Both males and females were used for experiments. We aimed at constant male and female representation among experimental groups in all experiments. Mice were bred in our specific pathogen-free facility (Institute of Molecular Genetics of the Czech Academy of Sciences; IMG) in accordance with the laws of the Czech Republic. Animal protocols (ID 11/2016, 115/2016, 72/2017 and AVCR 2378/2022 SOVII) were approved by the Resort Professional Commission for Approval of Projects of Experiments on Animals of the Czech Academy of Sciences, Czech Republic. The used congenic/transgenic strains were Ly5.1 (ref. [41]), $Cd3e^{KO/KO}$[42], OT-I $Rag2^{KO/KO}$[43,44], B3K508 $Rag2^{KO/KO}$[23], $Lck^{KO/KO}$[6], CD8.4 OT-I $Rag2^{KO/KO}$[11,27] and F5 $Rag1^{KO/KO}$[11,27]. All TCR transgenic mice used in this study had a $Rag2^{KO/KO}$ or $Rag1^{KO/KO}$ background. The colonies of all transgenic strains were established de novo in our animal facility by rederivation using embryo transfer or in vitro fertilization. Mice were fed with an irradiated standard rodent breeding diet and given reverse osmosis-filtered water ad libitum. Mice were kept in a facility with a 12-h light/12-h dark cycle and temperature and relative humidity maintained at $22 \pm 1\,°C$ and $55 \pm 5\%$, respectively.

$Lck^{C20.23A/C20.23A}$ and $Lck^{K273R/K273R}$ knock-in mice and $Lck^{KO/KO}$ mice were generated in the Czech Centre for Phenogenomics, IMG, using one-cell-stage embryos isolated from 3- to 5-week-old females mated with 9- to 35-week-old males and stimulated with 5 IU of pregnant mare serum gonadotropin (MSD Animal Health, Folligon PMSG) and 5 IU of human chorionic gonadotropin (Sigma, CG10)[45]. Both males and females were of the C57BL/6N strain (Charles River). The one-cell-stage embryos received a pronuclear microinjection of $Cas9$ mRNA (100 ng ml$^{-1}$) and guide RNA (50 ng ml$^{-1}$) together with single-stranded DNA templates (10 ng μl$^{-1}$) and were implanted into 7- to 15-week-old (and above 35 g of body weight) foster mothers of the CD1 strain (Charles River). The founders were back-crossed on the C57BL/6J background for at least five generations. Sequences of the oligonucleotides for the generation of mice and their genotyping are shown in the Supplementary Table 6.

### Cell counting and cell lines

Cells were counted using a Z2 Coulter Counter Analyzer (Beckman Coulter) or Cytek Aurora flow cytometer (Cytek).

Primary T cells were cultured in IMDM. Jurkat T cell lines[16,25] and T2-Kb cells (provided by E. Palmer, University Hospital Basel) were cultured in RPMI. HEK293 (provided by T. Brdicka, IMG) and MC-38 cells (provided by E. Palmer, University Hospital Basel) were cultured in DMEM. Medium was supplemented with 10% fetal bovine serum (FBS; Gibco), 100 U ml$^{-1}$ penicillin (BB Pharma), 100 mg ml$^{-1}$ streptomycin (Sigma-Aldrich) and 40 mg ml$^{-1}$ gentamicin (Sandoz). HEK293 cells are listed in the register of cell lines that are known to be misidentified through cross-contamination or other mechanisms (iclac.org/databases/cross-contaminations/)[46], because there was a case of their confusion with HeLa cells. We can exclude such a misidentification in our culture based on the morphology of the cells and their adhesion on the tissue culture plastic, which are clearly distinct between these two lines and which were checked in each experiment.

The parental human Jurkat leukemic line in this study was the $LCK^{KO}$ line[16] expressing the OT-I TCR[25]. This line was transduced with human $LCK$ variants ($LCK^{WT}$, $LCK^{C20.23A}$ and $LCK^{K273R}$) containing a C-terminal FLAG tag in pMSCV-IRES-LNGFR and eventually with human $CD8$ variants ($CD8α^{WT}CD8β$, $CD8α^{C215.217A}CD8β$) in pMSCV[25]. Human $CD8A$ and $CD8B$ genes were cloned de novo from human blood cDNA. The human LCK-encoding sequence was provided by T. Brdicka (IMG). The respective mutations in $LCK$ and $CD8A$ were introduced by PCR mutagenesis (Supplementary Table 6).

### Flow cytometry analysis

For the analysis of mouse thymocytes and T cells, the following antibodies were used: anti-CD4 (clone RM4-5, BioLegend 100536, 100545 and 130310, diluted 200×; RM4-4, BioLegend 116004, diluted 200×), anti-CD8α (clone 53-6.7, BioLegend 100738, 100753, 100708 and 100722, diluted 200×), anti-CD8β (clone YTS156.7.7, BioLegend 126615, diluted 200×), anti-CD24 (clone M1/69, BioLegend 101806, diluted 200×), anti-CD25 (clone PC61, BioLegend 102016, diluted 400× and 102006 and 102036 diluted 200×), anti-CD44 (clone IM7, BioLegend 103049, diluted 200×), anti-CD45.1 (clone A20, BioLegend 110723, diluted 200×), anti-CD45.2 (clone 104, BioLegend 109808, diluted 200×), anti-CD49d (clone R1-2, BioLegend 103618 and 103622, diluted 200×), anti-CD69 (clone H1.2F3, BioLegend 104508, diluted 200×), anti-TCRβ (clone H57-597, BioLegend 109218, 109206, 109243 and 109212 and BD Pharmingen 553171, diluted 200–400×), anti-PD-1 (clone 29F.1A12, BioLegend 135209, diluted 200×), anti-CXCR5 (clone L138D7, BioLegend 145504 and 145520, diluted 200×), anti-KLRG1 (clone 2F1/KLRG1, BioLegend 138421 and 138410, diluted 200×), anti-FOXP3 (clone FJK-16s, eBioscience 25-5773-82, diluted 100×) and anti-CD127 (clone A7R34, BioLegend 135013, diluted 200×). For analysis of Jurkat cell lines, anti-CD4 (clone MEM-241, Exbio A7-359-T100, diluted 50×), anti-CD8 (clone MEM-31, Exbio 1P-207-T025, diluted 50×), anti-CD69 (clone FN50, Exbio T7-552-T100, diluted 100×) and anti-CD271 (clone ME20.4, BioLegend 345108 and 345106, diluted 200×) were used. Antibodies were conjugated with various fluorophores by the manufacturers. LIVE/DEAD fixable near-IR dye (Thermo Fisher Scientific, L34975) was used for the viability staining.

The staining of live cells was performed in PBS/2% FBS/2 mM EDTA on ice usually for ~30 min. For the staining of FOXP3, the samples were fixed and permeabilized using the Foxp3/Transcription Factor Staining Buffer set (Thermo Fisher Scientific, 00-5523-00) according to the manufacturer's instructions.

For the analysis of thymocytes by phospho-specific flow cytometry, thymocytes from 4- to 9-week-old mice were immediately permeabilized in 2.5% formaldehyde (Sigma-Aldrich, F8775) in PBS for 10 min at ~21 °C and fixed in ~90% ice-cold methanol for 30 min on ice. The cells were stained with anti-phospho-ZAP70/SYK Y319 (polyclonal, Cell Signaling 2701, diluted 30×) and anti-pTCRζ-phycoerythrin (PE) (K25-407.69, BD Biosciences 558448, diluted 20×) overnight at 4 °C protected from light and then with antibodies for surface markers and with goat anti-rabbit-Alexa Fluor555 (polyclonal, Thermo Fisher Scientific A-32732, diluted 1,000×) in the case of pZAP70 staining at ~21 °C for 1 h protected from light. For the comparison of basal signaling in $Lck^{WT/WT}$, $Lck^{CA/CA}$ and $Lck^{CA/KR}$ mice, cells from individual mice were indexed by staining with anti-CD45.2-Alexa Fluor700 (clone 104, BioLegend 109822, diluted 200×) and/or anti-CD45.2-APC/Cy7 (clone 104, BD Pharmingen 560694 or BioLegend 109824, 200×) for 30 min on ice. Indexed cells were mixed in a 1:1:1 ($Lck^{WT/WT}$:$Lck^{CA/CA}$:$Lck^{CA/KR}$) ratio before staining with the phospho-specific antibodies.

For the unsupervised analysis of DP maturation, thymocytes were stained with TCRβ, CD4, CD8α, CD5, CD69, CD24, CD25 and LIVE/DEAD fixable near-IR dye, and $10^5$ live DP thymocytes (CD4$^+$CD8α$^+$) were downsampled from each individual mouse and concatenated together. Unbiased dimensional reduction and clustering by FlowSOM plugin was performed in FlowJo software. Graphs were mapped based on FlowSOM map using the EmbedSOM plugin in FlowJo software.

The samples were analyzed using a Cytek Aurora, BD LSRII or FACSSymphony flow cytometer. The data were analyzed using Flow Jo (version 10.6.2, BD Biosciences).

### Ex vivo activation assay

The human lymphoblast T2-Kb cell line expressing murine H2-K$^{b}$[47] was used for antigen presentation to OT-I T cells or Jurkat cell lines bearing OT-I TCR. Splenocytes from Ly5.1 mice were used for activation of B3K508 T cells. The antigen-presenting cells were pulsed with indicated

concentrations of indicated peptides and cocultured with isolated T cells at a 1:2 ratio overnight. CD69 expression was detected by flow cytometry. The results were fitted with a log (agonist) versus response (percentage of CD69+ T cells) function (least squares method) using PRISM (GraphPad Software).

For analysis of antibody-mediated activation, $2 \times 10^5$ T cells from LNs of OT-I $Rag2^{KO/KO}$ or B3K508 $Rag2^{KO/KO}$ mice were activated with titrated amounts of anti-CD3/CD28 beads (Gibco, 11453D) in 96-well plates (200 µl) for 16 h at 37 °C and 5% $CO_2$. Subsequently, the cells were analyzed by flow cytometry.

For the analysis of the antibody-mediated proliferation, $3 \times 10^6$ to $5 \times 10^6$ LN T cells from OT-I $Rag2^{KO/KO}$ or B3K508 $Rag2^{KO/KO}$ mice were loaded with 5 µM Cell Trace Violet dye (Thermo Fisher Scientific, C34557) in PBS for 10 min at 37 °C and 5% $CO_2$; $2 \times 10^5$ T cells per sample were used for activation with $2 \times 10^5$ anti-CD3/CD28 beads (Gibco, 11453D) and cultured in 48-well plates (600 µl) at 37 °C and 5% $CO_2$ for 72 h. After the incubation, proliferation was analyzed by flow cytometry.

## PKC inhibition assay
Live cells were incubated with 5 µM PKC inhibitor Ro-32-0432 (Sigma-Aldrich, 557525) overnight. CD4 expression was analyzed by flow cytometry (antibody clone RM4-5).

## Cloning of *Lck* variants and transfection into HEK293 cells
*Lck* WT, CA and KR open reading frames were amplified from cDNA obtained from the thymi of respective mice. FLAG tag was C-terminally fused to the *Lck* WT, CA and KR and cloned into pXJ41 vector (provided by T. Brdicka, IMG) using EcoRI/XhoI. ZAP70- and CD25-TCRζ-encoding genes (provided by T. Brdicka, IMG) were subcloned into pXJ41 (Supplementary Table 6).

HEK293 cells were grown to ~50% confluency and transfected with LCK variants and either ZAP70- or CD25-TCRζ-encoding pXJ41 plasmid. Thirty micrograms of DNA was mixed with 75 µg of polyethylenimine in 0.5 ml of DMEM/0.5% FBS for 10 min at room temperature. The mixture was then added onto cells in 3 ml of DMEM/0.5% FBS. The medium was replaced with DMEM/10% FBS/antibiotics after 3 h. Samples were collected 24 h after the transfection.

## Immunoprecipitation and immunoblotting
Total thymocytes were used for immunoprecipitation. Live cells ($2 \times 10^7$ to $3 \times 10^7$) were stained with biotinylated anti-CD8β (clone 53-5.8, BioLegend 140406, 2 µg) or anti-CD4 (clone GK1.5, BioLegend 100404, 2 µg). Cells were lysed in 1 ml of lysis buffer (1% lauryl-β-D-maltoside (Thermo Fisher Scientific), 30 mM Tris (pH 7.4), 120 mM NaCl, 2 mM KCl, 10% glycerol, complete protease inhibitors (Roche, 05056489001) and phosphoSTOP phosphatase inhibitors (Roche, 4906845001)), the lysate was cleared by centrifugation (20,000*g*), and supernatant was incubated with Streptavidin Mag Sepharose (GE Healthcare) for 2 h at 4 °C. Washed beads were lysed in Laemmli sample buffer. Samples were subjected to immunoblotting with murine anti-LCK (3A5, Santa Cruz sc-433, diluted 200×) and rabbit monoclonal anti-CD8α (D4W2Z, Cell Signaling, diluted 1,000×) or anti-CD4 (D7D2Z, Cell Signaling, diluted 1,000×).

For determination of endogenous LCK expression, $10^7$ thymocytes or LN T cells were lysed in 100 µl of lysis buffer, incubated for 30 min on ice, cleared by centrifugation (20,000*g*) and diluted in Laemli sample buffer. Samples were subjected to immunoblotting with murine anti-LCK (3A5, Santa Cruz sc-433, diluted 200×), rabbit anti-β-actin (4967, Cell Signaling, diluted 1,000×) and rabbit polyclonal anti-LAT serum[48].

For the analysis of basal TCRζ phosphorylation, thymi from 6- to 8-week-old female mice were lysed in lysis buffer (20 mM HEPES (pH 7.5), 150 mM NaCl, 2 mM EDTA (pH 8) and 0.5% Triton X-100) supplemented with protease inhibitor cocktail (Roche, 05056489001). Lysates were centrifuged at 15,000*g* for 15 min at 4 °C to remove cell debris.

Protein concentration was equalized using the Pierce BCA protein assay kit (Thermo Scientific). Proteins were denatured in 1× Laemmli sample buffer at 93 °C.

For the analysis of Jurkat cell activation by anti-TCR, Jurkat cells expressing LCK variants were starved for 30 min at 37 °C and stimulated for 2 min with anti-Jurkat TCR c305 supernatant (kindly provided by T. Brdicka) at 37 °C. Cells were then immediately lysed and denatured in 1× Laemmli sample buffer at 93 °C. The lysates were sonicated and used for immunoblotting.

The following primary antibodies were used for the analysis of basal and induced phosphorylation in primary T cells, Jurkat cells and HEK293 cells: anti-CD3-ζ (clone 6B10.2, Santa Cruz sc-1239, diluted 50×), LCK (clone 3A5, Santa Cruz sc-433, diluted 200–500×), anti-TCRζ (pY142; clone K25-407.69, BD Biosciences 558402, diluted 100×), anti-ZAP70 (clone 99F2, Cell Signaling 2705S, diluted 500×), phospho-Zap-70 (Try 319)/Syk (Tyr 352) (Cell Signaling 2701S, diluted 50×), anti-actin (Cell Signaling 4967, diluted 5,000×), anti-pTyr (clone 4G10, Sigma-Aldrich 05-321, diluted 5,000×) and anti-FLAG (clone M2, Sigma-Aldrich F1804-200UG, diluted 1,000×).

Both immunoprecipitation samples and lysates were visualized with secondary goat anti-rabbit or goat anti-mouse conjugated with horseradish peroxidase (Jackson ImmunoResearch Labs) on Azure c200 (Azure Biosystems) or Fusion Solo S (Vilber).

## BM chimeras
BM was isolated from 6- to 8-week-old $Lck^{WT/WT}$, $Lck^{CA/CA}$, or $Lck^{CA/KR}$ mice and mixed with supporting BM cells from Ly5.1 mice in a 1:1 ratio. Two million cells were transferred to lethally irradiated Ly5.1/Ly5.2 heterozygous donor mice. The mice received a dose of 6 Gy in an X-RAD 225XL Biological irradiator (Precision X-Ray). T cell development was analyzed 8 weeks after transplantation by flow cytometry.

## *Listeria* infection
LN T cells were isolated from B3K508 and OT-I mice. Cells were adoptively transferred to Ly5.1 congenic host mice. The following day, mice were injected with 5,000 colony-forming units of transgenic $Lm$ expressing OVA, T4, G4, 3K, P5R and P2A antigens[49–51]. Expansion of the responsive cells was analyzed by flow cytometry 5 d after infection.

## LCMV infection
LCMV (Armstrong) was obtained from D. Pinschewer (European Virus Archive Global). For batch production, hamster BHK-21 cells were infected at a multiplicity of infection of 0.01, and the virus-containing supernatant was collected 48 h after infection. Mice were infected by intraperitoneal injection of $2 \times 10^5$ plaque-forming units. Detection of LCMV in the spleen was performed by quantitative PCR with reverse transcription (RT–qPCR). Total RNA was isolated by TRIzol LS (Invitrogen, 10296010), and in-column DNase digestion was performed using an RNA Clean & Concentrator kit (Zymo Research), according to manufacturers' instructions. RNA was stored at −80 °C or transcribed immediately using RevertAid reverse transcriptase (Thermo Fisher Scientific, EP0442) with oligo(dT)18 primers (Thermo Fisher Scientific, SO131) according to the manufacturer's instructions. RT–qPCR was performed using LightCycler 480 SYBR green I master mix (Roche, 04887352001) and a LightCycler 480 II machine (Roche). All samples were measured in triplicates. The LCMV titers were quantified against a standard curve from cloned S-segment of LCMV in pBlueScript vector (Supplemetary Table 6). The quality of isolated RNA was tested by RT–qPCR analysis of an endogenous reference gene *Eef1a1* (Supplemetary Table 6).

## Tumor growth
The mouse MC-38 cell line derived from C57BL/6 colon adenocarcinoma[52] was transduced with ovalbumin protein-coding sequence via retroviral vector pMSCV-IRES-LNGFR. Five hundred thousand cells were

injected subcutaneously to the left side of the mouse. When $Cd3e^{-/-}$ mice were used as hosts, $2 \times 10^5$ OT-I cells were injected intravenously 5 d after tumor injection. Tumor size was measured by caliper, and tumor volume was estimated using the following formula: $V = (L \times S^2)/2$, where $L$ and $S$ are the longest and shortest diameters, respectively. The endpoint was the tumor volume exceeding 500 mm³ or the end of the experiment (day 22 after MC-38 injection for polyclonal mice and day 31 for $Cd3e^{-/-}$ mice). The approved animal protocols stated that mice with a tumor volume of 500 mm³ or larger must be killed, which was always followed.

## Cell isolation from tumors

Tumors were excised from mice, cut into small pieces and incubated with 100 µg ml⁻¹ Liberase (Roche, 5401020001) and 50 µg ml⁻¹ DNAse I (Roche, 101104159001) in wash buffer (1% bovine serum albumin and 1 mM EDTA in HBSS without Ca²⁺/Mg²⁺) at 37 °C and 350 r.p.m. shaking for 45 min. The mixture was resuspended with a 1,000-µl wide-bore pipette tip every 10 min. Undigested debris was removed by filtering through a 100-µm strainer. Cells were collected by centrifugation at 350g at 4 °C for 5 min. Pellets were resuspended in 10 ml of 40% Percoll (Cytiva, 17089101) in DMEM. Ten milliliters of 80% Percoll in DMEM was carefully laid on the bottom of the tube to create a gradient. Samples were centrifuged at 320g at ~21 °C for 23 min with minimal ascending/descending rates. Lymphocytes present at the interphase were collected, centrifuged at 400g at 4 °C for 5 min and processed for flow cytometry analysis.

## Fetal thymic organ cultures

Gelfoam gelatine sponge (Gelita-Spon Standard GS-002, Gelita Medical BV) was cut into pieces (1 cm²) and presoaked in medium for 10 min. Afterward, the sponges were placed in 2 ml of RPMI supplemented with 10% FBS, 50 µM β-mercaptoethanol, 2 mM L-glutamine, 100 U ml⁻¹ penicillin, 100 mg ml⁻¹ streptomycin, 50 mg ml⁻¹ gentamicin and 5 µg ml⁻¹ human β2-microglobulin (Sigma-Aldrich, 475823-M) in a six-well plate. A sterilized 0.45-µm cellulose membrane filter (Milipore, HAWP01s300) was soaked in medium for 5 s and placed on top of the sponge.

Fetal thymi were isolated from fetuses of embryonic age 15.5 and placed on the prepared filters and treated with corresponding concentrations of peptides. The medium was carefully exchanged every other day, and thymi were analyzed by flow cytometry on day 7.

## Flow cytometry-based TCR–ligand $k_{off}$ rate assay

Samples from $Lck$-variant OT-I mice were multiplexed by combination of staining with PE- and PerCP-Cy5.5-conjugated CD45.2 antibodies. The cells were then stained with Streptactin (IBA Lifesciences, 6-5010-001) multimerized with Alexa Fluor 488-conjugated pMHCI Kᵇ-OVA molecules[53,54]. Samples were measured at 5 °C and after 30 s of measurement, and the same volume of cold 2 mM D-biotin was added. Dissociation of the antigen was measured for an additional 10 min. For analysis, Streptactin and monomer fluorescence values of CD8⁺ OT-I T cells were exported from FlowJo (10.6.2) to Prism (GraphPad Software). The $t_{1/2}$ was calculated by fitting the data with a one-phase exponential decay curve.

## Tetramer binding

Tetramers were produced by refolding biotinylated monomers using streptavidin–PE conjugate in a molar ratio of 1:3. Streptavidin–PE (Thermo Fisher Scientific, S866) was added in three doses with a 20-min incubation on ice after each step. The following biotinylated monomers were used: H-2Kᵇ-OVA-PE (SIINFEKL), H-2Kᵇ-T4-PE (SIITFEKL) made in-house[55], H-2Dᵇ-NP396-PE (FQPQNGQFI), H-2Dᵇ-GP33-PE (KAVYNFATC) and I-Aᵇ-GP66-PE (DIYKGVYQFKSV) from the NIH Tetramer Core Facility.

For the detection of antigen-specific T cells, the tetramers (~100 nM) were added to the cocktail of antibodies for surface markers with the exception of the staining with I-Ab-GP66 tetramer, which was performed at ~21 °C in RPMI/2% FBS for 1 h.

For the quantitative binding analysis, peripheral T cells were isolated from OT-I $Rag2^{KO/KO}$ mice and incubated with a titrated dose of H-2Kᵇ-OVA-PE and H-2Kᵇ-T4-PE tetramers for 20 min on ice. The supernatant was replaced with PBS/2% FBS, and cells were immediately analyzed using a sample cooling system.

## Electron microscopy

LN cells were stained with anti-CD4 (clone RM4-5, BioLegend, diluted 50×) and washed and stained with 6-nm Colloidal Gold-AffiniPure goat anti-rat (polyclonal, Jackson ImmunoResearch 112-195-167, diluted 15×) on ice. Before processing, cell suspensions were diluted in 20% bovine serum albumin and rotated at 250g at 4 °C for 5 min. Subsequent cryofixation was done using a Leica EM ICE high-pressure freezer. Approximately 1 µl of each cell suspension variant was put into each of four type A 3-mm high-pressure freezer carrier sandwiches, which were rapidly frozen and dehydrated using a Leica EM AFS2 automatic freeze substitution unit under temperature slowly increasing from −90 °C to 0 °C over 4 d in 100% acetone enriched with 0.2% uranyl acetate, 0.2% glutaraldehyde, 0.01% osmium tetroxide and 5% water. Samples were then removed from AFS2 and infiltrated with 100% ethanol on ice and then with Quetol 651 resin diluted in 100% ethanol at 4 °C. Afterward, cells were embedded in Quetol NSA resin. After polymerization for 72 h at 60 °C, resin blocks were cut into 80-nm ultrathin sections using a Leica UC6 Ultra microtome with a diamond knife (Diatome), collected on copper slots with formvar membrane and air dried. After additional contrasting with 2% uranyl acetate in water, sections were examined with a JEOL JEM-1400Flash transmission electron microscope operated at 80 kV equipped with a Matataki Flash sCMOS camera (JEOL).

Electron microscopy images were analyzed using the open access application Pattern (pattern.img.cas.cz) developed by the Electron Microscopy Core Facility at IMG, Prague. Images were analyzed using one-dimensional analysis, where a region of interest was manually traced along the membrane. Size calibration was defined as 1.189 nm in 1 pixel. Clustering of gold particles was determined as pair correlation function value. Values were normalized to the predicted maximum standard deviation of the simulated pair correlation function value for analyzed density of individual cell staining.

## Ensemble fluorescence resonance energy transfer (FRET) measurements

Small unilamellar vesicles (97.5 mol-% 1-palmitoyl-2-oleoyl-glycero-3-phosphocholine, 2 mol-% 1,2-dioleoyl-$sn$-glycero-3-((N-(5-amino-1-carboxypentyl)iminodiacetic acid)succinyl) (nickel salt) and 0.5 mol-% PEG5000-DOPE (=18:1 PEG5000 PE)) were used to form a planar lipid bilayer on plasma-cleaned coverslips glued to eight-well Lab-Tek chambers (Nunc)[56]. Kᵇ-OVA-His12 conjugated with AF647 C2 Maleimide (1.5 ng per well; Thermo Fisher Scientific, A20347) and ICAM-1-His12 (0.1 ng per well) were added to the bilayers[57,58]. Control bilayers were exclusively functionalized with ICAM-1-His12. The Kᵇ-OVA molecular density of 40–80 molecules per µm² was determined by dividing the fluorescence signal per pixel by the single-molecule brightness recorded at the same settings and multiplying by 39.0625 (considering the effective pixel width of 160 nm, which results from the 100× magnification of a pixel width on the Andor iXon 897 EM-CCD Camera of 16 µm) to arrive at an area of 1 µm².

Fresh LN T cells were decorated with a site-specific AF555 C2 maleimide-conjugated (Thermo Fisher Scientific) scFV (J1), derived from the H57-597 monoclonal antibody that is reactive against the TCRβ chain (12.5 µg ml⁻¹)[58]. Stained T cells were then allowed to approach functionalized planar supported lipid bilayers to visualize and quantitate TCR–pMHCI binding using FRET donor recovery after acceptor photobleaching. For this, an image of the donor channel (AF555-H57-scFV) was recorded before and after acceptor photobleaching, and the pixel-averaged fluorescence signal $f_{pre}$ (before bleaching) and $f_{post}$ (after bleaching), respectively, was calculated for each

T cell synapse. The FRET efficiency was then given by $E = (f_{post} - f_{pre})/(f_{post} - \text{camera background})$. Bleaching time was 250 ms. Illumination time was 50 ms, with a 18-ms delay. The time lag between images of donor before and after bleaching was 550 ms.

In the microscopy system, excitation was achieved by coupling 532-nm (OBIS LS, Coherent) and 640-nm (iBeam smart, Toptica Photonics) laser lines into a Ti-E inverted microscope (Nikon) via a dichroic mirror ZT405/488/532/640rpc (Chroma) into a ×100 objective (SR Apo TIRF, Nikon) in an objective-based total internal reflection setting. Emission was split using an Optosplit II (Cairn Research) equipped with a 640-nm dichroic mirror (ZT640rdc, Chroma) and emission filters (ET575/50, ET655LP, Chroma) and simultaneously imaged on an Andor iXon Ultra 897 EM-CCD camera (Andor Technology). All devices were controlled by MetaMorph imaging software (MM 7.10.1.161, Molecular Devices).

### Analysis of TCR repertoires

Thymi and LNs were isolated from 4- to 5-week-old mice. Mature thymocytes were enriched using biotinylated anti-CD24A (clone M1/69, BioLegend 101804), whereas T cells from LNs were enriched using anti-B220 (clone RA3-6B2, BioLegend 103204) followed by depletion of labelled cells using Biotin binder magnetic beads (Thermo Fisher Scientific, 11047). The enriched populations from thymi and LNs were stained for flow cytometry and sorted as CD24$^-$, TCRβ$^+$ and CD4$^+$CD8α$^-$ or CD4$^-$CD8α$^+$ using a FACS ARIA IIu cell sorter (BD Biosciences) into TRIzol LS reagent (Thermo Fisher Scientific, 10296010) and stored at −80 °C. RNA was isolated using an RNA Clean & Concentrator-5 kit (Zymo Research R1014) according to the manufacturer's instructions. The maximum number of cells was sorted from each mouse. The following numbers of cells were sorted and used for the library preparation (in $10^5$ cells): $Lck^{WT/WT}$: SP8 thymocytes 7.7, 5 and 2.2, SP4 thymocytes 15.8 and 4.3, CD4$^+$ T cells 18 and 13.5 and CD8$^+$ T cells 13, 8.6 and 10.0; $Lck^{CA/CA}$: SP8 thymocytes 4.1 and 2.6, SP4 thymocytes 4.7 and 3.4, CD4$^+$ T cells, 8.0, 5.4 and 4.1 and CD8$^+$ T cells 4.4 and 5.2; $Lck^{CA/KR}$: SP8 thymocytes 0.7 and 0.4, SP4 thymocytes 1.9, 1.2 and 4.8, CD4$^+$ T cells 9.1 and 4.9 and CD8$^+$ T cells 5.6 and 2.7.

TCR libraries were prepared using the NEBNext Immune Sequencing kit (Mouse; NEB, E6330S) and additional NEBNext i7 primers NEBNext i707-i712 (NEB, E6347AA, E6348AA, E6349AA, E6352AA, E6353AA and E6354AA) according to the manufacturer's instructions. Libraries were sequenced on an Illumina MiSeq using the 600-cycle V3 MiSeq reagent kit (Illumina, MS-102-3003) and 2 × 300-base pair paired-end reads. Demultiplexed fastq files were deposited in the Sequence Read Archive (PRJNA872031) and processed with the pRESTO NEBNext Immune Sequencing Kit Workflow (v3.2.0) on Galaxy (https://usegalaxy.org/u/bradlanghorst/w/presto-nebnext-immune-seq-workflow-v320) with default parameters. Resulting fastq files were aligned using MiXCR (v3.0.13). Clonotype analysis was performed in R (version 4.1.2) using the packages immunarch (v0.6.9), pheatmap (v1.0.12), factoextra (v1.0.7) and tidyverse (v1.3.1). Unique molecular identifier (UMI)-corrected counts of TCRα (TRA) and TCRβ (TRB) CDR3 amino acid sequences were normalized, and the frequency of clones present in at least five CD4 or at least five CD8 samples (principal-component analysis) or the frequency of all clones (all other analyses) was compared between $Lck^{WT/WT}$, $Lck^{CA/CA}$ and $Lck^{CA/KR}$ mice. Invariant NKT TCRs containing TRAV11-TRAJ18 segments were removed before the analysis of the CDR3 repertoires.

### Statistical analysis

No statistical methods were used to predetermine sample sizes, but our sample sizes are similar to those reported in previous publications[6,27,59].

The mice and transgenic cells lines were allocated to experimental groups solely based on their genotype. If more experimental conditions were used in a single experiment (such as different *Listeria* strains), the allocation of the mice was random (that is, based on mouse ID in the database before the experimenter had any contact with them) in the way that sex- and age-matched animals with different genotypes were compared (preferably littermates). For the cell line transfection/transduction experiments, identical cell culture aliquots of the split parental culture were used; thus, no randomization was required.

Allocation of mice was based solely on their genotype. The experimenter processed the mice based on their ID number (that is, without the information about the genotype). ID was matched with the genotype only during data analysis at the end of the experiment. Because no subjective scoring method was used, the analysis of the mice was not explicitly blinded. Ex vivo experiments with primary cells and cell lines were not blinded. Because no subjective scoring method was used for the analysis, blinding was not necessary.

Some rare experiments/samples were excluded because of technical failures based on preestablished criteria. The only sample excluded after analysis was one sample for LCMV titer determination (Fig. 3a), which had a very late reference gene amplification in the RT–qPCR assay ($Lck^{WT/WT}$ day 6 after infection).

Statistical analyses were performed using Prism 5 (GraphPad Software). All the statistical tests were two tailed, if applicable. Adjustments for multiple comparisons were performed if indicated in the respective figure legends. For comparison of individual groups, a two-tailed Mann–Whitney test was used. For multirank comparison of survival curves for tumor growth, a log-rank (Mantel–Cox) test was used, and for comparison of individual survival curves, a Gehan–Breslow–Wilcoxon test was used. For multirank comparison of CD69 upregulation curves, an extra sum of squares $F$-test was used to test differences in the maximum and/or half-maximum effective concentration (EC$_{50}$) values of the non-linear regression fits. For the comparison of normalized data (to the $Lck^{WT/WT}$ strain), a two-tailed Wilcoxon signed-rank test or a two-tailed one-sample $t$-test was used. The latter was used for the exceptional cases where the low sample size did not allow us to use non-parametric tests. Data distribution was assumed to be normal in this case, but this was not formally tested because it was not possible given the sample size.

### Reporting summary

Further information on research design is available in the Nature Portfolio Reporting Summary linked to this article.

## Data availability

Mouse strains, cell lines and vectors generated within this work are available for non-commercial research purposes following a reasonable request. Materials transfer agreements will be required. Data of the TCR repertoire analysis are available in the Sequence Read Archive (PRJNA872031). Raw flow cytometry data and microscopy images are available following a reasonable request to the corresponding author. Source data are provided with this paper. All other data generated or analyzed during this study are included in this published article (and its Supplementary Information files).

## Code availability

Code for TCR repertoire analysis is available on GitHub (https://github.com/Lab-of-Adaptive-Immunity/lck-tcrseq).

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

## Acknowledgements

We thank L. Cupak for technical assistance and genotyping of mice. We thank the NIH Tetramer Core Facility for providing us with LCMV-specific tetramers. We thank I. Beck and P. Kasparek from the Czech Centre of Phenogenomics for generating mice with genetic variants of *Lck*. We thank the Electron Microscopy and the Flow Cytometry Core Facilities of the IMG for generation of electron microscopy data and cell sorting, respectively. We thank D. Pinschewer (European Virus Archive Global) for LCMV. We thank T. Brdicka for LAT and C305 antibody, DNA vectors and HEK293 cells. We thank A. Singer (NIH) for CD8.4 and F5 *Rag1*^KO/KO mice and E. Huseby (University of Massachusetts) for B3K508 mice. We thank E. Palmer (University Hospital Basel) for providing cells. We thank P. Draber, E. Palmer and W.-L. Lo for valuable comments on the manuscript draft. V.H., D.P., V.N., A.P., V.U., S.J. and E.S. were students of the Faculty of Science, Charles University, Prague, during their participation in this project.

This project was supported by the Swiss National Science Foundation grant Promys, IZ11Z0_166538 (O.S.), National Institute of Virology and Bacteriology (Programme EXCELES, ID421 project number LX22NPO5103), funded by the European Union, Next Generation EU (O.S.), European Union's Horizon 2020 Research and Innovation Programme under grant agreement number 802878, ERC Starting Grant FunDiT (O.S.), IMG of the Czech Academy of Sciences RVO 68378050 (O.S.), NIAID P01 AI091580 by the US National Institutes of Health (A.W.), Doctoral Program Cellular Communication in Health and Disease (CCHD), FWF of the Austrian Science Foundation (M.K. and J.B.H.), Boehringer Ingelheim Fonds, Germany (R.P.), and Technical University Munich SFB 1054 B09 (D.H.B.) and Technical University Munich SFB 1321 P17 (D.H.B.).

Parts of this project were performed at National Infrastructures hosted by the IMG funded by the following grants: Czech Ministry of Education, Youth and Sports grant LM2015042 (Czech Centre for Phenogenomics), Operational program of Czech Ministry of Education, Youth and Sport and the European Regional Development Fund RDI CZ.1.05/2.1.00/19.0395 (Czech Centre for Phenogenomics), Operational Program of Czech Ministry of Education, Youth and Sport and the European Regional Development Fund RDI BIOCEV CZ.1.05/1.1.00/02.0109 (Czech Centre for Phenogenomics), Operational Program of Czech Ministry of Education, Youth and Sport and the European Structural and Investment Funds CZ.02.1.01/0.0/0.0 /16_013/0001789 (Czech Centre for Phenogenomics), Czech Ministry of Education, Youth and Sports grant LM2018129 (Czech-BioImaging), Operational Program of Czech Ministry of Education, Youth and Sport and the European Regional Development Fund CZ.02.1.01/0. 0/0.0/18_046/0016045 (Czech-BioImaging), Operational Program of Czech Ministry of Education, Youth and Sport and the European Regional Development Fund CZ.02.1.01/0.0/0.0/16_013/0001775 (Czech-BioImaging).

## Author contributions

O.S. conceived, managed and administrated the project. V.H., A.D., D.P., V.N., A.P., V.U., D.G., M.K., M.C., S.J., E.S., O.T., T.A.K., K.K., R.P. and K.S. performed and analyzed experiments. O.S. analyzed experiments. V.H., A.D., V.N., M.K., R.P., D.H.B., A.W., K.S., J.B.H. and O.S. developed the methodology. D.H.B., A.W., J.B.H. and O.S. acquired funding and supervised the project. V.H. and O.S. wrote the manuscript. All authors reviewed and edited the manuscript.

## Competing interests

The authors declare no competing interests.

## Additional information

**Extended data** is available for this paper at https://doi.org/10.1038/s41590-022-01366-0.

**Correspondence and requests for materials** should be addressed to Ondrej Stepanek.

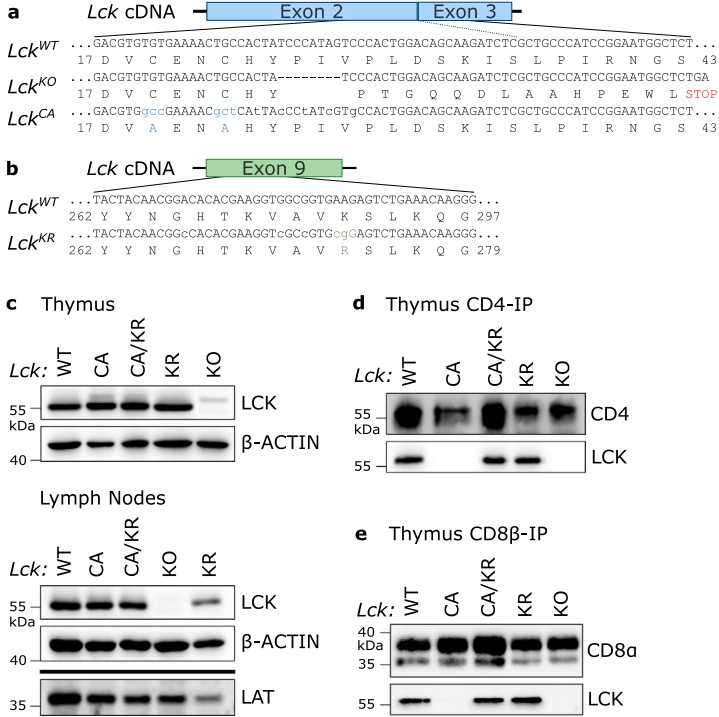

**Extended Data Fig. 1 | Generation of Lck variant mouse models.** (a) A scheme of targeting of the Lck locus for the generation of the *Lck*<sup>CA</sup> allele by CRISPR/Cas9. Resulting *Lck*<sup>CA</sup> (repaired via homologous recombination according to the DNA template) and *Lck*<sup>KO</sup> (non-homologous end joining causing a short deletion) alleles and the respective LCK protein sequences are shown. (b) A scheme of targeting of the *Lck* locus for the generation of the *Lck*<sup>KR</sup> allele by CRISPR/Cas9. Resulting *Lck*<sup>KR</sup> (repaired via homologous recombination according to the DNA template) allele and the respective LCK protein sequence are shown.

(c) LCK protein levels in thymi and LNs of *Lck* variant mice were detected by immunoblotting showing the comparable expression of *Lck* in these strains. β-actin staining was used as a loading control. LAT staining was used as a control for T-cell derived proteins in the lysates. A representative experiment out of 3 in total is shown. (d-e) Immunoprecipitation of CD4 (d) and CD8β (e) followed by immunoblotting and LCK staining in indicated *Lck* variant mice. A representative experiment out of 2 (d) or 3 (e) in total is shown.

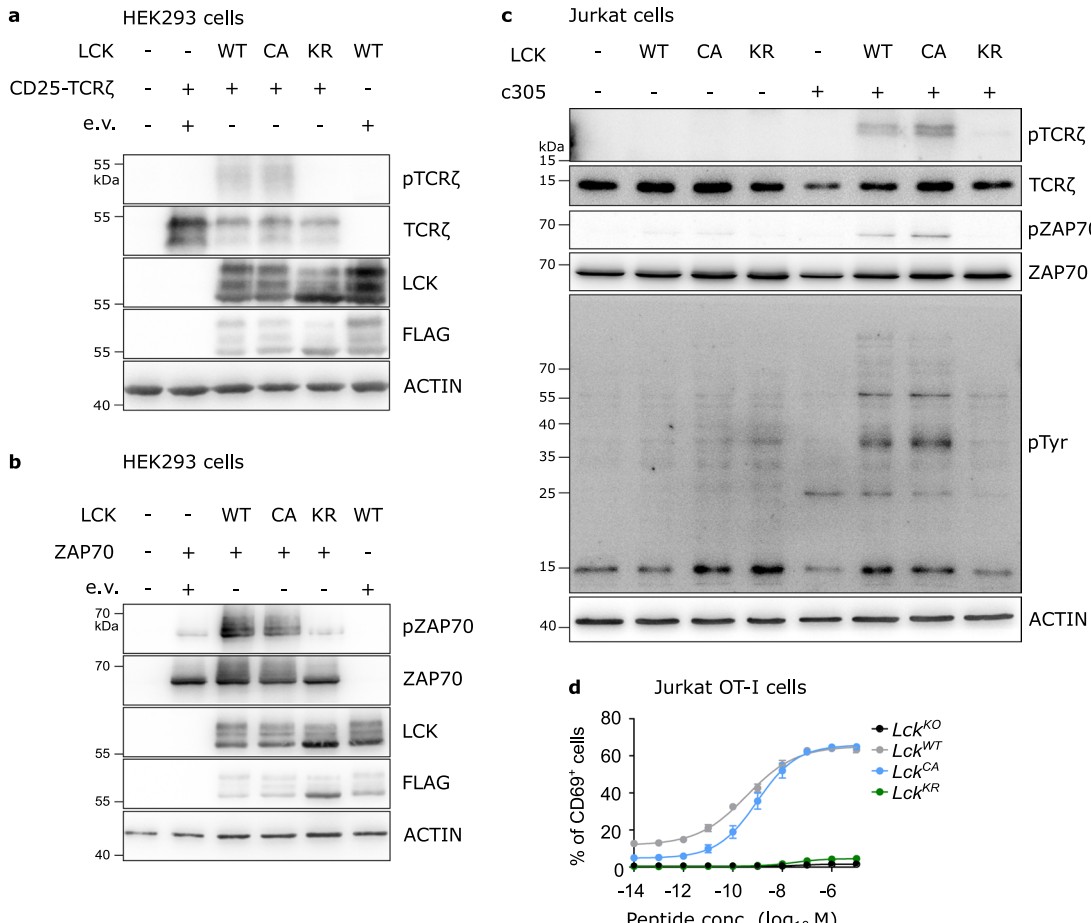

**Extended Data Fig. 2 | The enzymatic activities of LCK<sup>WT</sup> and LCK<sup>CA</sup> in cells are comparable.** (a) HEK293 cells were co-transfected with LCK-FLAG variants and CD25-TCRζ fusion protein or empty vector (e.v.) and harvested after 24 h. The phosphorylation of CD25-TCRζ was by detected immunoblotting. A representative experiment out of 2 in total. (b) HEK293 cells were co-transfected with LCK-FLAG variants and ZAP70 or empty vector (e.v.) and harvested after 24 h. The phosphorylation of ZAP70 was by detected immunoblotting. A representative experiment out of 2 in total. (c) *LCK<sup>KO</sup>* Jurkat cells reconstituted with indicated *LCK* variants were activated by anti-TCR antibody for 90 s. The phosphorylation and expression of indicated markers was detected by immunoblotting. A representative experiment out of 3 independent experiments. (d) *LCK<sup>KO</sup>* Jurkat cells expressing OT-I TCR were reconstituted with various *LCK* variants and activated with OVA-loaded T2-Kb cells overnight and analyzed for CD69 expression by flow cytometry. Mean +/- s.e.m. is shown. n = 3 independent experiments.

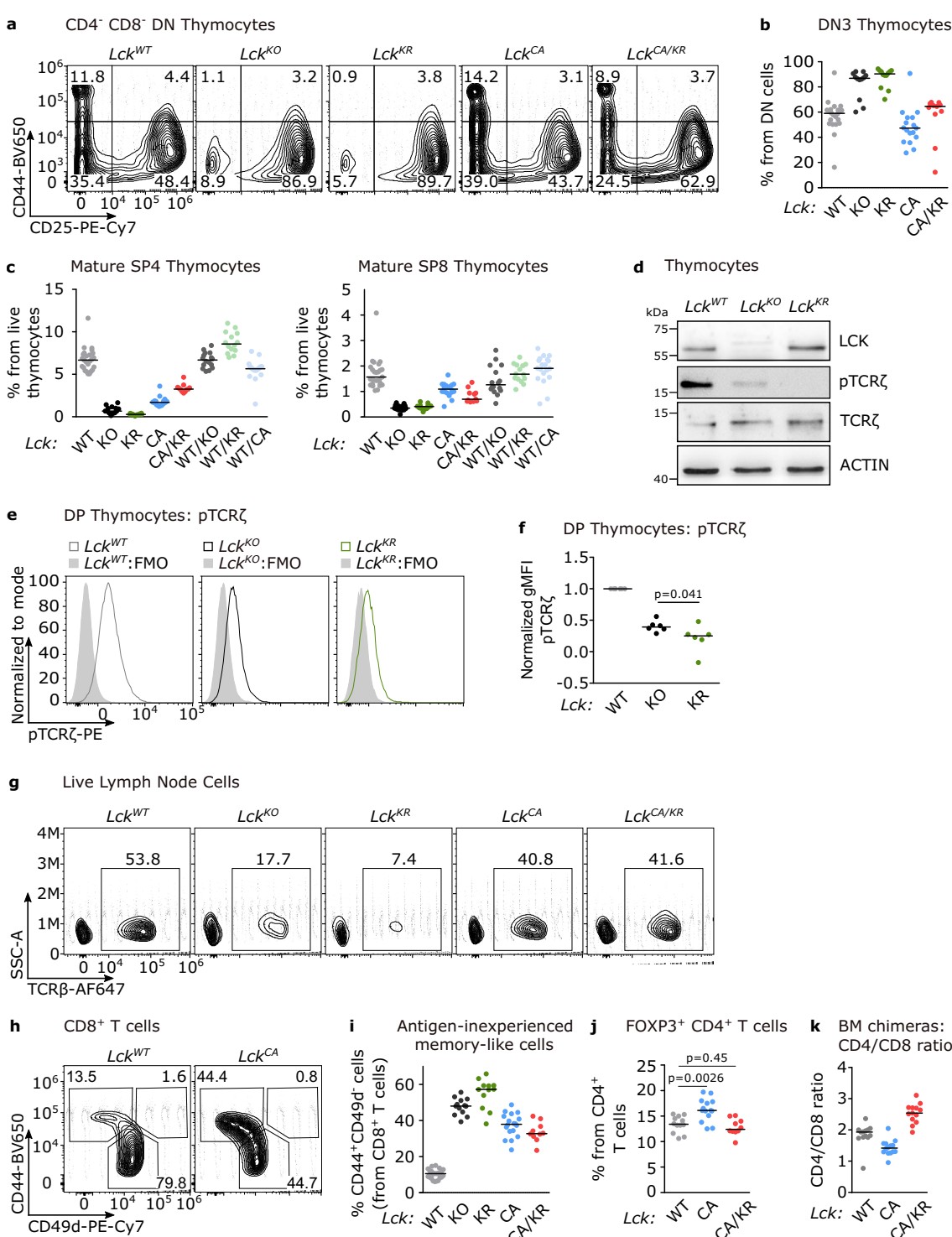

**Extended Data Fig. 3 | See next page for caption.**

**Extended Data Fig. 3 | Characterization of the T-cell compartment in Lck variant mice.** (a-f) Thymocytes and (g-j) LN cells from the indicated mice were analyzed by flow cytometry. (a-b) The percentage of cells at the DN3 stage (CD44⁻ CD25⁺) out of all DN (viable CD4⁻ CD8α⁻) thymocytes is shown. (a) Representative mice. (b) Individual values and medians are shown. $Lck^{WT/WT}$: n = 22 mice in 10 independent experiments, $Lck^{KO/KO}$: 12/6, $Lck^{KR/KR}$: 12/6, $Lck^{CA/CA}$: 17/7, $Lck^{CA/KR}$: 11/5. (c) Percentages of mature SP4 (TCRβ⁺ CD24⁻CD4⁺CD8α⁻) and mature SP8 (TCRβ⁺ CD24⁻CD4⁻ CD8α⁺) thymocytes. $Lck^{WT/WT}$: n = 25 mice in 11 independent experiments, $Lck^{KO/KO}$: 13/7 independent experiments, $Lck^{KR/KR}$: 12/6 independent experiments, $Lck^{CA/CA}$: 18/8, $Lck^{CA/KR}$: 11/5, $Lck^{WT/KO}$: 16/7, $Lck^{WT/KR}$: 14/8, $Lck^{WT/CA}$: 19/8. Individual values and medians are shown. (d) Phosphorylation of TCRζ was in thymocytes from the indicated mice was analyzed by immunoblotting. A representative experiment out of 3 in total. (e-f) Phosphorylation of TCRζ in thymocyte subpopulations determined by flow cytometry. (e) A representative experiment out of 6 in total. (f) Normalized TCRζ phosphorylation (net gMFI of $Lck^{WT}$ mice were set as 1). n = 6 independent experiments/mice. Means are shown. The statistical significance was determined using a Mann Whitney test. (g) Frequency of TCRβ⁺ LN cells was analyzed by flow cytometry. A representative experiment out of 11 for $Lck^{WT/WT}$, 7 for $Lck^{KO/KO}$, 6 for $Lck^{KR/KR}$, 8 for $Lck^{CA/CA}$ and 5 for $Lck^{CA/KR}$ mice in total. (h-i) A percentage of CD44⁺ CD49d⁻ T cells out of CD8⁺ T cells was determined by flow cytometry. (h) Representative samples of $Lck^{WT/WT}$ and $Lck^{CA/CA}$ mice are shown. (i) Individual values and medians are shown. $Lck^{WT/WT}$: 25 mice in 11 independent experiments, $Lck^{KO/KO}$: 13/7, $Lck^{KR/KR}$: 12/6, $Lck^{CA/CA}$:18/8, $Lck^{CA/KR}$: 10/5. (j) Frequency of regulatory T cells from overall CD4⁺ T cells in $Lck$ variant mice is shown. n = 12 $Lck^{WT/WT}$, 14 $Lck^{CA/CA}$, 10 $Lck^{CA/KR}$ mice in 3 independent experiments. Statistical significance was calculated using a Mann-Whitney test. (k) Ratio of CD4⁺/CD8⁺ T cell counts derived from bone marrows of $Lck^{WT/WT}$, $Lck^{CA/CA}$, and $Lck^{CA/KR}$ donors in the mixed bone marrow chimeras (Fig. 1g-h). n = 13 $Lck^{WT/WT}$, 14 $Lck^{CA/CA}$, 13 $Lck^{CA/KR}$ mice from 3 independent experiments. Individual values and medians are shown.

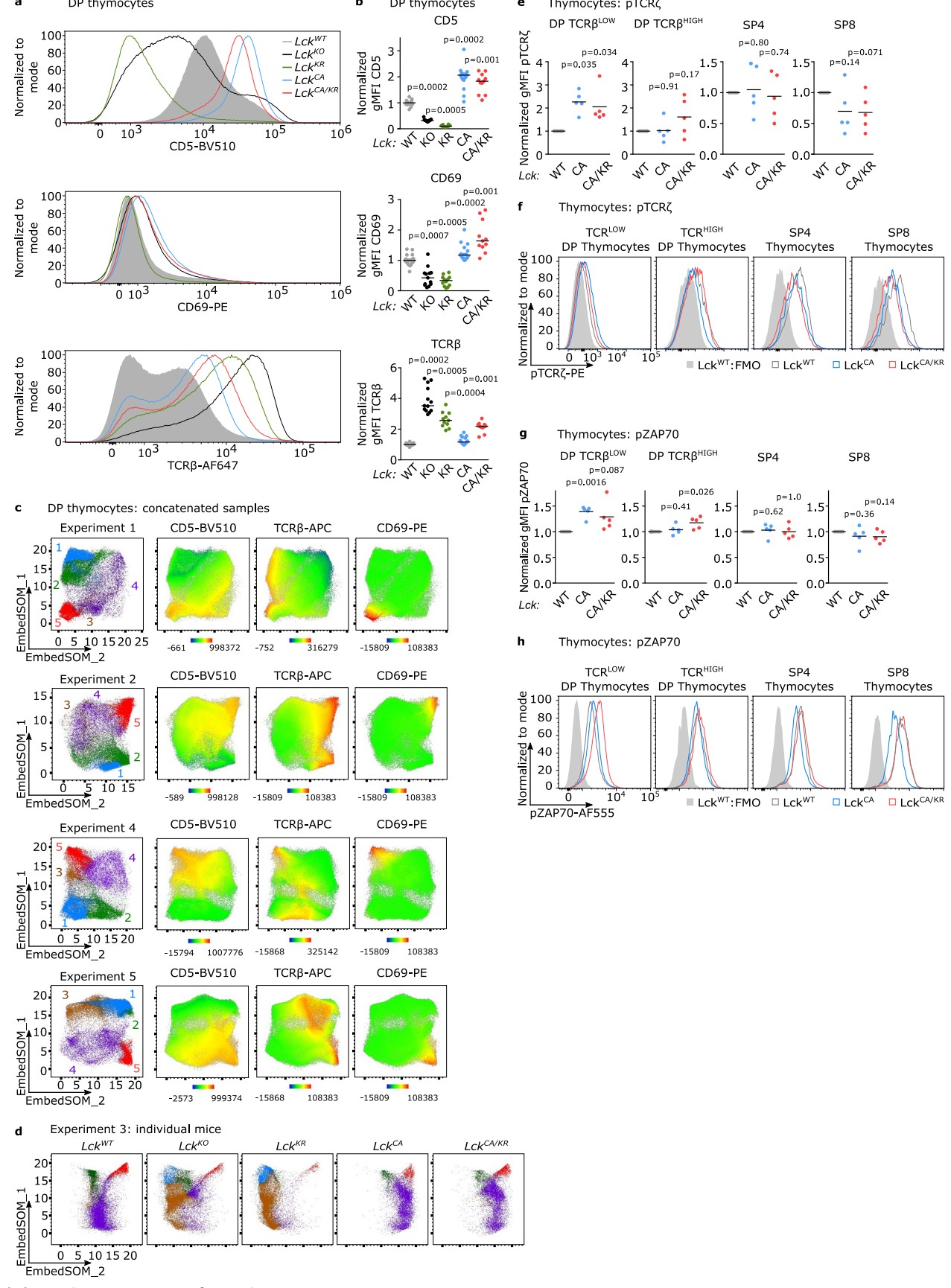

**Extended Data Fig. 4 | See next page for caption.**

**Extended Data Fig. 4 | Development of DP thymocytes in Lck variant mice.**
(a-b) Thymocytes from *Lck* variant mice were analyzed by flow cytometry.
(a) Representative histograms are shown. (b) Normalized expression of indicated
surface markers in the *Lck* variant mice (average gMFI in *Lck*$^{WT/WT}$ mice was
set as 1). Individual values and medians are shown. *Lck*$^{WT/WT}$: 25 mice in 11
independent experiments, *Lck*$^{KO/KO}$: 13/7, *Lck*$^{KR/KR}$: 12/6, *Lck*$^{CA/CA}$:18/8, *Lck*$^{CA/KR}$:
11/5. The statistical significance was calculated by Wilcoxon Signed Rank T
est (vs the value of 1). (c) Additional 4 independent experiments described in

Fig. 2a-b. (d) Representative mice from Experiment 3 described in Fig. 2a-b.
(e-h) Basal phosphorylation of TCRζ (e-f) and ZAP70 (g-h) in the indicated
thymocyte subpopulations (TCRβ$^{low}$ DP, TCRβ$^{high}$ DP, TCRβ$^{high}$ SP8, TCRβhigh
SP4) in the indicated *Lck* variant mouse strains was determined by flow
cytometry. (e,g) Overall data are shown. Data normalized to gMFI of *Lck*$^{WT/WT}$ mice
are displayed. n = 5 independent experiments/mice. The statistical significance
was tested using one sample t-test values. (f, h) Representative experiments out
of 5 in total.

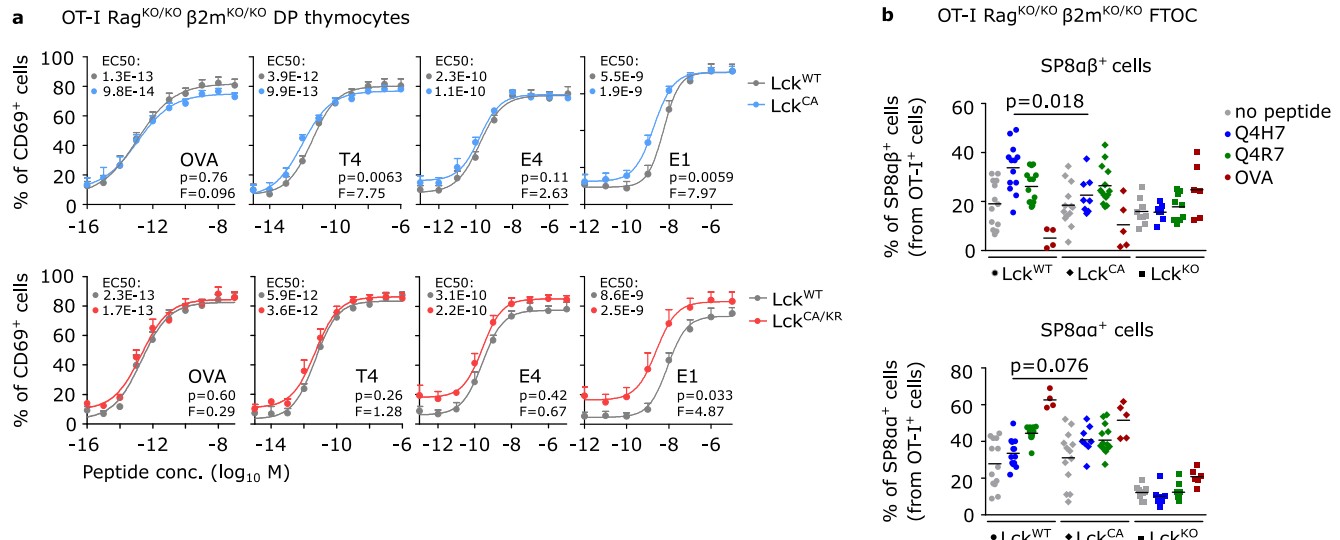

**Extended Data Fig. 5 | Signaling in DP thymocytes in Lck variant mice.** (a) Thymocytes of indicated *Lck* variants OT-I *Rag2^(KO/KO) β2m^(KO/KO)* mice were activated ex vivo with T2-Kb cells loaded with OVA peptide or APLs with decreasing affinity (OVA > T4 > E4 > E1) overnight and analyzed for the CD69 expression by flow cytometry. Mean values +/- s.e.m. are shown. n = 6 independent experiments/ mice for Lck^WT vs. Lck^CA (upper row), n = 3 independent experiments/mice for Lck^WT vs. Lck^CA/KR (bottom row). Differences in the EC50 of the fitted non-linear regression curves were tested using extra sum-of-squares F test. F, p, and fitted EC50 values are shown. (b) Fetal thymic organ cultures from indicated *Lck* variant OT-I *Rag2^(KO/KO) β2m^(KO/KO)* mice were stimulated with OVA peptide (20 μM) or its APLs decreasing affinity (Q4R7 > Q4H7; 2 μM) and analyzed by flow cytometry after 7 days. Individual mice and means are shown. n = 13 for *Lck^WT/WT* no peptide and Q4H7 in 5 independent experiments, n = 14 for *Lck^WT/WT* Q4R7 in 5 independent experiments, n = 4 for *Lck^WT/WT* OVA in 2 independent experiments, n = 13 for *Lck^CA/CA* no peptide in 5 independent experiments, n = 8 for *Lck^CA/CA* Q4H7 in 3 independent experiments, n = 14 for *Lck^CA/CA* Q4R7 in 5 independent experiments, n = 5 for *Lck^CACA* OVA in 2 independent experiments, n = 10 for *Lck^KO/KO* no peptide in 4 independent experiments, n = 8 for *Lck^KO/KO* Q4H7 in 3 independent experiments, n = 9 for *Lck^KO/KO* Q4R7 in 3 independent experiments, n = 6 for *Lck^KO/KO* OVA in 2 independent experiments. Statistical significance was calculated using a Mann-Whitney test.

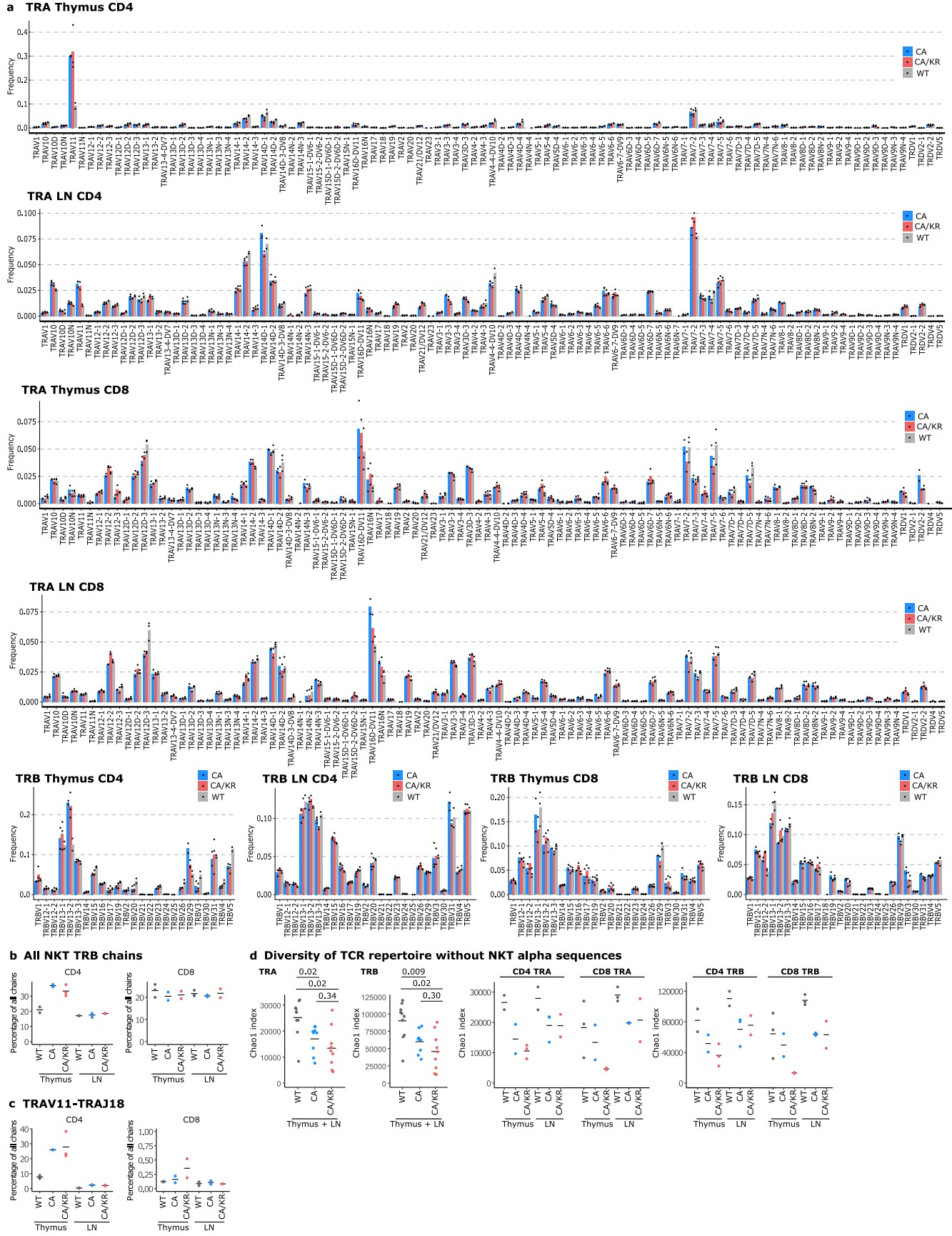

**Extended Data Fig. 6 | See next page for caption.**

**Extended Data Fig. 6 | Gene segment usage analysis of TCR repertoires in Lck variant mice.** (a-d) Analysis of TCR repertoires in CD4⁺ and CD8⁺ LN T cells, and mature SP4 and SP8 thymocytes from *Lck^{WT/WT}* (gray), *Lck^{CA/CA}* (blue) and *Lck^{CA/KR}* (red) mice. The information about the sample size are in Supplementary Table 6. Individual mice and means are shown. Related to Fig. 2e-g. (a) The usage of TRAV and TRBV gene segments in the indicated mice. Each bar represents the average frequency of the gene segment among all CDR3 sequences in a particular group of samples. (b) A total percentage of typical TCRβ (TRB) gene segments used by NKT cells (TRBV1,TRBV13-2, and TRBV9) in the indicated samples from the indicated mice. (c) Percentage of TCRs containing TRAV11-TRAJ18 gene segments typically used by NKT cells in the indicated cells from the indicated mice. (d) Diversity of the repertoire of the TCRα (TRA) and TCRβ (TRB) CDR3 amino acid sequences in the indicated cells from the indicated mice calculated using the Chao1 richness estimator. CD4 and CD8 T-cell CDR3 amino acid sequences were analyzed together (left) or separately (right). Invariant NKT TRAV11-TRAJ18 CDR3 amino acid sequences were removed before this analysis. The statistical significance was calculated using a Kruskal-Wallis test with multiple comparison adjustment using the Holm method.

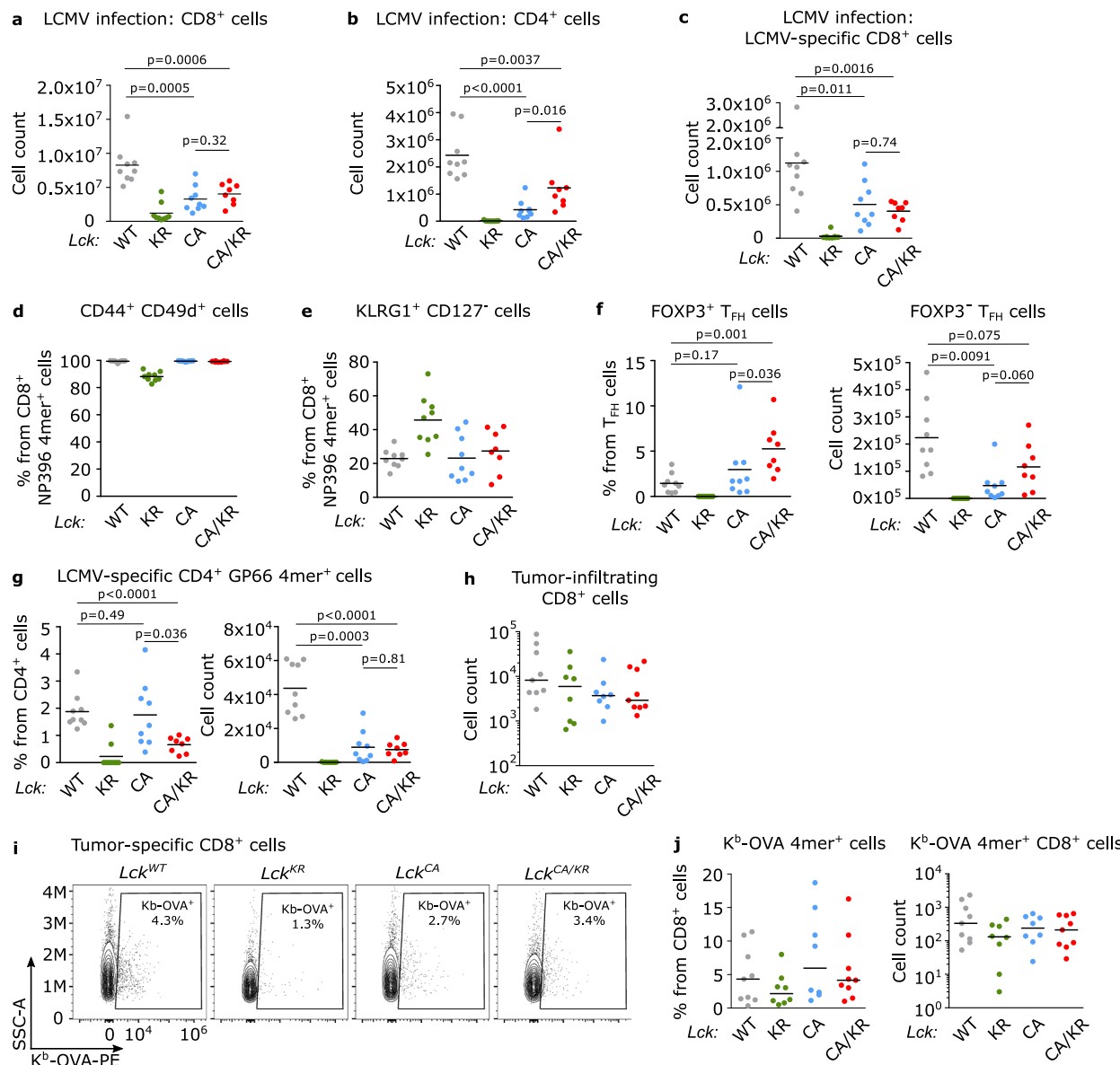

**Extended Data Fig. 7 | Analysis of T-cell compartment in anti-viral and anti-tumor responses.** (a-g) Indicated *Lck* variant mice were infected with LCMV and the splenocytes were analyzed on day 8 post-infection. n = 8 (*Lck^CA/KR*) or 9 (other strains) mice in 2 (*Lck^CA/CA*) or 3 (other strains) independent experiments. Individual mice and means are shown. The statistical significance was calculated using a Mann-Whitney test. Related to Fig. 3a-e. (a) Counts of CD8⁺ cells in the spleens. (b) Counts of CD4⁺ cells in the spleens. (c) Counts of GP33 4mer⁺ and NP396 4mer⁺ CD8⁺ cells in the spleens. (d) Frequencies of CD44⁺ CD49d⁺ cells from NP396 4mer⁺ CD8⁺ cells in the spleens. (e) Frequencies of KLRG1⁺ CD127⁻

cells from NP396 4mer⁺ CD8⁺ cells in the spleens. (f) Frequencies of T_FH cells expressing FOXP3 and counts of FOXP3⁻ T_FH cells in the spleens. (g) Frequencies and counts of GP66 4mer⁺ CD4⁺ cells in the spleens. (h-j) Tumor-infiltrating CD8⁺ T cells were isolated from MC-38-OVA tumors from *Lck* variant mice and analyzed by flow cytometry. n = 9 for *Lck^WT/WT*, *Lck^CA/KR*, n = 8 for *Lck^CA/CA*, *Lck^KR/KR* from 3 independent experiments. (j) Counts of tumor-infiltrating CD8⁺ T cells. Individual mice and medians are shown. (k) An example of gating strategy for the detection of K^b-OVA 4mer⁺ cells. (i) Frequencies and counts of K^b-OVA 4mer⁺ cells in tumors. Individual mice and medians are shown.

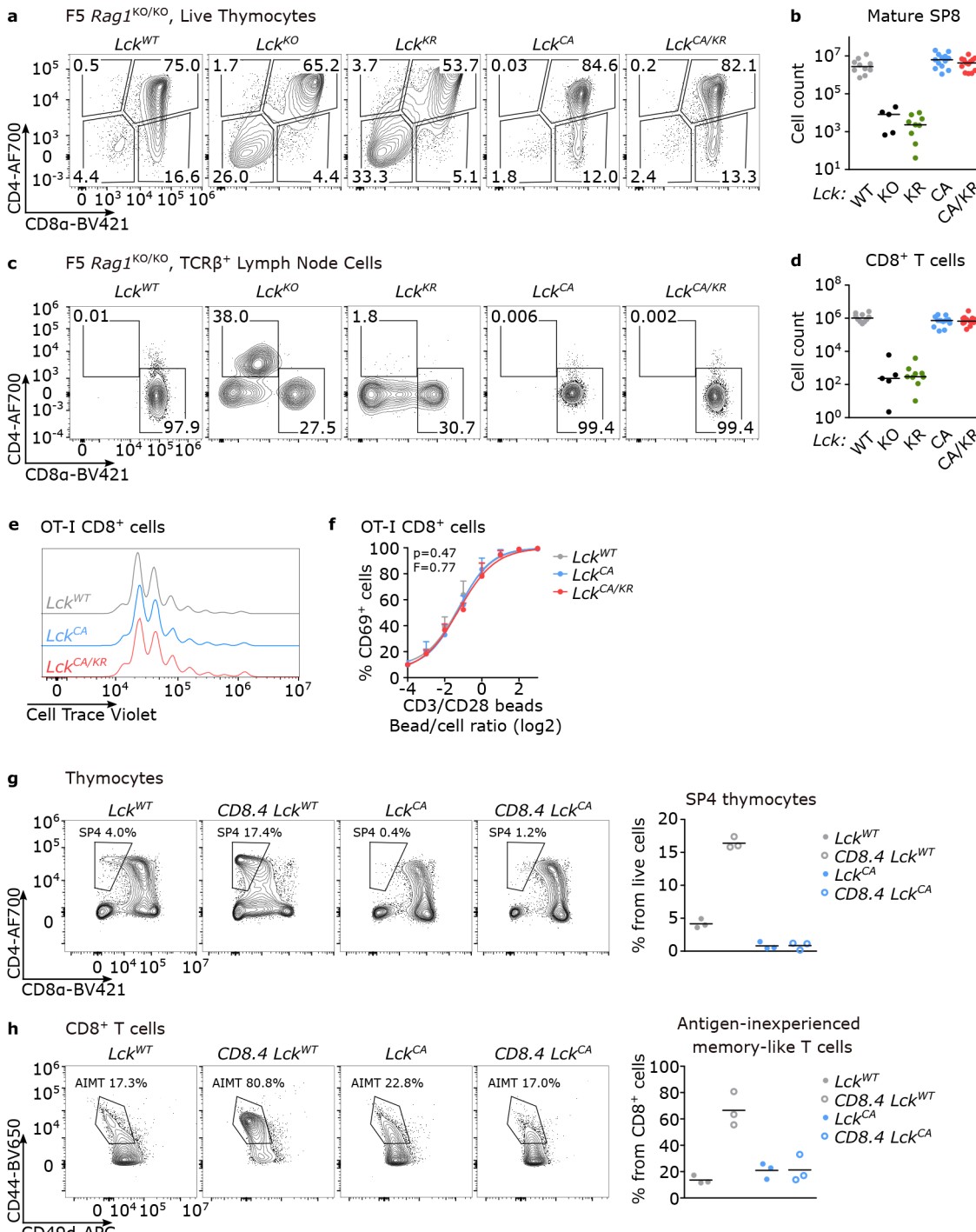

**Extended Data Fig. 8 | Characterization of CD8⁺ T cells in the Lck variant mice.** (a-d) Thymi (a-b) and LNs (c-d) of indicated *Lck* variant F5 *Rag1^{KO/KO}* mice were analyzed by flow cytometry. (a) Expression of CD4 and CD8α in representative mice (gated on viable cells). (b) Counts of mature SP8 (viable CD4⁻ CD8α⁺ CD24⁻ TCRβ⁺) thymocytes. Individual mice and medians are shown. n = 10 *Lck^{WT/WT}* in 7 independent experiments, 5 *Lck^{KO/KO}* in 3 independent experiments, 9 *Lck^{KR/KR}*, 13 *Lck^{CA/CA}* and *Lck^{CA/KR}* in 8 independent experiments. (c) Expression of CD4 and CD8α in representative mice (gated on viable cells). (d) Total numbers of CD8⁺ T cells in LNs. Individual mice and medians are shown. n = 10 *Lck^{WT/WT}* in 7 independent experiments, 5 *Lck^{KO/KO}* in 3 independent experiments, 9 *Lck^{KR/KR}*, 12 *Lck^{CA/CA}* and 13 *Lck^{CA/KR}* in 8 independent experiments. (e-f) LN cells from indicated *Lck* variant OT-I mice were loaded with Cell Trace violet (CTV) and stimulated with anti-CD3/CD28 beads. (e) The proliferation was evaluated

based on the CTV dilution at 72 hours after activation by flow cytometry. A representative experiment/mice out of 4 in total. (f) Upregulation of CD69 was analyzed by flow cytometry at 16 hours after activation. Mean + s.e.m. n = 3 independent experiments/mice. Differences in the EC50 and/or maximum of the fitted non-linear regression curves were tested using extra sum-of-squares F test. F and p values are shown. (g) Thymocytes from indicated *Lck* variant CD8 WT or CD8.4 OT-I mice were analyzed by flow cytometry. Representative mice and the frequencies of SP4 T cells from 3 independent experiments/mice are shown. (h) LN cells from indicated Lck variant CD8 WT or CD8.4 OT-I mice were analyzed by flow cytometry. Representative mice and the frequencies of CD44⁺ CD49d⁻ antigen-inexperienced memory-like cells (individual values and means) from 3 independent experiments/mice are shown.

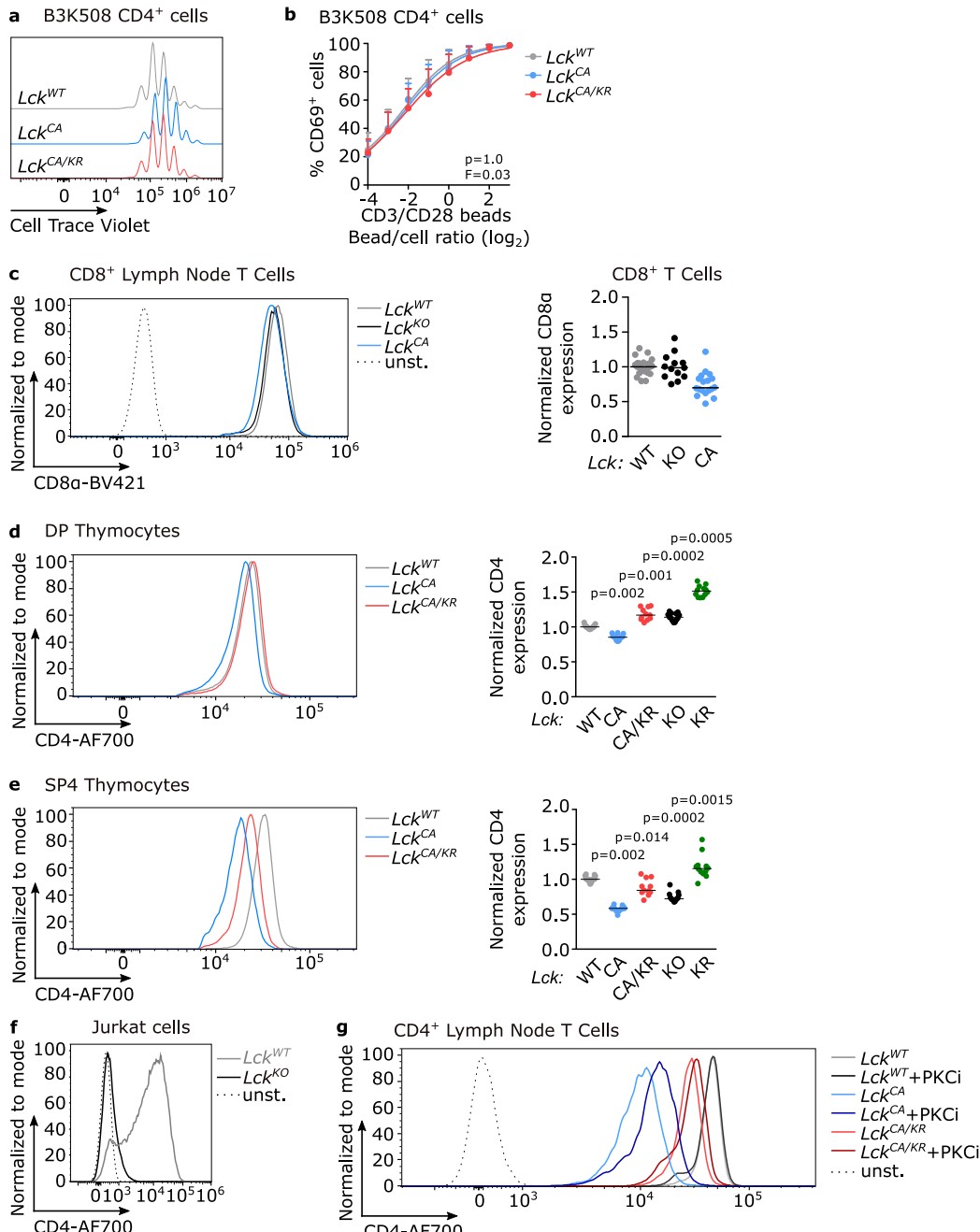

**Extended Data Fig. 9 | Characterization of CD4 + T cells in the Lck variant mice.** (a-b) LN cells from indicated *Lck* variant B3K508 mice were loaded with Cell Trace violet (CTV) and stimulated with anti-CD3/CD28 beads. (a) The proliferation was evaluated based on CTV dilution by flow cytometry at 72 hours after activation. A representative experiment/mice out of 4 in total. (b) Upregulation of CD69 was analyzed by flow cytometry at 16 hours after activation. Mean + SEM. n = 3 independent experiments/mice. Differences in the EC50 and/or maximum of the fitted non-linear regression curves were tested using extra sum-of-squares F test. F and p values are shown. (c) Surface levels of CD8α on CD8⁺ LN T cells in indicated *Lck* variant mice. A representative experiment and normalized CD8α surface levels (average gMFI of *Lck^WT/WT* was set as 1 in each experiment). n = 25 *Lck^WT/WT* in 11 independent experiments, 13 *Lck^KO/KO* in 7 independent experiments and 18 *Lck^CA/CA* in 8 independent experiments.

Individual values and means. (d-e) CD4 surface levels on DP thymocytes (d) and mature SP4 thymocytes (e) of *Lck* variant mice were analyzed by flow cytometry. Representative histograms and the quantification of normalized CD4 surface levels (average gMFI of *Lck^WT/WT* was set as 1 in each experiment). n = 25 *Lck^WT/WT* in 11 independent experiments, 13 *Lck^KO/KO* in 7 independent experiments, 12 *Lck^KR/KR* in 6 independent experiments, 18 *Lck^CA/CA* in 8 independent experiments, 11 *Lck^CA/KR* mice in 5 independent experiments. Individual values and means. The statistical significance was calculated using a Wilcoxon Signed Rank Test (vs the value of 1). (f) CD4 surface levels on WT Jurkat and *LCK^KO* Jurkat cells were analyzed by flow cytometry. A representative experiment out of 3 in total. (g) A representative experiment showing CD4 surface levels on CD4⁺ T cells upon the overnight treatment with PKC inhibitor as in Fig. 7d.

# Reporting Summary

## Statistics

For all statistical analyses, confirm that the following items are present in the figure legend, table legend, main text, or Methods section.

| n/a | Confirmed | |
|---|---|---|
| ☐ | ☒ | The exact sample size (*n*) for each experimental group/condition, given as a discrete number and unit of measurement |
| ☐ | ☒ | A statement on whether measurements were taken from distinct samples or whether the same sample was measured repeatedly |
| ☐ | ☒ | The statistical test(s) used AND whether they are one- or two-sided *Only common tests should be described solely by name; describe more complex techniques in the Methods section.* |
| ☒ | ☐ | A description of all covariates tested |
| ☐ | ☒ | A description of any assumptions or corrections, such as tests of normality and adjustment for multiple comparisons |
| ☐ | ☒ | A full description of the statistical parameters including central tendency (e.g. means) or other basic estimates (e.g. regression coefficient) AND variation (e.g. standard deviation) or associated estimates of uncertainty (e.g. confidence intervals) |
| ☐ | ☒ | For null hypothesis testing, the test statistic (e.g. *F*, *t*, *r*) with confidence intervals, effect sizes, degrees of freedom and *P* value noted *Give P values as exact values whenever suitable.* |
| ☒ | ☐ | For Bayesian analysis, information on the choice of priors and Markov chain Monte Carlo settings |
| ☒ | ☐ | For hierarchical and complex designs, identification of the appropriate level for tests and full reporting of outcomes |
| ☒ | ☐ | Estimates of effect sizes (e.g. Cohen's *d*, Pearson's *r*), indicating how they were calculated |

*Our web collection on statistics for biologists contains articles on many of the points above.*

## Software and code

Policy information about availability of computer code

| Data collection | MetaMorph imaging software (MM 7.10.1.161, Molecular Devices) was used for the collection of microscopy images for FRET measurements. |
|---|---|
| Data analysis | GraphPad Prism for Windows Version 5.04, FlowJo 10.6.2 with EmbedSOM plugin, ImageJ 1.53c, Pattern (pattern.img.cas.cz), R (version 4.2.1) including the packages immunarch (v0.6.9), pheatmap (v1.0.12) factoextra (v1.0.7) and tidyverse (v1.3.1), MiXCR (v3.0.13) |

For manuscripts utilizing custom algorithms or software that are central to the research but not yet described in published literature, software must be made available to editors and reviewers. We strongly encourage code deposition in a community repository (e.g. GitHub). See the Nature Portfolio guidelines for submitting code & software for further information.

## Data

Policy information about availability of data

All manuscripts must include a data availability statement. This statement should provide the following information, where applicable:
- Accession codes, unique identifiers, or web links for publicly available datasets
- A description of any restrictions on data availability
- For clinical datasets or third party data, please ensure that the statement adheres to our policy

Data of the TCR repertoire analysis are available in the Sequence Read Archive (PRJNA872031). Raw flow cytometry data and microscopy images are available upon a reasonable request to the corresponding author. All other data generated or analyzed during this study are included in this published article (and its supplementary information files).

# Field-specific reporting

Please select the one below that is the best fit for your research. If you are not sure, read the appropriate sections before making your selection.

☒ Life sciences ☐ Behavioural & social sciences ☐ Ecological, evolutionary & environmental sciences

For a reference copy of the document with all sections, see nature.com/documents/nr-reporting-summary-flat.pdf

# Life sciences study design

All studies must disclose on these points even when the disclosure is negative.

| | |
|---|---|
| Sample size | No sample size calculation was performed. In animal experiments, we usually aimed at the minimal number of 10 mice per group in at least 3 independent experiments, which was based on our prior experience. The actual number of mice and experiments (usually higher than the minimal numbers) was given by the availability of mice with particular genotypes, which could be not be predicted in advance as we largely used littermates for experiments. For ex vivo experiments with primary cells and cell lines, at least 2-3 independent experiments were performed, which is the minimal number to assess the reproducibility. |
| Data exclusions | 1. Very rare flow cytometry samples with apparent technical issues, such as bubbles in the flow cell or clogged machine, were identified based on abnormal FSC vs. SSC profile and excluded before the analysis. This is a common practice and thus, this is a pre-established criterium.<br>2. At one point, new stocks of Listeria were wrong. The reason was unknown, but the probable cause was little experience of the person, who prepared them for the first time. This was realized upon the CFU number counting. We calculate the CFU concentration from OD values and verify it by plating the Listeria suspension on plates and manual colony counting on the next day. In this case, there was almost ten times fewer colonies than expected and the experiment was excluded. The whole respective experiments were excluded. This was a pre-established criterium.<br>3. In one experiment focused on LCMV titers, one sample was removed because of bad RNA quality. The reference host gene was amplified at much later cycles than usual (i.e., more than 4 cycles). This was not an explicitly pre-established criterium, but we decided to exclude this sample that was apparently wrong from the technical point of view.<br>4. We originally aimed to have 3 samples (3 independent experiments) per group for the TCR sequencing experiment. Unfortunately, several of the libraries failed the QC on Agilent and were not included in the sequencing run, resulting in only 2 samples in some groups. |
| Replication | All data were replicated in at least 2, but usually 3 or more, independent experiments (this is indicated in the Figure Legend). The aggregate data are shown in the paper. There were no unsuccessful attempts to replicate the experimental results. |
| Randomization | The mice and transgenic cells lines were allocated to experimental groups solely based on their genotype. If more experimental conditions were used in a single experiments (such as different Listeria strains), the allocation of the mice were random (i.e., based on mouse ID in the database, before the experimenter had any contact with them) in the way that or sex- and age-matched animals with different genotypes were compared (preferably littermates). For the cell line transfection/transduction experiments, identical cell culture aliquots of the split parental culture were used, thus no randomization was required. |
| Blinding | The allocation of the mice was based solely on their genotype. The experimenter processed the mice based on their ID number (i.e., without the information about the genotype). ID was matched with the genotype only during the data analysis at the end of the experiment. Because no subjective scoring method was used, the analysis of the mice was not explicitly blinded. Ex vivo experiments with primary cells and cell lines were not blinded. Because no subjective scoring method was used for the analysis, the blinding was not necessary. |

# Reporting for specific materials, systems and methods

We require information from authors about some types of materials, experimental systems and methods used in many studies. Here, indicate whether each material, system or method listed is relevant to your study. If you are not sure if a list item applies to your research, read the appropriate section before selecting a response.

## Materials & experimental systems

| n/a | Involved in the study |
|---|---|
| ☐ | ☒ Antibodies |
| ☐ | ☒ Eukaryotic cell lines |
| ☒ | ☐ Palaeontology and archaeology |
| ☐ | ☒ Animals and other organisms |
| ☒ | ☐ Human research participants |
| ☒ | ☐ Clinical data |
| ☒ | ☐ Dual use research of concern |

## Methods

| n/a | Involved in the study |
|---|---|
| ☒ | ☐ ChIP-seq |
| ☐ | ☒ Flow cytometry |
| ☒ | ☐ MRI-based neuroimaging |

## Antibodies

| | |
|---|---|
| Antibodies used | For the analysis of murine thymocytes and T cells, the following antibodies were used: anti-CD4 (clone RM4-5, BioLegend #100536, #100545, #130310, diluted 200× and RM4-4, BioLegend #116004, diluted 200x), anti-CD8α (clone 53-6.7, BioLegend #100738, |

#100753, #100708, #100722, diluted 200x), anti-CD8β (clone YTS156.7.7, BioLegend #126615, diluted 200×), anti-CD24 (clone M1/69, BioLegend #101806, diluted 200×), anti-CD25 (clone PC61, BioLegend #102016 diluted 400x, #102006 and #102036, diluted 200x), anti-CD44 (clone IM7, BioLegend #103049, diluted 200×), anti-CD45.1 (clone A20, BioLegend #110723, diluted 200×), anti-CD45.2 (clone 104, BioLegend #109808, diluted 200×), anti-CD49d (clone R1-2, BioLegend #103618, #103622, diluted 200×), anti-CD69 (clone H1.2F3, BioLegend #104508, diluted 200×), anti-TCRβ (clone H57-597, BioLegend #109218, #109206, #109243, #109212, BD Pharmingen #553171, diluted 200-400x), anti-PD-1 (clone 29F.1A12, BioLegend #135209, diluted 200x), anti-CXCR5 (clone L138D7, BioLegend #145504, #145520, diluted 200x), anti-KLRG1 (clone 2F1/KLRG1, BioLegend #138421, #138410, diluted 200x), anti-FOXP3 (clone FJK-16s, eBioscience #25-5773-82, diluted 100x), anti-CD127 (clone A7R34, BioLegend #135013, diluted 200x). For analysis of Jurkat cell lines anti-CD4 (clone MEM-241, Exbio #A7-359-T100, diluted 50×), anti-CD8 (clone MEM-31, Exbio #1P-207-T025, diluted 50×), anti-CD69 (clone FN50, Exbio #T7-552-T100, diluted 100×), anti-CD271 (clone ME20.4, BioLegend #345108, #345106, diluted 200×) antibodies were used. Antibodies were conjugated with various fluorophores by the manufacturers.

For basal signaling analysis, the cells were stained with anti-Phospho-ZAP-70/SYK Y319 (polyclonal, Cell Signaling #2701, diluted 30×) and pTCRζ-PE (K25-407.69, BD Biosciences #558448, diluted 20x) antibodies overnight at 4°C protected from light and then with antibodies for surface markers and with goat-anti-rabbit-Alexa Fluor555 antibody (polyclonal, Thermo Fisher Scientific #A-32732, lot:1858260, 1000x diluted) in the case of pZAP70 staining

For surface CD4 and CD8 immunoprecipitation, 2-3×10e7 of live cells were stained with biotinylated anti-CD8β (clone 53-5.8, BioLegend, #140406, 2 μg) or anti-CD4 (clone H129.19, BD, #553649, 2 μg) antibodies.

For immunoblotting, following antibodies were used: murine anti-LCK (3A5, Santa Cruz, #sc-433, diluted 200x) and rabbit mAb anti-CD8α (D4W2Z, Cell Signaling, diluted 1000x) or anti-CD4 (D7D2Z, Cell Signaling, diluted 1000x), anti-LCK (3A5, Santa Cruz, #sc-433, diluted 200x), rabbit anti-β actin (#4967, Cell Signaling, diluted 1000x) and rabbit polyclonal anti-LAT antiserum, anti-CD3-ζ (clone 6B10.2, Santa Cruz #sc-1239, diluted 50x), LCK (clone 3A5, Santa Cruz #sc-433, diluted 200-500x), anti-TCRζ (pY142) (clone K25-407.69, BD Biosciences #558402, diluted 100x), anti-ZAP70 (clone 99F2, Cell Signaling #2705S, diluted 500x), phospho-Zap-70 (Try319)/Syk (Tyr352) (Cell Signaling #2701S, diluted 50x), anti-Actin (Cell Signaling #4967, diluted 5000x), anti-pTyr (clone 4G10, Sigma Aldrich # 05-321, diluted 5000x), and anti-FLAG (clone M2, Sigma-Aldrich # F1804-200UG, diluted 1000x).

| Validation | Commercially available antibodies were used in this study and we believe the trustful manufacturers performed a proper validation and QC control. We never experienced any sign that any antibody does not work properly during our work. We encourage anyone interested to check the manufacturers' web sites for specific information on validation.<br><br>The only non-commercial antibody was the anti-LAT serum. This was validated by the MW of the stained band (Extended Data Fig. 1c and the respective Source data). |
| --- | --- |

## Eukaryotic cell lines

Policy information about cell lines

| Cell line source(s) | No cell lines available at repositories or commercial vendors were used. Jurkat LCK KO cells, and Jurkat LCK KO cells reconstituted with LCK-Flag and/or OT-I TCR GFP were generated in the research groups of some of the authors previously and were described in Courtney, A.H., et al. Mol Cell, 2017. 67(3): p. 498-511.e6 and Lo, W.L., et al. Nature Immunology, 2018. 19(7): p. 733-741, respectively.<br><br>HEK293 cells were kindly provided by Dr. Tomas Brdicka (Institute of Molecular Genetics of the Czech Academy of Sciences, Czechia).<br><br>MC38 cells were kindly provided by Prof. Ed Palmer (Department of Biomedicine, University Hospital of Basel, Switzerland).<br>T2-Kb were kindly provided by Prof. Ed Palmer (Department of Biomedicine, University Hospital of Basel, Switzerland). |
| --- | --- |
| Authentication | The identity of the Jurkat lines were confirmed by the LCK-deficiency or expression of GFP, CD8, and OT-I TCR, respectively. The identity of MC38 and HEK293 was only based on their known morphology and adhesion to the tissue culture plastic. |
| Mycoplasma contamination | All cell lines were negative for mycoplasma as revealed by regular PCR testing. |
| Commonly misidentified lines (See ICLAC register) | HEK293 cells are listed in the Register of cell lines that are known to be misidentified through cross-contamination or other mechanisms (iclac.org/databases/cross-contaminations/), because there was a case of their confusion with HeLa cells. We can exclude such a misidentification in our culture based on the morphology of the cells and their adhesion on the tissue culture plastic (HeLa cells need to be trypsinized), which are clearly distinct between these two lines and which were checked in each experiment. The reason for using HEK293 cells was that they are easily transfectable and negative for ZAP70, LCK, and TCRzeta, as confirmed by Immunoblotting (Extended Data Figure 2a-b). |

## Animals and other organisms

Policy information about studies involving animals; ARRIVE guidelines recommended for reporting animal research

| Laboratory animals | The mice used in experiments had C57BL/6J background (Charles River). For the isolation of thymi for immunoblotting, 4-8 weeks old mice were used. For fetal thymic organ cultures, embryos of embryonic age E15.5 were used. In other experiments, 6-12 weeks old mice were used. Both males and females were used for experiments. We aimed at constant males and female representation among experimental groups in all experiments. The used congenic/transgenic strains were: Ly5.1 1, Cd3εKO/KO 2, OT-I Rag2KO/KO 3, 4, B3K508 Rag2KO/KO 5, Lck KO/KO 6, CD8.4 OT-I Rag2KO/KO 7,8. The colonies of all transgenic strains were established de novo in our animal facility by rederivation using embryotransfer or in vitro fertilization.<br><br>LckC20.23A/C20.23A and LckK273R/K273R knock-in mice and LckKO/KO mice were generated in the Czech Centre for Phenogenomics, IMG using 3-5 weeks old females and 9-35 weeks old males of C57BL/6N strain as parents. The founders were back-crossed on C57BL/6J background for at least 5 generations.<br><br>Mice were fed with an irradiated standard rodent breeding diet and given reverse osmosis filtered water ad libitum. They were kept in a facility with a 12h/12h light/dark cycle and temperature and relative humidity maintained at 22 ± 1 °C and 55 ± 5 %, respectively. |
| --- | --- |

| Wild animals | The study did not include wild animals. |
|---|---|
| Field-collected samples | The study did not include field collected samples. |
| Ethics oversight | Animal protocols (ID 11/2016, 115/2016, 72/2017, AVCR 2378/2022 SOVII) were approved by the Resort Professional Commission for Approval of Projects of Experiments on Animals of the Czech Academy of Sciences, Czech Republic. |

Note that full information on the approval of the study protocol must also be provided in the manuscript.

# Flow Cytometry

## Plots

Confirm that:

☒ The axis labels state the marker and fluorochrome used (e.g. CD4-FITC).

☒ The axis scales are clearly visible. Include numbers along axes only for bottom left plot of group (a 'group' is an analysis of identical markers).

☒ All plots are contour plots with outliers or pseudocolor plots.

☒ A numerical value for number of cells or percentage (with statistics) is provided.

## Methodology

| Sample preparation | Single cell suspensions from the spleen, lymph nodes, and thymus were prepared by gentle meshing the organs with syringe plungers. The samples from MC38 tumors were prepared as follows: Tumors were excised from mice, cut into small pieces and incubated with 100 µg/ml Liberase (Roche #5401020001) and 50 µg/ml DNAse I (Roche #101104159001) in wash buffer (1% BSA, 1mM EDTA in HBSS w/o Ca2+/Mg2+) at 37 °C and 350 rpm shaking for 45 minutes. Mixture was resuspended with a 1000 µl wide bore pipette tip every 10 min. Undigested debris was removed by filtering through a 100 µm strainer. Cells were harvested by centrifugation at 350g at 4° C for 5 min. Pellets were resuspended in 10 ml of 40% Percoll (Cytiva #17089101) in DMEM. 10 ml of 80% Percoll in DMEM was carefully laid to the bottom of the tube to create a gradient. Samples were centrifuged at 320g at ~21 °C for 23 min with minimal ascending/descending rates. Lymphocytes present at the interphase were collected, centrifuged at 400g at 4 °C for 5 min and processed for flow cytometry analysis. |
|---|---|
| Instrument | Aurora (Cytek™ Biosciences), LSRII and FACSymphony (BD Biosciences) |
| Software | FlowJo 10.6.2 (BD Biosciences) |
| Cell population abundance | We did not sort cells in this project. |
| Gating strategy | The lymphocytes were gated using FSC vs SSC. The singlets were gated based on the area vs height of FSC. Viable cells were gated based on the exclusion of near-infrared LIVE/DEAD Near-IR viability dye (ThermoFisher). In the next step, cell subsets were gated based on the antibody signal, as shown in the Figures and Supplemental Figures. |

☒ Tick this box to confirm that a figure exemplifying the gating strategy is provided in the Supplementary Information.

