## [Peer Review File · Nature Immunology]

Peer Review Information

Journal: Nature Immunology

Manuscript Title: Unique roles of coreceptor-bound LCK in helper and cytotoxic T cells

Corresponding author name(s): Ondrej Stepanek

Reviewer Comments & Decisions:

EA, duplicate for each version as needed, and then delete this instruction.

Decision Letter, initial version:
--

7th Jan 2022

Dear Ondrej,

Your manuscript entitled, "Unique roles of coreceptor-bound LCK in helper and cytotoxic T cells" has now been seen by 2 referees. Both referees expressed interest in the study, but they have raised quite several concerns (posted below) that must be addressed. Both expressed that the thymic phenotype of the Lck-CA, Lck-KR and the Lck-CA/KR mice need to be better characterized - especially to determine if there are any defects in the DP selection checkpoint. Although not explicitly stated by either referee, it might be useful to perform TCR repertoire analyses to interrogate any differences between the mutant Lck and WT mice. In light of these comments, we cannot accept the current manuscript for publication, but would be very interested in considering a revised version that addresses these concerns.

We invite you to submit a substantially revised manuscript, however please bear in mind that we will be reluctant to approach the referees again in the absence of major revisions.

When you revise your manuscript, please take into account all reviewer and editor comments, please highlight all changes in the manuscript text file in Microsoft Word format.

* If you have not done so already please begin to revise your manuscript so that it conforms to our Article format instructions at <http://www.nature.com/ni/authors/index.html>. Refer also to any guidelines provided in this letter.

The Reporting Summary can be found here:

[REDACTED]

If you wish to submit a suitably revised manuscript we would hope to receive it within 6 months. If you cannot send it within this time, please let us know. We will be happy to consider your revision so long as nothing similar has been accepted for publication at Nature Immunology or published elsewhere.

Nature Immunology is committed to improving transparency in authorship. As part of our efforts in this direction, we are now requesting that all authors identified as 'corresponding author' on published papers create and link their Open Researcher and Contributor Identifier (ORCID) with their account on the Manuscript Tracking System (MTS), prior to acceptance. ORCID helps the scientific community achieve unambiguous attribution of all scholarly contributions. You can create and link your ORCID from the home page of the MTS by clicking on 'Modify my Springer Nature account'. For more information please visit www.springernature.com/orcid.

Thank you for the opportunity to review your work.

Kind regards,

Laurie

Laurie A. Dempsey, Ph.D.
Senior Editor
Nature Immunology
l.dempsey@us.nature.com
ORCID: 0000-0002-3304-796X

Referee expertise:

Referee #1: T cell development

Referee #2: TCR signaling

Reviewers' Comments:

Reviewer #1:

Remarks to the Author:

In this study, the authors generated a series of *lck* mutant mice to study the roles of CD4- and CD8-*lck* binding in T-cell maturation and activation. By comparing the phenotypes of the different *lck* variant strains, they found out a substantial functional difference of CD4 and CD8 coreceptor in using *lck* kinase. The main conclusion is that CD4 coreceptor counts more on *lck* binding rather than its activity for CD4 T cell development and activation, while CD8 coreceptor relies more on *lck* kinase activity for CD8 T cell maturation as well as to respond to suboptimal antigens. This finding is interesting and important, which can clarify many previous observations in the field. The conclusions are generally supported by the experimental results.

The generation of *lck*^{CA/CA} and *lck*^{CA/KR} mice enabled the authors to study thymocyte maturation and function since these two strains exempt the DN blockade phenotype in *lck* KO and KR mice. However, the major concern also comes from the models since a partial block of positive selection still exists in these mouse strains and it might affect the subsequent functional analysis in peripheral.

Major issues:

In figure 1d and more profoundly in figure 1g-h, the results indicated a defective positive selection in *lck*^{CA/CA} and *lck*^{CA/KR} mice. If compared to the WT mice, the mature SP thymocytes derived from these two mutants are 5 time less. The possibility that this defect in the selection, as well as the compensation effect, could affect the subsequent peripheral T-cell activation and effector function readouts needs to be discussed.

In figure 2, the *lck* KO mice are included in the tumor experiment, but not in the LCMV infection experiment. Based on the fact that both the KO and KR strains contain severe immunodeficient due to early thymic development blockade, it is not necessary to include these two strains in both experiments.

In Figure 3AB, if we compare the cell number of mature CD8 SP thymocytes, it is clear that the OT-I Tg expression not only rescued the cell number loss in figure 1d but also promote the generation of more cells. Another difference between the results in Figure 1 and Figure 3 is that the OT-I *CA/KR* mice contain fewer CD8 SP cells when compared to WT and CA mice. This is not true without OT-I TCR transgene. These differences need to be described and discussed in the manuscript.

Figure 3e clearly showed a decreased responses of *lck*^{CA/CA} and *lck*^{CA/KR} CD8 to lower-affinity ligands. I would expect that it might be similar for double-positive thymocytes during their positive selection. The same experiment needs to be performed in DP thymocytes. It would also need to be discussed, whether it will influence the results obtained from peripheral mature CD8.

In Fig 5a, similarly, the B3K508 TCR transgenic background also rendered Lck CA/CA mice more CD4 SP generation compared to WT mice. Unlike what happened in B6 (Figure 1d), the KR rescue CD4 SP number, the KR here reversed the increased CD4 SP cell number in CA/CA mice. These differences need to be discussed.

Figure 6A has shown the downregulation of CD4 molecules in KO and CA LN CD4 T cells. However, the downregulation of CD4 happens as early as at the DP stage, as could be seen in Figures 1b, 1e 5a, and 5c. Thus, KR seems to do a better job in promoting CD4 surface expression in all different settings. This needs to be described and explained.

Reviewer #3:

Remarks to the Author:

Horkova et al. presented an interesting and timely research on Lck function by elegant design of mouse models. In particular, the novel Lck CA/KR mutant mice derived as heterozygotes from Lck CA and KR lines express 'free' Lck with kinase activity and coreceptor bound/kinase-dead Lck which permit study of two forms of Lck. It is a highly debated topic in T cell biology to prove how exactly Lck is functioning. The authors obtained mouse models expressing normal levels of Lck by CRISPR/Cas9 mediated knock-in and performed in vivo experiments to elucidate T cell function in such Lck mutant mice. The findings in this study are novel and the data is convincing and of high quality. The results from this study may help resolve the current controversies of Lck.

Suggested improvements:

Major points:

1. To quote the authors "Overall, the heterozygous LckCA/KR mice showed impaired anti-tumor and anti-viral responses in comparison to the LckWT/WT and LckCA/CA mice, suggesting that the CD8-coupled kinase-dead LCK blocks, not promotes, TCR activation", but in Figure 2, for both LCMV infection and tumor progression models, no specific experimental data concerning CD8+ T cells were provided. I understand that in the result sections afterward, TCR transgenic mice on Lck mutant backgrounds are analyzed. In this figure, in mice without transgenic TCR the authors still should provide CD8 T cell data at least for the LCMV model to explain that Lck CA mice had comparable viral load to Lck WT mice and KR mutation displayed impaired anti-viral activity.
2. The major advancement of this study as stated by the authors is that coreceptor bound Lck molecules are studied in vivo for their function in immune response. I suggest in Figure 2, the authors should stain endogenous LCMV-specific CD4 or CD8 T cells by tetramers.

Minor points:

1. It will be interesting for the readers to know that CA mutant protein selectively affects Lck binding to coreceptors but not affects its interaction with the other key substrates such as Zap70/CD3 family proteins in the knock-in model of this study. As a previous study, reference 12, showed that CA mutant proteins had stronger in vivo signaling in their transgenic mice.

2. In Figure 1, in Lck engineered mice without TCR transgenes, the authors found that CD4+ T cells depending more on coreceptor bound Lck. In thymi of Lck AC/AC mice, both CD4+ and CD8+ mature cells were slightly lower in numbers, but only in CD4+ cells KR mutant allele increased the cell number. Kinase-independent role of coreceptor bound Lck was not found for CD8+ T cells. CD8+ LN cells were the same between WT, Lck CA/CA and Lck AC/AR mice. However, the bone marrow competitive model showed obvious deficiency of all the Lck mutant mice for both CD4+ and CD8+ T cells. In Supp. Fig.2, Lck CA mice had 3-fold more CD44 high cells than Lck WT mice (44.4% versus 13.5%), even though these two groups of mice had very mild differences in thymus analyses. Such data may suggest that in Lck CA mice, T cells expressing TCR with high affinity pass thymic selections and they can be autoreactive. From the data presented, the authors made precise and reasonable interpretations by describing “These results suggested a kinase-independent role of CD4-LCK, but not CD8-LCK, in the positive selection of thymocytes”. But in the discussion section, it might be misleading for the authors to say “Our experiments with monoclonal TCR transgenic mice revealed that the interactions of LCK with CD8 and CD4 are not essential for maturation of conventional MHCII restricted T cells as proposed previously [12]”, considering the obvious deficiency of Lck CA mice in bone marrow chimera model, and extraordinarily large proportion of CD44high CD8+ T cells in LN.

3. In Figure 1, it is also necessary to show activation and proliferation of T cells from Lck CA/CA and LCK CA/KR mice following in vitro TCR stimulation with WT controls.

4. “The LckKR/KR mice showed even more severe phenotype than the LckKO/KO mice, suggesting that LCKKR is a dominant negative variant, plausibly preventing the phosphorylation of the TCR complex by other kinases, such as FYN”. Data for phosphorylation of the TCR complex should be provided.

5. In Figure 2, the authors should analyze regulatory T cells which are also PD-1+CXCR5+ to avoid Tfh contamination.

6. “Jurkat cells expressing CD8WT and CD8CA showed ~330 fold and ~35 fold higher signaling potency (measured as 1/EC50) to OVA pulsed antigen presenting cells than CD8 negative cells, respectively (Fig. 3f, S3b)”, results in S3b are not related to Jurkat cell.

7. To quote the authors, “These results indicated that the CD4-LCK interaction is not required for proper commitment of pMHCII-restricted T cells to the CD4+ T-cell lineage”. I would like this is rephrased since loss of CD4-Lck interaction in Lck CA T cells resulted in defective competition in bone marrow chimera

experiment. In the transgenic Lck-MUT model described in reference 12, the identical CA mutations severely inhibited positive selection signaling by MHC-restricted AND TCR.

8. “The LckCA/CA B3K508 T-cells exhibited weaker ex vivo antigenic responses to the cognate 3K peptide and its intermediate- and low-affinity variants (P5R and P2A, respectively) than the LckWT/WT B3K508 T cells (Fig. 4e). The LckCA/KR B3K508 T cells partially rescued defective responses to high-affinity 3K and intermediate-affinity P5R antigens, but not to low-affinity antigen P2A (Fig. 4e). Accordingly, we observed weaker response of the LckCA/CA B3K508 T cells to Lm expressing 3K or low-affinity P2A in vivo (Fig. 4f). The LckCA/KR T cells showed rescued responses to Lm-3K, but not to P2A (Fig. 4f)”. The figure numbers are mistakenly presented.

9. For the discussion section, it is a big step forward for understanding of Lck as this study provided in vivo data of innovative models and found the differences between CD4-Lck and CD8-Lck interactions. The authors may consider how these new findings promote current understanding of Lck during T cell development, differentiation in health and disease. In the discussion section, I largely disagree when the authors cited reference 24 and connect it to what the authors wrote “This suggests that there might be as few as one position at the TCR complex, where LCK can efficiently phosphorylate the ITAMs. This unique position would be occupied by an LCK molecule recruited by a coreceptor preferentially”. The reference 24 by Hartl et al. suggests that the RK motif in CD3e boosts both TCR and CAR signaling due to direct recruitment of Lck and irrespective of Lck association with coreceptors.

10. For the last paragraph of discussion, from the data in this study, my understanding is that inhibition of Lck-coreceptor interaction may result in selection of high affinity autoreactive TCR.

Author Rebuttal to Initial comments

Point-by-point Response to the Comments by the Editor and Referees

Editor:

Both expressed that the thymic phenotype of the Lck-CA, Lck-KR and the Lck-CA/KR mice need to be better characterized - especially to determine if there are any defects in the DP selection checkpoint. Although not explicitly stated by either referee, it might be useful to perform TCR repertoire analyses to interrogate any differences between the mutant Lck and WT mice.

We are very thankful for the positive evaluation of our manuscript by the editor and both referees and for their valuable comments. We did our best to address all the comments, which substantially increased our understanding of the role of coreceptor-LCK interaction in the development and function of T cells in the manuscript and improved the quality of the manuscript.

A large part of the new data focuses on the development of DP thymocytes in our model strains, which was analyzed from different angles (i.e., maturation of DP cells, steady-state TCR signaling analysis of thymocytes, ex vivo signaling of monoclonal DP/SP and pre-selection DP thymocytes, fetal thymic organ cultures, analysis of TCR repertoires in thymocytes and peripheral T cells). Overall, all these assays pointed to the fact that the maturation of DP T cells is largely normal with a partial defect at the very final stage in the Lck^{CA/CA} mice. Actually, the signaling of pre-selection DP T cells seems to be normal or even slightly stronger in the Lck^{CA/CA} mice in comparison to the Lck^{WT/WT} mice. This can be caused by the fact that the lack of the LCK binding to the proper coreceptor (i.e., CD8 for MHCI-restricted thymocytes and CD4 for MHCII-restricted thymocytes) is compensated by the 'free' LCK released from the other coreceptor. Another potentially important factor is that the role of LCK in the stabilization of CD4 is much less pronounced in DP thymocytes than in SP4 thymocytes and peripheral CD4⁺ T cells. Our data document that the major developmental block in the Lck^{CA/CA} occurs at the post-selection stages, probably at the very transition from the DP stage to the SP stages, and/or during the SP stage, where additional, so far unappreciated, TCR signaling-dependent checkpoints might occur. We believe that it would be interesting to study the role of the coreceptor-LCK in the thymic development in a greater detail, as there are still a few unanswered questions. However, we are convinced that this is beyond the scope of our current manuscript, which mostly focused on the direct and indirect roles of coreceptor-LCK interaction in the T-cell immune responses.

We analyzed the TCR repertoire in mature SP4 and SP8 thymocytes and in peripheral CD4⁺ and CD8⁺ T cells from the Lck^{WT/WT}, Lck^{CA/CA}, Lck^{CA/KR} (Fig. 2g-i, Extended Data Figure 7, Supplemental Fig. 1-3). Briefly, the TCR repertoires in thymocytes showed clear differences between the strains. Lck^{CA/CA} and Lck^{CA/KR} showed a high frequency of NKT TCR sequences, reflecting the advantage of the unconventional MHCI/II-unrestricted T cells in these mice. However, the TCR repertoires in the periphery were much more similar among the strains as revealed by the V-segment usage, PCA, frequency of top 20 TCRA and TCRB sequences and the overlap of identical CDR3 sequences, although the repertoires of Lck^{CA/CA} and Lck^{CA/KR} were less diverse than that of Lck^{WT/WT}.

Moreover, we identified normal or only slightly reduced numbers of virus-specific CD8⁺ and CD4⁺ T cells during the LCMV infection and tumor-specific CD8⁺ T cells in the tumor and tumor-draining lymph nodes in the Lck^{CA/CA} and Lck^{CA/KR} mice by specific MHC tetramer staining. This shows that the T-cell pool contains conventional T cells recognizing the immunogenic epitopes of viruses and tumors presented by MHCI and

MHCII. Thus, the coreceptor-LCK interactions are not essential for the development of MHCII-restricted T cells and for their proper CD4/CD8 lineage commitment. The disruption of the coreceptor-LCK interaction induces quantitative changes in T-cell repertoires, which are smaller than one might expect based on current literature.

Reviewer #1:

Remarks to the Author:

In this study, the authors generated a series of lck mutant mice to study the roles of CD4- and CD8-lck binding in T-cell maturation and activation. By comparing the phenotypes of the different lck variant strains, they found out a substantial functional difference of CD4 and CD8 coreceptor in using lck kinase. The main conclusion is that CD4 coreceptor counts more on lck binding rather than its activity for CD4 T cell development and activation, while CD8 coreceptor relies more on lck kinase activity for CD8 T cell maturation as well as to respond to suboptimal antigens. This finding is interesting and important, which can clarify many previous observations in the field. The conclusions are generally supported by the experimental results.

The generation of lckCA/CA and lckCA/KR mice enabled the authors to study thymocyte maturation and function since these two strains exempt the DN blockade phenotype in lck KO and KR mice. However, the major concern also comes from the models since a partial block of positive selection still exists in these mouse strains and it might affect the subsequent functional analysis in peripheral.

We are thankful for the positive evaluation of our study and for the specific comments that motivated us to carry out a bunch of additional experiments which substantially increased our understanding of what is going on during the thymic development in these mice. Please, see, how we addressed the specific comments below.

Major issues:

1. In figure 1d and more profoundly in figure 1g-h, the results indicated a defective positive selection in lck CA/CA and lck CA/KR mice. If compared to the WT mice, the mature SP thymocytes derived from these two mutants are 5 times less. The possibility that this defect in the selection, as well as the compensation effect, could affect the subsequent peripheral T-cell activation and effector function readouts needs to be discussed.

First of all, we would like to mention that we were primarily interested in the overall role of the coreceptor-LCK interaction in the immune response and we initially did not focus much on whether the effect is caused by defective thymic selection or defective peripheral responses. Actually, the immune response of Lck^{CA/CA} is surprisingly normal (or almost normal), which suggests that both the thymic selection as well as the peripheral response work relatively well. However, we agree that the additional information about the selection process is of a great interest and improves the quality and the importance of our study.

The detailed analysis of the thymic development revealed that the signaling and development of DP thymocytes is surprisingly normal up to the very late DP stage, which explains the near-normal TCR repertoire in the $Lck^{CA/CA}$ and $Lck^{CA/KR}$ mice.

We did following experiments:

1. We analyzed the DP compartments using an unsupervised dimensional reduction and clustering approach (SOM), which revealed that the frequency of the most mature DP thymocytes (corresponding to CD5, CD69, TCR β triple positive) is reduced in the $Lck^{CA/CA}$ mice, but this reduction is relatively small in comparison to $Lck^{WT/WT}$ mice, and is rescued in the $Lck^{CA/KR}$ mice (Fig. 2a-b, Extended Data Fig. 5a-c). This indicates that the major developmental block occurs at the transition into the SP stage or even afterwards and probably does not impact the positive selection of DP thymocytes.
2. We analyzed the steady-state TCR signaling in different thymocyte stages of the $Lck^{WT/WT}$, $Lck^{CA/CA}$ and $Lck^{CA/KR}$ mice by detecting the phosphorylation of TCR ζ and ZAP70 (Fig. 2c-d, Extended Data Figure 5e-f). We observed that the phosphorylation of these proximal TCR signaling molecules increases as the cells progress through the maturation. Interestingly, the signaling in (mostly pre-selection) TCR^{low} DP thymocytes is even slightly stronger in the $Lck^{CA/CA}$ and $Lck^{CA/KR}$ mice than in the $Lck^{WT/WT}$ mice, however this difference disappears during the maturation.
3. We performed the analysis of the TCR repertoire in mature SP thymocytes and in peripheral LN cells of the $Lck^{WT/WT}$, $Lck^{CA/CA}$ and $Lck^{CA/KR}$ mice by high-throughput sequencing (Fig. 2g-i, Extended Data Fig. 7, Supplemental Fig. 1-3). We observed relatively striking differences in the mature SP thymocytes. For instance, these mice have much higher frequency of NKT-cell TCRs than the WT mice. However, the differences between the peripheral repertoires are much smaller as shown by the PCA, the high frequency of dominant WT clones in all mice, and by the overlap between unique CDR3 sequences. The only major difference in the peripheral repertoires was slightly lower diversity of T cells in the $Lck^{CA/CA}$ and $Lck^{CA/KR}$ mice.
4. We compared the TCR response of monoclonal thymocytes to the antigens with different affinities *ex vivo*. Surprisingly, the signaling of $Lck^{CA/CA}$ and $Lck^{CA/KR}$ OT-I as well as B3K508 cells was not defective at the DP stage, but only at the SP stage (Fig. 2e-f). This corresponded well with the analysis of thymic subpopulations and steady-state signaling as described above. Accordingly, the response of pre-selection thymocytes from OT-I $Rag2^{KO/KO}$ $B2m^{KO/KO}$ mice was comparable in the $Lck^{WT/WT}$, $Lck^{CA/CA}$ and $Lck^{CA/KR}$ mice and if there were any differences, it was a slightly increased signaling in the cells from the $Lck^{CA/CA}$ knock-in mice (Extended Data Fig. 6b).
5. We performed a fetal thymic organ culture experiments using $Lck^{WT/WT}$, $Lck^{CA/CA}$ and control $Lck^{KO/KO}$ thymi on the $Rag2^{KO/KO}$ $B2m^{KO/KO}$ background in positive and negative selecting conditions. We did not observe substantial differences between $Lck^{WT/WT}$ and $Lck^{CA/CA}$ thymi. If there was remarkable difference, it was a higher proportion of CD8 $\alpha\alpha$ /CD8 $\alpha\beta$ SP T cells in the $Lck^{CA/CA}$ thymi upon the addition of a positive selecting peptide Q4H7. This indicated that the pre-selection DP thymocytes of $Lck^{CA/CA}$ mice have slightly stronger signaling than the ones from the $Lck^{WT/WT}$ mice.

Overall, we think that the defect in the thymic development does not occur at the positive selection step of DP T cells. This is in line with the TCR sequencing analysis, which did not reveal much differences in the peripheral TCR repertoires, which is quite surprising. Moreover, our results from signaling experiments using DP thymocytes from monoclonal mice, which do not have reduced numbers of mature SP thymocytes

(also see our response to Comments 3 and 5 of this Referee), are in a good agreement with the phenotype of the polyclonal T cells/mice (e.g., ex vivo signaling in monoclonals and steady state signaling in polyclonals). However, although the thymic developmental phenotype is much weaker than expected in the $Lck^{CA/CA}$ and $Lck^{CA/KR}$ mice, we cannot formally exclude that the different thymic history imprints differences in peripheral T cells from these strains. We added this point to the Discussion.

2. In figure 2, the lck KO mice are included in the tumor experiment, but not in the LCMV infection experiment. Based on the fact that both the KO and KR strains contain severe immunodeficient due to early thymic development blockade, it is not necessary to include these two strains in both experiments.

We agree with the reviewer that the analysis of the peripheral responses of both these strains is not necessary. However, we included it in these experiments to have a control showing that the response to LCMV and tumors depends on the LCK activity (of course largely indirectly via the block of T-cell development here). The decision for not including Lck KO mice in the LCMV experiment was that these mice were not available at the time of these experiments together with the fact that it was indeed not that important to have them. Although these data might not be that exciting, we decided to keep it in the revised version of the manuscript, as we do not think this can make any harm. During the revision, we mostly focused on the comparison of the $Lck^{WT/WT}$, $Lck^{CA/CA}$ and $Lck^{CA/KR}$ mice, which is the major focus of this study. We also excluded the $Lck^{KO/KO}$ mice from the additional experiments in MC38-OVA tumors (Extended Data Fig. 8j-l and Supplemental Fig. 4e-h).

3. In Figure 3AB, if we compare the cell number of mature CD8 SP thymocytes, it is clear that the OT-I Tg expression not only rescued the cell number loss in figure 1d but also promote the generation of more cells. Another difference between the results in Figure 1 and Figure 3 is that the OT-I CA/KR mice contain fewer CD8 SP cells when compared to WT and CA mice. This is not true without OT-I TCR transgene. These differences need to be described and discussed in the manuscript.

The referee is correct that surprisingly, we do not see any effect of the $Lck^{CA/CA}$ or $Lck^{CA/KR}$ knock-ins on the maturation of OT-I (and also F5) monoclonal T cells, which is very different to polyclonal mice. Actually, $Lck^{CA/CA}$ OT-I T cells form even slightly more SP8 T cells than their WT counterparts. We believe that the major reason is the lack of a major phenotype at the DP stage coupled with the early expression of the monoclonal TCR and/or the lack of competition with unconventional MHCII-unrestricted TCRs (which have a comparative advantage against MHCII-restricted thymocytes in mice with disrupted coreceptor-LCK binding).

We explicitly mention this observation in the revised version of the manuscript and we provide the possible explanations.

4. Figure 3e clearly showed a decreased responses of lck CA/CA and lck CA/KR CD8 to lower-affinity ligands. I would expect that it might be similar for double-positive thymocytes during their positive selection. The same experiment needs to be performed in DP thymocytes. It would also need to be discussed, whether it will influence the results obtained from peripheral mature CD8.

We agree with the reviewer that it can be expected that DP thymocytes from LCK^{CA/CA} and Lck^{CA/KR} T cells will be hyper-responsive to weak antigens. Surprisingly, we observe that this is the case only in single positive thymocytes, but not in DP thymocytes, from OT-I or B3K508 Rag2^{KO/KO} mice (Fig. 2e-f). We obtained similar results when we used OT-I Rag2^{KO/KO} B2m^{KO/KO} thymocytes, which are arrested in the pre-selection stage. Actually, the signaling in the DP thymocytes was even slightly stronger in the Lck^{CA/CA} than in Lck^{WT/WT} (Extended Data Fig. 6b). Although these results are very surprising at the very first sight, they are in a good agreement with the results of complementary experiments focused on the development of DP thymocytes, steady-state signaling in pre-selection DP thymocytes, and the repertoire analysis of peripheral T cells in polyclonal mice (see our response to the Comment 1 of this Referee).

The most plausible explanation is that the LCK^{CA} does not bind either of the coreceptors. From the point of view of the MHC I-restricted T cells there are three pools of LCK, CD8-bound, CD4-bound and 'free'. Whereas the CD8-bound LCK augments the signaling most efficiently, 'free' Lck is intermediate and the CD4-bound LCK is probably the worst. In Lck^{CA/CA} mice, CD8-Lck is lost, but it can be compensated by the concomitant loss of CD4-LCK, as only the 'free' LCK is present. The similar reciprocal principle probably applies for MHC II-restricted thymocytes. This effect does not apply for SP thymocytes and mature T cells, as they lack the non-signaling coreceptor.

Another important point is that the LCK-mediated stabilization of CD4 is much less pronounced in DP thymocytes than in SP thymocytes and mature T cells (Supplementary Fig. 4d-e, see our response to the Comment 6 of this Referee).

We did not observe any difference in the response of OT-I T cells from the Lck^{WT/WT}, Lck^{CA/CA}, or Lck^{CA/KR} mice to the anti-CD3/CD28-mediated stimulation (Extended Data Fig. 9f-g, see response to the Minor point 3 of the Referee 2). This documents that their response to coreceptor-independent stimulation is comparable and suggests that they probably do not have substantial differences imprinted in the thymus that would diverge their TCR signaling responses.

We added these new experiments and their discussion to the new version of the manuscript.

5. In Fig 5a, similarly, the B3K508 TCR transgenic background also rendered Lck CA/CA mice more CD4 SP generation compared to WT mice. Unlike what happened in B6 (Figure 1d), the KR rescue CD4 SP number, the KR here reversed the increased CD4 SP cell number in CA/CA mice. These differences need to be discussed.

This is a similar situation to the MHC I-restricted monoclonal T cells (OT-I and F5). The reasons for the different phenotype of the Lck^{CA/CA} than was observed in the polyclonal compartments are probably again non-physiological timing of the monoclonal TCR (coupled with slightly stronger signaling in early DP stages in the Lck^{CA/CA} mice than in the WT Lck^{CA/CA} and only a marginal role of LCK in stabilizing CD4 in DP thymocytes) and/or the lack of competition between the MHC II-restricted and unconventional T cells in the monoclonal mice.

Importantly, we did not observe any difference in the response of B3K508 T cells from the Lck^{WT/WT}, Lck^{CA/CA}, or Lck^{CA/KR} mice to the anti-CD3/CD28-mediated stimulation (Extended Data Fig. 10a-b, see response to the Minor point 3 of the Referee 2). This documents that their response to coreceptor-independent stimulation

is comparable and suggests that they probably do not have substantial differences imprinted in the thymus that would diverge their TCR signaling responses.

We mention these observations in the Results and discuss them in the Discussion of the revised version of the manuscript.

The reason, why the $Lck^{CA/KR}$ does not increase the number of SP4 thymocytes in comparison to the $Lck^{CA/CA}$, as observed in the polyclonal settings, is probably the fact that the $Lck^{CA/CA}$ do not show any defect in their formation to be rescued, at the first place. The Lck^{KR} allele has ambiguous roles: (i) stabilization of CD4 and (ii) dominant negative role over coreceptor-free LCK. Although the latter was uncovered only in $CD8^+$ T-cells, we assume that it might be present also in CD4-dependent signaling, but is often masked by the beneficial effect of CD4 stabilization. Thus, the net effect of the Lck^{KR} allele might be the reversal of the Lck^{CA} -mediated enhancement of SP4 numbers in this case. This ambiguous role of the Lck^{KR} allele is discussed in the Discussion of the revised version of the manuscript.

6. Figure 6A has shown the downregulation of CD4 molecules in KO and CA LN CD4 T cells. However, the downregulation of CD4 happens as early as at the DP stage, as could be seen in Figures 1b, 1e 5a, and 5c. Thus, KR seems to do a better job in promoting CD4 surface expression in all different settings. This needs to be described and explained.

We are very thankful for this comment. We carefully quantified the levels of CD4 in DP and SP4 thymocytes in the Lck variant strains in polyclonal mice in multiple mice from multiple experiments (Extended Data Fig. 10d-e). Although, we observed that Lck stabilizes CD4 at both of these thymocytes stages, the effect was much weaker in DP thymocytes than in SP thymocytes, which resembled peripheral T cells in this respect (Fig. 7a-b). These new results are in line with our observation that the development, signaling, and selection of DP thymocytes is not very substantially altered in the $Lck^{CA/CA}$ mice. We would like to note that the CD4 levels in DP thymocytes in $Lck^{KO/KO}$ and especially in $Lck^{KR/KR}$ mice are higher than in $Lck^{WT/WT}$. This cannot be explained just by the LCK mediated stabilization and it is probably caused by the defective signaling and selection in these two strains with a severe phenotype at the DN and DP stages (Fig. 2a).

The FC plots in Fig. 1b were selected as representatives for the frequencies of particular thymocyte subsets. We realized that they are not representative for the CD4 levels. The representative experiment for CD4 levels on DP and SP4 levels is shown in Extended Data Fig. 10d-e along with the quantification from all experiments (of course including the one shown in Fig1b).

The FC plots in Fig. 6a (Fig. 5a in the revised version of the manuscript) corresponds to the polyclonal Lck variant strains (Extended Data Fig. 10d-e), as it shows lower CD4 levels in SP4 thymocytes of the $Lck^{CA/CA}$ mice in comparison to $Lck^{WT/WT}$ and $Lck^{CA/KR}$ mice, but very little differences in the CD4 levels in DP thymocytes. The CD4 levels are high on DP thymocytes from $Lck^{KO/KO}$ and especially from $Lck^{KR/KR}$ mice, which also corresponds to the polyclonal mice (Extended Data Fig. 10d-e) as mentioned above.

Reviewer #3:

Remarks to the Author:

Horkova et al. presented an interesting and timely research on Lck function by elegant design of mouse models. In particular, the novel Lck CA/KR mutant mice derived as heterozygotes from Lck CA and KR lines express 'free' Lck with kinase activity and coreceptor bound/kinase-dead Lck which permit study of two forms of Lck. It is a highly debated topic in T cell biology to prove how exactly Lck is functioning. The authors obtained mouse models expressing normal levels of Lck by CRISPR/Cas9 mediated knock-in and performed in vivo experiments to elucidate T cell function in such Lck mutant mice. The findings in this study are novel and the data is convincing and of high quality. The results from this study may help resolve the current controversies of Lck.

We are very thankful for the positive evaluation of our study by this reviewer and for their valuable comments that helped us a lot to increase the quality of the manuscript.

Suggested improvements:

Major points:

1. To quote the authors "Overall, the heterozygous LckCA/KR mice showed impaired anti-tumor and anti-viral responses in comparison to the LckWT/WT and LckCA/CA mice, suggesting that the CD8-coupled kinase-dead LCK blocks, not promotes, TCR activation", but in Figure 2, for both LCMV infection and tumor progression models, no specific experimental data concerning CD8+ T cells were provided. I understand that in the result sections afterward, TCR transgenic mice on Lck mutant backgrounds are analyzed. In this figure, in mice without transgenic TCR the authors still should provide CD8 T cell data at least for the LCMV model to explain that Lck CA mice had comparable viral load to Lck WT mice and KR mutation displayed impaired anti-viral activity.

We performed additional experiments focused on the analysis of CD4 and CD8 T cells during the LCMV infection and MC38 tumor experiments (Fig. 3e, Extended Data Figure 8a-l, Supplemental Fig. 4h). We quantified the total numbers of CD8 and CD4 T cells and their activation status using CD44⁺CD49d⁺ staining of activated T cells and the formation of KLRG1⁺CD127⁻ short-lived effectors T cells. We observed lower number of CD8⁺ as well as CD4⁺ T cells in the infected Lck^{CA/CA} mice, which was partially rescued in the Lck^{CA/KR} mice for the CD4 T-cell compartment (Extended Figure 8a-b). We did not observe clear differences in the activation status of CD8+ T cells (with the exception of the immunodeficient Lck^{KR/KR} mice) in all T cells (not shown) or in the LCMV-specific cells (Extended Data Fig. 8e-f, Supplemental Fig. 4b-c, see reply to the comment 2 of this reviewer).

Based on these data, the impaired anti-viral activity in Lck^{KR/KR} mice can be explained by impaired formation and/or expansion of LCMV-specific CD8⁺ T cells. The delayed anti-viral activity in the Lck^{CA/KR} mice can be at least partially explained by the low expansion of LCMV-specific CD8⁺ T cells (Fig. 3b, see reply to the Comment 2 of this Referee).

2. The major advancement of this study as stated by the authors is that coreceptor bound Lck molecules

are studied in vivo for their function in immune response. I suggest in Figure 2, the authors should stain endogenous LCMV-specific CD4 or CD8 T cells by tetramers.

We did stain polyclonal T cells with MHC tetramers for the specificity to LCMV dominant immunogenic antigens Db-NP396, Db-GP33, and I-Ab-GP66 on day 8 post-infection during the revision experiments (Fig. 3b, Extended Data Fig. 8c-f, h-l, Supplemental Fig. 4a). We were able to detect substantial numbers of LCMV-specific T cells in the $Lck^{WT/WT}$, $Lck^{CA/CA}$, and $Lck^{CA/KR}$ mice, showing that the selection of MHCI/II-restricted T cells is not severely impaired by the disruption of the coreceptor-LCK interaction. Moreover, we observed quantitative differences such as decreased frequency of LCMV-specific CD8⁺ T cells in the $Lck^{CA/KR}$ mice (Fig. 3b) and lower frequency of T_{FH} cells within the LCMV-specific CD4⁺ T cells (Fig. 3e).

We also identified T cells specific for the MC38-OVA tumor using Kb-OVA tetramers. Again, the tumor specific MHCI-restricted cells were present in all tested mouse strains. The frequencies of OVA-specific cells was higher in the tumor than in the tumor-draining lymph nodes showing the antigen-specific T cells infiltrate the tumor (Extended Data Fig. 8k-l, Supplemental Fig. 4f-h). Surprisingly, we did not observe a striking differences between the $Lck^{WT/WT}$, $Lck^{CA/CA}$, and $Lck^{CA/KR}$ mice, suggesting that the cytotoxic activity of the tumor-specific CD8⁺ T cells, rather than their infiltration into the tumor, is defective in the $Lck^{CA/KR}$ mice.

Minor points:

1. It will be interesting for the readers to know that CA mutant protein selectively affects Lck binding to coreceptors but not affects its interaction with the other key substrates such as Zap70/CD3 family proteins in the knock-in model of this study. As a previous study, reference 12, showed that CA mutant proteins had stronger in vivo signaling in their transgenic mice.

The referee probably refers to the Fig. 1c of the study by Van Laethem et al. (PMID: 24034254) showing higher CD5 levels on thymocytes in $Lck^{KO/KO}$ mice reconstituted with a Lck^{CA} transgene than in those reconstituted with a Lck^{WT} transgene. Accordingly, we also observed higher CD5 levels on DP thymocytes in the $Lck^{CA/CA}$ mice than in the $Lck^{WT/WT}$ mice (Extended Data Fig. 5a-b). It is true that CD5 is a proxy for self-antigen-mediated signaling in thymocytes. This is in line with our data showing that (pre-selection) DP thymocytes the $Lck^{CA/CA}$ mice experience slightly stronger TCR signaling (new Fig. 2, Extended Data Figure 6 in the revised version of the manuscript). We believe that this is probably caused by the compensation of the missing LCK from the engaged coreceptor (i.e., CD4 for MHCI-restricted and CD8 for MHCI-restricted thymocytes) by the release of LCK from the other non-signaling coreceptor (i.e., please, see also our response to the Comment 1 by the Referee 1).

Importantly, the levels of CD5 were only marginally increased in OT-I $Lck^{CA/CA}$ DP thymocytes and even slightly decreased in B3K508 $Lck^{CA/CA}$ DP thymocytes in comparison to their $Lck^{WT/WT}$ counterparts (Fig. 1 of this document), suggesting that the increased levels of CD5 are not caused by the hypothetical hyperactivity of Lck^{CA} , but rather by the selectively enhanced signaling in specific T-cell clones (conventional and/or unconventional).

To study, whether LCK^{CA} is intrinsically more active than LCK^{WT} , we tested all three LCK variants (including LCK^{KR}) in coreceptor-independent activation. First, we co-transfected LCK and their substrates (CD25-TCR ζ fusion construct, ZAP70) in Hek293 cells and observed similar levels of the substrate phosphorylation in

LCK^{WT} and LCK^{CA} (Extended Data Fig. 2a-b). Second, we activated LCK^{KO} Jurkat T cells reconstituted with the LCK variants using anti-TCR antibody (C305) and observed a comparable level of phosphorylation of TCR ζ and ZAP70 and over-all tyrosine phosphorylation by immunoblotting in LCK^{WT} and LCK^{CA} expressing cells (Extended Data Fig. 2c). Third, we activated OT-I transgenic LCK^{KO} Jurkat T cells (not expressing CD8) reconstituted with the LCK variants with OVA-loaded T2-Kb cells and observed a similar response of LCK^{WT} and LCK^{CA} expressing cells measured as CD69 up-regulation by flow cytometry (Extended Data Fig. 2d). Moreover, it is not very likely that the CA mutation would influence the activity of LCK as it is located at the membrane-proximal unstructured N-terminal part, far from the catalytic domain and the regulatory elements (SH2, SH3 domains) of the kinase. Moreover, we did not observe any difference in the response of monoclonal CD4 or CD8 T cells from the $Lck^{WT/WT}$, $Lck^{CA/CA}$, and $Lck^{CA/KR}$ mice to the anti-CD3/CD28-mediated stimulation (Extended Data Fig. 9f-g, Extended Data Fig. 10a-b, see response to the Minor point 3 of this referee), which suggests that there is no substantial intrinsic difference in the activity of the WT and CA variants of LCK.

Overall, we do not find any evidence or rationale of intrinsically increased activity of the LCK^{CA} kinase variant. However, it is still possible that the interaction of the kinase with the coreceptors increases or decreased its activity as was proposed by studies by Liaunardy-Jopeace et al (PMID: 29083415) and Wei et al. (PMID: 29083415), respectively, which can have some impact on the observed phenotypes. We included this point in the discussion of the revised version of the manuscript.

Figure 1 Normalized CD5 levels in DP thymocytes from the indicated mouse strains.

2. In Figure 1, in Lck engineered mice without TCR transgenes, the authors found that CD4+ T cells depending more on coreceptor bound Lck. In thymi of Lck AC/AC mice, both CD4+ and CD8+ mature cells were slightly lower in numbers, but only in CD4+ cells KR mutant allele increased the cell number. Kinase-independent role of coreceptor bound Lck was not found for CD8+ T cells. CD8+ LN cells were the same between WT, Lck CA/CA and Lck AC/AR mice. However, the bone marrow competitive model showed obvious deficiency of all the Lck mutant mice for both CD4+ and CD8+ T cells. In Supp. Fig.2, Lck CA mice had 3-fold more CD44 high cells than Lck WT mice (44.4% versus 13.5%), even though these two groups of mice had very mild differences in thymus analyses. Such data may suggest that in Lck CA mice, T cells expressing TCR with high affinity pass thymic selections and they can be autoreactive. From the data presented, the authors made precise and reasonable interpretations by describing “These results suggested a kinase-independent role of CD4-LCK, but not CD8-LCK, in the positive selection of thymocytes”. But in the discussion section, it might be misleading for the authors to say “Our experiments with monoclonal TCR transgenic mice revealed that the interactions of LCK with CD8 and CD4 are not essential for maturation of conventional MHCII restricted T cells as proposed previously [12]”, considering the obvious deficiency of Lck CA mice in bone marrow chimera model, and extraordinarily large proportion of CD44high CD8+ T cells in LN.

We agree with the reviewer that the quoted expression might be misleading. We added a large body of new experimental work focusing on the maturation and signaling of DP thymocytes, thymic and peripheral TCR repertoires (Fig. 2, Extended Data Fig. 7, Supplemental Fig. 1-3, see also the response to the Comment 1 by the Reviewer 1) and we discuss these results extensively in the Results and Discussion of the revised version of the manuscript.

Concerning the CD44⁺ CD49d⁻ T cells, we believe that these are not overtly self-reactive T-cells because of the relatively normal development of DP thymocytes in the Lck^{CA/CA} mice and relatively normal TCR repertoires in these mice (Fig. 2g-l, Extended Fig. 7, Supplemental Fig. 1-3). We believe that these cells are antigen-independent memory-like (AIMT) T cells, which are induced by lymphopenia-induced proliferation. Our lab has published a couple of studies on AIMT cells previously (e.g., Drobek et al. PMID: 29752423, Moudra et al. PMID: 33858960), so we have a relatively good hands-on experience with these cells. The most plausible scenario is that Lck^{CA/CA} mice have a decreased thymic output (actually the difference in the thymus is not very small, please not the logarithmic scale of the Y axis in the Fig. 1d, and the real output can be even smaller), which caused peripheral lymphopenia, which then induces the homeostatic proliferation of the CD8⁺ T cells, which is coupled with their differentiation into CD44⁺ CD49d⁻ AIMT cells. Thus, the high percentage of the AIMT cells in the Lck^{CA/CA} mice is probably not caused by a hypothetical high level of self-reactivity of the CD8⁺ T cell pool in these mice. We modified the Results and Discussion to be more clear in this point.

3. In Figure 1, it is also necessary to show activation and proliferation of T cells from Lck CA/CA and LCK CA/KR mice following in vitro TCR stimulation with WT controls.

We performed the anti-CD3/anti-CD28 ex vivo stimulation of peripheral T cells isolated from the Lck^{WT/WT}, Lck^{CA/CA}, and Lck^{CA/KR} mice (Extended Data Fig. 9f-g, 10a-b) with the upregulation of CD69 at 16 hours post-activation and proliferation on day 3 post-activation as read-outs. These experiments revealed no clear differences between the strains. We decided to perform these experiments on monoclonal OT-I and B3K508 cells to normalize for eventual differences in the phenotype of the T cells from different mice in the polyclonal setup (e.g., different frequency of AIMT cells – Extended Data Fig. 4b-c or Treg cells – Extended Data Fig. 4d) and to use these experiments for addressing the Comments 3 and 5 of the Referee 1 as well.

4. “The LckKR/KR mice showed even more severe phenotype than the LckKO/KO mice, suggesting that LCKKR is a dominant negative variant, plausibly preventing the phosphorylation of the TCR complex by other kinases, such as FYN”. Data for phosphorylation of the TCR complex should be provided.

We addressed the phosphorylation of TCRζ in the whole thymic lysates by immunoblotting and specifically in DP thymocytes using flow cytometry (Extended Data Fig. 2e, Extended Data Fig. 2f-h). Both these assays showed a lower level of TCRζ phosphorylation in the Lck^{KR/KR} mice than in the Lck^{KO/KO} mice, which supports our conclusions.

5. In Figure 2, the authors should analyze regulatory T cells which are also PD-1⁺CXCR5⁺ to avoid Tfh contamination.

The reviewer is correct that follicular regulatory T cells (sometimes called T_{FR} cells) have the FOXP3⁺ PD-1⁺ CXCR5⁺ phenotype. We repeated the experiment and included the intracellular staining of FOXP3 to the

panel. We observed that only a small percentage of PD-1⁺ CXCR5⁺ T cells are FOXP3⁺ in the LCMV infection (Extended Data Fig. 8g). When we compared the counts of FOXP3⁺ PD-1⁺ CXCR5⁺ T cells (Extended Data Fig. 8g), we observed a very comparable pattern to the original experiment, i.e, the WT>CA/KR>CA>KR hierarchy (Fig. 3d).

6. “Jurkat cells expressing CD8WT and CD8CA showed ~330 fold and ~35 fold higher signaling potency (measured as 1/EC50) to OVA pulsed antigen presenting cells than CD8 negative cells, respectively (Fig. 3f, S3b)”, results in S3b are not related to Jurkat cell.

Thank you for spotting this error. We have corrected it. The respective experiment is shown in the Fig. 4f of the revised manuscript.

7. To quote the authors, “These results indicated that the CD4-LCK interaction is not required for proper commitment of pMHCII-restricted T cells to the CD4+ T-cell lineage”. I would like this is rephrased since loss of CD4-Lck interaction in Lck CA T cells resulted in defective competition in bone marrow chimera experiment. In the transgenic Lck-MUT model described in reference 12, the identical CA mutations severely inhibited positive selection signaling by MHC-restricted AND TCR.

We changed the discussion accordingly. We also briefly discuss that the AND TCR in the transgenic system largely required the coreceptor-LCK interaction (Van Laethem et al., PMID: 24034254). It is unclear, why none of our three monoclonal tgTCRs tested showed even a marginal developmental impairment in the Lck^{CA/CA} mice. In the discussion, we provide two possible explanations.

First, it can be somehow connected with the transgenic system of Lck expression driven from the human Cd2 promoter. They compared the AND TCR in WT mice vs Lck^{KO/KO} + tgLck^{CA}, which is not ideal (Fig. 6c-d in the study by Van Laethem et al. 2013). The best experimental setup would be Lck^{KO/KO} + tgLck^{CA} vs Lck^{KO/KO} + tgLck^{WT}.

Second, the cause can be a uniqueness of the AND TCR. We observed that the peripheral TCR repertoires are relatively normal in the Lck^{CA/CA} mice (Fig. 2g-i), but they have lower diversity than the TCRs from the Lck^{WT/WT} mice (Extended Data Fig. 7d). Thus, it is possible that a fraction of TCRs does not mature in the Lck^{CA/CA} mice (but they do in the Lck^{WT/WT} mice). However, it is unclear what would be the specific feature of these TCRs. We can exclude the apparent possibility that these are TCRs with a low level of self-reactivity, because the F5 TCR has a relatively weak self-reactivity (e.g., Drobek et al. PMID: 29752423), whereas the AND TCR is relatively highly self-reactive (Mandl et al. PMID: 23290521).

8. “The LckCA/CA B3K508 T-cells exhibited weaker ex vivo antigenic responses to the cognate 3K peptide and its intermediate- and low-affinity variants (P5R and P2A, respectively) than the LckWT/WT B3K508 T cells (Fig. 4e). The LckCA/KR B3K508 T cells partially rescued defective responses to high-affinity 3K and intermediate-affinity P5R antigens, but not to low-affinity antigen P2A (Fig. 4e). Accordingly, we observed weaker response of the LckCA/CA B3K508 T cells to Lm expressing 3K or low-affinity P2A in vivo (Fig. 4f). The LckCA/KR T cells showed rescued responses to Lm-3K, but not to P2A (Fig. 4f)”. The figure numbers are mistakenly presented.

Thank you very much for spotting this error. We have corrected it.

9. For the discussion section, it is a big step forward for understanding of Lck as this study provided in vivo data of innovative models and found the differences between CD4-Lck and CD8-Lck interactions. The authors may consider how these new findings promote current understanding of Lck during T cell development, differentiation in health and disease. In the discussion section, I largely disagree when the authors cited reference 24 and connect it to what the authors wrote “This suggests that there might be as few as one position at the TCR complex, where LCK can efficiently phosphorylate the ITAMs. This unique position would be occupied by an LCK molecule recruited by a coreceptor preferentially”. The reference 24 by Hartl et al. suggests that the RK motif in CD3e boosts both TCR and CAR signaling due to direct recruitment of Lck and irrespective of Lck association with coreceptors.

We believe that this is a misunderstanding, as we did not want to propose that this RK motif in CD3e depends on the coreceptors or that it could be occupied only by a coreceptor-bound LCK. Rather we find it plausible that there is a higher chance that this site will be occupied by a coreceptor-bound LCK than by a free LCK molecule in situations when coreceptors are involved (i.e., not in coreceptor-independent signaling such as in CARs). This speculation is based on the assumption that co-receptor-bound LCK is on average recruited to the TCR faster than the free LCK. We are sorry for not being clear in the original version of the manuscript. We now realize that this idea is rather speculative and beyond the current scope of the manuscript. Also the discussion is now much longer because of the addition of new data. For this reason, we deleted the speculation referring to the study by Hart et al.

10. For the last paragraph of discussion, from the data in this study, my understanding is that inhibition of Lck-coreceptor interaction may result in selection of high affinity autoreactive TCR.

The new data (mostly Fig. 2, Extended Fig. 8, Supplemental Fig. 1-3 – discussed above) suggest that the selection window is probably not shifted to select more self-reactive T cells in the Lck^{CA/CA} mice, at least not to a large extent. This is also suggested by normal development of weakly self-reactive F5 T cells in the Lck^{CA/CA} mice (Extended Data Fig. 9a-d). Moreover, the hypothetical treatment with inhibitor(s) of the CD4-LCK and/or CD8-LCK interactions in adults would not probably severely affect the formation of the peripheral repertoire, because of the reduced thymic production at the adult age. Of course, there are multiple risks connected with such hypothetical therapy and this approach needs to be thoroughly tested in preclinical studies and eventually in clinical trials.

Decision Letter, first revision:

Our ref: NI-A33237A

19th Sep 2022

Dear Ondrej,

Thank you for submitting your revised manuscript "Unique roles of coreceptor-bound LCK in helper and cytotoxic T cells" (NI-A33237A). It has now been seen by the original referees and their comments are below. The reviewers find that the paper has improved in revision, and therefore we'll be happy in principle to publish it in Nature Immunology, pending minor revisions to satisfy the referees' final requests and to comply with our editorial and formatting guidelines.

We will now perform detailed checks on your paper and will send you a checklist detailing our editorial and formatting requirements in about a week. Please do not upload the final materials and make any revisions until you receive this additional information from us.

If you had not uploaded a Word file for the current version of the manuscript, we will need one before beginning the editing process; please email that to immunology@us.nature.com at your earliest convenience.

Thank you again for your interest in Nature Immunology Please do not hesitate to contact me if you have any questions.

Kind regards,

Laurie

Laurie A. Dempsey, Ph.D.
Senior Editor
Nature Immunology
l.dempsey@us.nature.com
ORCID: 0000-0002-3304-796X

Reviewer #1 (Remarks to the Author):

The authors have performed extensive analysis to support the idea that thymic selection is minimally affected in the key mouse models. After careful review of the newly added data, I believe that this point of view could largely be supported. Thus the major concern has been properly addressed.

Reviewer #3 (Remarks to the Author):

In the revised manuscript, the authors answered all my questions with experimental data. The revision for the discussion section is also sufficient. The experimental results to exclude the possibility of Lck CA mutation caused TCR signaling intensity change are quite impressive. It is convincing for the authors to conclude that there is no substantial intrinsic difference in the activity of the WT and CA variants of LCK. There are two more minor points:

1. Line 195, the description of the result "A negative role of CD8-bound kinase dead LCK in anti-viral and anti-tumor responses" may need to be rephrased in that LCK CA/KR phenotype change from LCK CA/CA could be the result of loss in one CA allele. The authors can conclude the negative role of KR allele by comparing LCK CA/KO versus LCK CA/KR if the latter gives rise to stronger phenotype. Or alternatively LCK CA/CA is equivalent to LCK CA/KO in function, which supports the conclusion of negative role of KR allele. Here the Lck KO genotype was not introduced to experiments. Is it more reasonable to stress the function of CA allele, instead of the negative role of KR?

2. Line 75/399, thorough language check throughout the manuscript is still needed.

Author Rebuttal, first revision:

Point-by-point response

Reviewer #1 (Remarks to the Author):

The authors have performed extensive analysis to support the idea that thymic selection is minimally affected in the key mouse models. After careful review of the newly added data, I believe that this point of view could largely be supported. Thus the major concern has been properly addressed.

We are happy that this Reviewer is satisfied with our revisions.

Reviewer #3 (Remarks to the Author):

In the revised manuscript, the authors answered all my questions with experimental data. The revision for the discussion section is also sufficient. The experimental results to exclude the possibility of Lck CA mutation caused TCR signaling intensity change are quite impressive. It is convincing for the authors to conclude that there is no substantial intrinsic difference in the activity of the WT and CA variants of LCK. There are two more minor points:

We are happy that this Reviewer is largely satisfied with our revisions.

1. Line 195, the description of the result "A negative role of CD8-bound kinase dead LCK in anti-viral and anti-tumor responses" may need to be rephrased in that LCK CA/KR phenotype change from LCK CA/CA could be the result of loss in one CA allele. The authors can conclude the negative role of KR allele by comparing LCK CA/KO versus LCK CA/KR if the latter gives rise to stronger phenotype. Or alternatively LCK CA/CA is equivalent to LCK CA/KO in function, which supports the conclusion of negative role of KR allele. Here the Lck KO genotype was not introduced to experiments. Is it more reasonable to stress the function of CA allele, instead of the negative role of KR?

We have modified this part of the manuscript to emphasize role of the LCK CA allele.

2. Line 75/399, thorough language check throughout the manuscript is still needed.

We have performed a thorough language check.

We are thankful to both Reviewers for their valuable comments and advice.

Final Decision Letter:

In reply please quote: NI-A33237B

Dear Ondrej,

I am delighted to accept your manuscript entitled "Unique roles of coreceptor-bound LCK in helper and cytotoxic T cells" for publication in an upcoming issue of Nature Immunology.

Over the next few weeks, your paper will be copyedited to ensure that it conforms to Nature Immunology style. Once your paper is typeset, you will receive an email with a link to choose the appropriate publishing options for your paper and our Author Services team will be in touch regarding any additional information that may be required.

Please note that *Nature Immunology* is a Transformative Journal (TJ). Authors may publish their research with us through the traditional subscription access route or make their paper immediately open access through payment of an article-processing charge (APC). Authors will not be required to make a final decision about access to their article until it has been accepted. [Find out more about Transformative Journals](https://www.springernature.com/gp/open-research/transformative-journals).

Authors may need to take specific actions to achieve [a compliance](https://www.springernature.com/gp/open-research/funding/policy-compliance-faqs) with funder and institutional open access mandates. If your research is supported by a funder that requires immediate open access (e.g. according to [Plan S principles](https://www.springernature.com/gp/open-research/plan-s-compliance))

then you should select the gold OA route, and we will direct you to the compliant route where possible. For authors selecting the subscription publication route, the journal's standard licensing terms will need to be accepted, including [self-archiving policies](https://www.springernature.com/gp/open-research/policies/journal-policies). Those licensing terms will supersede any other terms that the author or any third party may assert apply to any version of the manuscript.

Your paper will be published online soon after we receive your corrections and will appear in print in the next available issue. Content is published online weekly on Mondays and Thursdays, and the embargo is set at 16:00 London time (GMT)/11:00 am US Eastern time (EST) on the day of publication. Now is the time to inform your Public Relations or Press Office about your paper, as they might be interested in promoting its publication. This will allow them time to prepare an accurate and satisfactory press release. Include your manuscript tracking number (NI-A33237B) and the name of the journal, which they will need when they contact our office.

About one week before your paper is published online, we shall be distributing a press release to news organizations worldwide, which may very well include details of your work. We are happy for your institution or funding agency to prepare its own press release, but it must mention the embargo date and Nature Immunology. Our Press Office will contact you closer to the time of publication, but if you or your Press Office have any enquiries in the meantime, please contact press@nature.com.

Also, if you have any spectacular or outstanding figures or graphics associated with your manuscript - though not necessarily included with your submission - we'd be delighted to consider them as candidates for our cover. Simply send an electronic version (accompanied by a hard copy) to us with a possible cover caption enclosed.

If you have not already done so, we strongly recommend that you upload the step-by-step protocols used in this manuscript to the Protocol Exchange. Protocol Exchange is an open online resource that allows researchers to share their detailed experimental know-how. All uploaded protocols are made freely available, assigned DOIs for ease of citation and fully searchable through nature.com. Protocols

can be linked to any publications in which they are used and will be linked to from your article. You can also establish a dedicated page to collect all your lab Protocols. By uploading your Protocols to Protocol Exchange, you are enabling researchers to more readily reproduce or adapt the methodology you use, as well as increasing the visibility of your protocols and papers. Upload your Protocols at www.nature.com/protocolexchange/. Further information can be found at www.nature.com/protocolexchange/about .

Please note that we encourage the authors to self-archive their manuscript (the accepted version before copy editing) in their institutional repository, and in their funders' archives, six months after publication. Nature Portfolio recognizes the efforts of funding bodies to increase access of the research they fund, and strongly encourages authors to participate in such efforts. For information about our editorial policy, including license agreement and author copyright, please visit www.nature.com/ni/about/ed_policies/index.html

Kind regards,

Laurie

Laurie A. Dempsey, Ph.D.
Senior Editor
Nature Immunology
l.dempsey@us.nature.com
ORCID: 0000-0002-3304-796X